



# V2Karst V1.0: A parsimonious large-scale integrated vegetation-recharge model to simulate the impact of climate and land cover change in karst regions

Fanny Sarrazin[1], Andreas Hartmann[1,2], Francesca Pianosi[1], Thorsten Wagener[1,3]

[1] Department of Civil Engineering, University of Bristol, Bristol, BS8 1TR, UK
[2] Institute of Earth and Environmental Sciences, University of Freiburg, Germany
[3] Cabot Institute, University of Bristol, Bristol, BS8 1UJ, UK

*Correspondence to*: Fanny Sarrazin (fanny.sarrazin@bristol.ac.uk)

**Abstract.** Karst aquifers are an important source of drinking water in many regions of the world. Karst areas are highly permeable and produce large amounts of groundwater recharge, while surface runoff is typically negligible. As a result, recharge in these systems may have a different sensitivity to climate and land cover changes compared to other less permeable systems. However, little effort has been directed toward assessing the impact of climate and land cover change in karst areas at large-scales. In this study, we address this gap by (1) introducing the first large-scale hydrological model including an explicit representation of both karst and land cover properties, and by (2) analysing the model's recharge production behaviour. To achieve these points, we first improve the evapotranspiration estimation of a previous large-scale karst recharge model (VarKarst). The new model (V2Karst V1.0) includes a parsimonious representation of relevant ET processes for climate and land cover change impact studies. We demonstrate the plausibility of V2Karst simulations at carbonate rock FLUXNET sites using soft rules and global sensitivity analysis. Then, we use virtual experiments with synthetic data to assess the sensitivity of simulated recharge to precipitation characteristics and land cover. Results reveal how both vegetation and soil parameters control the model behaviour, and they suggest that simulated recharge is sensitive to both precipitation (overall amount and temporal distribution) and land cover. Large-scale assessment of future karst groundwater recharge should therefore consider the combined impact of changes in land cover and precipitation properties, if it is to produce realistic projections of future change impacts.



# 1. Introduction

Carbonate rocks, from which karst systems typically develop, are estimated to cover 10-15% of the world continental areas (Ford and Williams, 2007). Karst aquifers are an important source of drinking water for almost a quarter of the wold population (Ford and Williams, 2007) and have a critical role in sustaining food production because most karst areas present
some form of agricultural activity (Coxon, 2011). In particular, in Europe, carbonate rock areas cover 14-29% of the land area, and some European countries such as Austria and Slovenia derive up to 50% of their total water supply from karst aquifers (Chen et al., 2017; COST, 1995).

Karst systems are characterised by a high spatial variability of bedrock and soil permeability due to the presence of preferential flow pathways (Hartmann et al., 2014). The soluble carbonate bedrock is structured by large dissolution fissures
or conduits (Williams, 1983, 2008) and the typically clayey soil often contains cracks (Blume et al., 2010; Lu et al., 2016) where infiltrating water is concentrated. Therefore, a large part of the groundwater recharge occurs as concentrated and fast flow in large apertures and the other part as diffuse and slow flow in the matrix (Hartmann and Baker, 2017). Preferential flow pathways are particularly developed in karst, but they are also widely found in many other systems, due to root and organism activities, discontinuous subsurface layers, surface depressions, soil desiccation, tectonic processes and physical
and chemical weathering (Beven and Germann, 2013; Hendrickx and Flury, 2001; Uhlenbrook, 2006)

Preferential infiltration is typically triggered when thresholds on the rain intensity and soil moisture are exceeded (Rahman and Rosolem, 2017; Tritz et al., 2011). When activated, preferential infiltration pathways may enhance groundwater recharge while limiting surface runoff (e.g. (Bargués Tobella et al., 2014)). In karst, permeability is often so high that surface runoff is negligible, and virtually all precipitation infiltrates (Contreras et al., 2008; Fleury et al., 2007; Hartmann et al., 2014).
Furthermore, preferential infiltration pathways can affect the temporal dynamics of recharge. For instance (Cuthbert et al., 2013) showed that macro-pores in the soil can generate quick responses in the water table, and (Arbel et al., 2010) observed that dripping rates in a karst cave can fluctuate following precipitation inter-seasonal and intra-seasonal variations.

Changes in weather patterns (e.g. due to climate change), and specifically in the precipitation intensity and frequency, may
alter the activation of preferential flow pathways. From previous studies in non-karst areas we can learn that changes in the intensity and frequency of precipitation events have an impact on the water yield. For instance, using an analytical framework and synthetic experiments, (Porporato et al., 2004) established a dependency between the soil water balance and both the frequency and intensity of precipitation events, while (Jothityangkoon and Sivapalan, 2009) determined different theoretical hydrological regimes based on the intensity and frequency of the precipitation input over Australian catchments.
A modelling study by (Weiß and Alcamo, 2011) showed that for a given change in the total precipitation amount, a change in the intensity of precipitation events have a larger impact on water availability than a change in the number of wet days over European river basins. Regarding groundwater recharge, observation records are scarce but data indicate a sensitivity to



extreme rainfall in a semi-arid tropical region (Taylor et al., 2013) and in a seasonally humid tropical region (Owor et al., 2009).

In karst areas, few modelling studies showed that groundwater recharge (Hartmann et al., 2012; Loáiciga et al., 2000), spring discharge (Hao et al., 2006), and streamflow (Samuels et al., 2010) respond to changes in climate. However, to the

authors' knowledge, only one study by (Hartmann et al., 2017) analysed quantitatively the sensitivity of karst groundwater recharge to specific precipitation characteristics, namely the mean precipitation and the intensity of heavy precipitation events, and compared the results obtained over karst and non-karst areas. That study suggests that, due to the presence of preferential flow pathways, recharge in karst systems tends to show higher sensitivity to mean precipitation and to the intensity of heavy precipitation events in dry climates, and lower sensitivity in wet climates compared to non-karst systems.

Land cover/use change could also have a major impact on hydrological processes in the future (DeFries and Eshleman, 2004; Vörösmarty, 2002). Changes in land cover/use can impact the partitioning between green water (evapotranspiration losses) and blue water (water potentially available for human activities, namely groundwater recharge and runoff). Green water tends to be higher for forested areas than for shorter vegetation (e.g. (Brown et al., 2005)), which has also been found

in few local studies in karst areas (Ford and Williams, 2007; Williams, 1993). Significant land cover/use changes are expected to occur in the future, including in European and Mediterranean karst areas. These are partly due to modifications in socio-economic factors, such as changes in the food and wood demand or changes in the agricultural yields (see e.g. (Hurtt et al., 2011) for a global assessment and (Holman et al., 2017) for a European assessment). Future changing environmental conditions such as modifications in atmospheric $CO_2$, nitrate deposition and climate, and natural disturbances

such as wildfire, storm, bark beetle could also cause changes in land cover and in vegetation characteristics (e.g. leaf area index) (Seidl et al., 2014; Zhu et al., 2016).

The above review of the literature reveals that changes in climate characteristics (e.g. precipitation intensity and frequency) and in land cover properties are expected to have significant combined impacts on karst hydrology. Yet, the

impact of preferential pathways on the partitioning between green and blue water and the effect of climate and land cover change has not been studied systematically. Determining how sensitive groundwater recharge is to climate and land cover may change in the presence of preferential pathways, and therefore to what extent findings obtained for non-karst areas may be extrapolated to karst ones, is also essential to improve our understanding of future groundwater recharge at large-scales and ultimately to improve water resources management (Archfield et al., 2015). In this study we introduce a novel large-

scale model that includes explicit representation of both karst and vegetation properties and systematically explore the sensitivity of its simulated recharge to climate and vegetation inputs. Our model builds on an existing karst hydrology model, called VarKarst, which was recently developed for large-scale applications and demonstrated over European and Mediterranean carbonate rock areas (Hartmann et al., 2015). However, VarKarst has a very simplistic representation of



evapotranspiration and does not include land cover properties explicitly, which, up to now, prevented its application in land cover change impact studies.

The present study has two objectives that help us to overcome the previous limitation. First, we aim to add an explicit representation of land cover properties into VarKarst by improving the evapotranspiration (ET) estimation. While we seek to keep the model structure parsimonious, we want the new version of the model, called V2Karst (V1.0), to be appropriate for combined land cover and climate change impact studies. We test the plausibility of the V2Karst model behaviour by comparing its predictions against observations available at carbonate rock FLUXNET sites, and by analysing the dominant controls of simulated recharge. Second, we aim to understand the sensitivity of simulated groundwater recharge with V2Karst to changes in the vegetation characteristics and climate. We use a set of virtual experiments that allow us to control variations in climate and vegetation inputs, so that we can better explore their individual and combined effects on model outputs.

## 2 New version of VarKarst with explicit representation of land cover properties (V2Karst)

In this section, we first introduce our rationale to explicitly represent land cover properties into VarKarst (Sec. 2.1), we then briefly describe the previous ET component of VarKarst (Sect. 2.2) and we finally present the new V2Karst model (Sect. 2.3).

### 2.1 Rationale to explicitly represent land cover properties into VarKarst

The  new version of the VarKarst model should be appropriate to assess the impact of climate and land cover change on karst groundwater recharge, and should consider the range of challenges related to modelling ET at large-scales, namely a lack of ET observations to compare with model predictions, a lack of observations of vegetation properties (e.g. rooting depth, stomatal resistance, canopy interception storage capacity), and uncertainty in large-scale forcing weather variables (specifically air temperature, net radiation, humidity and wind speed). Further details on these three challenges are reported in Sect. S1 of our supplemental material. According to that, we define the three following criteria to represent ET in the VarKarst model:

1. **The model should assess separately all three main ET components (bare soil evaporation in presence of sparse canopy, transpiration and evaporation from canopy interception).** In fact, these fluxes exhibit different dynamics and sensitivity to environmental conditions and therefore, they are likely to respond differently to climate and land cover changes (Gerrits, 2010; Maxwell and Condon, 2016; Savenije, 2004; Wang and Dickinson, 2012).

2. **The model should use Penman-Monteith formulation for potential evapotranspiration (PET)** (Monteith, 1965), to separate the effects of climate and land cover and assess specific rates for the different ET components. In





fact, empirical PET formulations such as the Priestley-Taylor equation (Priestley and Taylor, 1972) do not represent explicitly land cover properties.

3. **All processes should be represented parsimoniously** in accordance with the modelling philosophy underpinning the first version of VarKarst (Hartmann et al., 2015).This criteria aims to avoid over-parameterisation given the limited amount of available information to constrain and test model simulations at large-scales, to limit the computational time for model simulations and to allow for assessing the impact of modelling choices and the uncertainty and sensitivity of model output using Monte-Carlo simulation.

We review the different approaches currently used to represent explicitly land cover properties in existing hydrological models of karst areas and in existing large-scale models, to assess their consistency with the three above-mentioned criteria and to determine whether we could directly adopt some of these ET representations for VarKarst. We report a summary of our findings in the following paragraphs, while more details on the processes and parameterisations of the ET components of other large-scale models are reported in Tables (A1-A3), and a detailed list of all parameters involved in the representation of ET in large-scale models can be found in Section S2 of our supplementary material.

With respect to existing models of karst areas, to the authors' knowledge, only four models that explicitly include ET and land cover processes were applied in karst studies, all of which were local studies where detailed on-site information was available. Three of these models (Canora et al., 2008; Doummar et al., 2012; Zhang et al., 2011) were not specifically developed for karst areas, but they are distributed models that simply utilised the flexibility of their parameters to represent the variability in soil and bedrock properties. These models are heavily parameterised, which hampers their application at large-scales, and does not comply with criterion 3 (parsimony). The fourth model introduced in (Sauter, 1992) is lumped and is much more parsimonious than the three other models. However, the model does not represent soil evaporation, and uses empirical PET equations, which does not allow to separate the effect of climate and land cover (disagreement with criteria 1 and 2). Moreover, the model of (Sauter, 1992) has a rather sophisticated interception routine, which includes both canopy and trunk interception.

Regarding the representation of ET in existing large-scale models, our review showed that we cannot directly adopt any of their ET representation into VarKarst. In fact, as shown in Tables (A1-A3), parsimonious models (WBM, LaD and WaterGap) neglect some ET components and/or use empirical PET equations, which contradicts criteria 1 and 2, while models that comply with criteria 1 and 2 (PCR-GLOBWB, ISBA and VIC and the model of (Kergoat, 1998)) use heavily parameterised schemes, such as a Jarvis type parameterisation of surface resistance (Jarvis, 1976; Stewart, 1988), and therefore do not satisfy criterion 3 (parsimony). Moreover, we found that large-scale models include empirical schemes with no clear origin, such as the reference crop formulation used in the PCR-GLOBWB model for PET calculation or the interception model used in LPJ and in the model of (Kergoat, 1998). Importantly, our review revealed the tremendous variability of approaches used in large-scale models when it comes to representing ET processes. Consequently, no clear



indication emerged regarding a 'best way' to parameterise the different ET processes at large-scales, which leaves us with a large range of different formulations to choose from to implement an explicit representation of land cover processes into VarKarst.

The next sections provide more details on the specific assumptions and choices made to develop the new ET component for the VarKarst model, which satisfies the three criteria defined in this section and utilises some of the schemes from other large-scale models.

## 2.2 Previous representation of ET processes in VarKarst

VarKarst [Hartmann et al., 2015] is currently the only karst recharge model developed for large-scale applications. It is a conceptual semi-distributed model that simulates daily karst potential recharge (Fig. 1.a). VarKarst includes two horizontal subsurface layers, a top layer called 'soil' and a deeper layer called 'epikarst'. The soil layer corresponds to the layer from which ET can occur. The epikarst layer corresponds to the uppermost layer of weathered carbonate rocks where it is assumed that water cannot be lost through ET. Groundwater recharge predicted by VarKarst includes both the diffuse and concentration fractions, because for each model grid cell, the water balance is evaluated separately over a number of vertical compartments with varying soil and epikarst properties. The ET component of the VarKarst model is very simple and does not include explicit representation of land cover properties. ET is lumped in the soil layer and is estimated from PET and reduced by a water stress factor, which is estimated as a linear function of soil moisture. The PET rate is calculated with the empirical Priestley-Taylor equation (Priestley and Taylor, 1972) using a spatially and temporally uniform value of the empirical coefficient. This approach does not allow to separate the effect of climate and land cover, since the empirical coefficient reflects both climate and vegetation characteristics simultaneously. Therefore, the ET component of VarKarst needs to be modified if the model is to be used for large-scale land cover change impact assessment.

## 2.3 V2Karst: the new version of VarKarst for integrated vegetation-recharge simulations over karst areas

In this section, we propose a new version of the VarKarst model, called V2Karst (Figure 1b). In accordance with the criteria 1 and 2 defined in Sect. 2.1, compared to VarKarst, the new V2Karst model (1) includes a physically based PET equation, (2) separates the evapotranspiration flux into three components (transpiration, bare soil evaporation and evaporation from canopy interception), (3) comprises three soil layers. Additionally, V2Karst represents parsimoniously the seasonal changes in the vegetation properties, which will allow us to analyse the importance of this process on simulated recharge. We assumed homogeneous above ground vegetation properties across model compartments.

We note that V2Karst has a total of 15 parameters (described in Table 1 and Figure 1), including the 4 parameters of VarKarst and 11 new parameters in the new ET component, that replaces the Priestley-Taylor empirical coefficient $\alpha$ used in VarKarst. In agreement with criteria 3 of Sect. 2.1 (parsimony), we sought to represent parsimoniously the different ET processes into VarKarst. In fact, V2Karst uses 12 parameters to represent ET and vegetation seasonality (including the 11



newly introduced parameters and the soil water capacity parameters $V_{soi}$ already present in VarKarst). This is less than other existing large-scale models that use Penman-Monteith equation and separate the three ET components, since these models have over 15 parameters in their ET component (PCR-GLOBWB, ISBA and VIC and model of (Kergoat, 1998) in Tables A1-A3). The new model is forced by time series of precipitation $P$, air temperature $T$ and net radiation $R_n$ as VarKarst.
5    Additionally, time series of relative humidity RH and wind speed WS are now needed for PET calculation.

### 2.3.1 Definition of soil and epikarst properties in V2Karst

The computation of water storage capacity of the entire soil column $V_{S,i}$ [mm] and of the epikarst $V_{E,i}$ [mm], and the epikarst outflow coefficient $K_{E,i}$ [d] for the $i$th model compartment is done as before in VarKarst:

$$V_{S,i} = V_{max,S}\left(\frac{i}{n_c}\right)^a,$$

$$V_{E,i} = V_{max,E}\left(\frac{i}{n_c}\right)^a,$$

$$K_{E,i} = K_{max,E}\left(\frac{n_c - i + 1}{n_c}\right)^a. \tag{1}$$

where $V_{max,S}$ [mm] is the maximum soil storage capacity over all model compartments, $V_{max,E}$ [mm] is the maximum
10   epikarst storage capacity, $K_{max,E}$ [d] is the maximum outflow coefficient, $n_c$ [−] is the number of model compartments, which is set to 15 following (Hartmann et al., 2013, 2015) and $a$ [−] is the spatial variability coefficient. A previous study showed that $V_{S,i}$, $V_{E,i}$ and $K_{E,i}$ can be determined using the same distribution coefficient $a$ (Hartmann et al., 2013). In V2Karst, $V_{max,S}$, $V_{max,E}$ and $K_{E,i}$ are computed as a function of the average properties of the cell using the following the formulas:

$$V_{max,S} = \frac{V_{soi} n_c}{\sum_{i=1}^{n}\left(\frac{i}{n_c}\right)^a},$$

$$V_{max,E} = \frac{V_{epi} n_c}{\sum_{i=1}^{n}\left(\frac{i}{n_c}\right)^a},$$

$$K_{max,E} = \frac{K_{epi} n_c}{\sum_{i=1}^{n}\left(\frac{i}{n_c}\right)^a}, \tag{2}$$



where $V_{soi}$ [mm] is the mean soil storage capacity, $V_{epi}$ [mm] is the mean epikarst storage capacity and $K_{epi}$ [mm] is the mean epikarst outflow coefficient. We note that the definition of the three parameters $V_{soi}$, $V_{epi}$ and $K_{epi}$ is revised compared to VarKarst.

As in VarKarst, we neglect ET from the epikarst. Several studies showed that in presence of shallow soil and dry climate,
plants can take up water in the weathered bedrock where soil pockets can sustain roots development (Schwinning, 2010). However, given the uncertainty in soil depth for large-scale applications, V2Kast does not allow ET from the epikarst to avoid over-parameterisation. Therefore, the V2Karst soil layer must be interpreted as a conceptual layer that does not exactly correspond to the physical soil layer (layer of loose material), but is defined as the portion of the subsurface where ET losses can occur.

In V2Karst, the soil layer is further divided into a shallow top layer from which water can be lost from both evaporation and transpiration, a second middle layer where only transpiration can occur and a third deeper layer below the root zone where transpiration can only take place when the first two layers are depleted. The maximum storage capacity of the first layer is noted as $V_e$ [mm], and the maximum storage capacity of first and second layers combined is noted as $V_r$ [mm],which corresponds to the maximum storage capacity of the root zone. The model assumes that $V_e$ is smaller than $V_r$, which is in turn
smaller than the storage capacity of the deeper model compartment $V_{S,n}$.

**2.3.2 Soil water balance**

The soil water storage $V_{soi,i}^j(t)$ [mm] in the $i$th compartment and the $j$th soil layer $j = 1,2,3$ is updated at the end of each time step $t$ as follows:

$$V_{soi,i}^1(t) = V_{soi,i}^1(t-1) + T_f(t) + Q_{lat,i-1\to i}(t) - Es_{act,i}(t) - T_{act,i}^1(t) - R_{12,i}(t),$$
$$V_{soi,i}^2(t) = V_{soi,i}^2(t-1) + R_{12,i}(t) - T_{act,i}^2(t) - R_{23,i}(t), \qquad (3)$$
$$V_{soi,i}^3(t) = V_{soi,i}^3(t-1) + R_{23,i}(t) - T_{act,i}^3(t) - R_{epi,i}(t).$$

Where $T_f(t)$ [mm] is the throughfall i.e. the fraction of precipitation that is not evaporated from the interception store,
$Q_{lat,i-1\to i}(t)$ [mm] is the lateral flow from the $(i-1)$th to the $i$th model compartment (Sect. 2.3.4), $Es_{act,i}(t)$ [mm] is the actual soil evaporation (Eq. (7)), $T_{act,i}^j(t)$ [mm] is the actual transpiration in the $j$th soil layer (Eq. (9-10)), $R_{12,i}(t)$ [mm] is the downward flow from the first to the second soil layer, $R_{23,i}(t)$ [mm] is the downward flow from the second to the third soil layer and $R_{epi,i}(t)$ [mm] is the downward flow from the soil to the epikarst.

It is assumed that percolation from the unsaturated soil to the epikarst is negligible due to low permeability of the soil. This
assumption seems reasonable since karst soils usually have a high clay content (Blume et al., 2010; Clapp and Hornberger, 1978). However, clayey soil typically present cracks (Lu et al., 2016), and therefore when the soil reaches saturation, preferential flow starts to occur in the soil cracks, which causes all saturation excess to quickly infiltrate to the epikarst. Just



as in VarKarst, such preferential vertical flow is represented by the variable $R_{epi,i}(t)$ (used in Eq. (3)) and is set equal to the saturation excess in the (lowest) soil layer. In V2Karst, a similar approach is also used to assess the other vertical flows from one soil layer to another ($R_{12,i}(t)$ and $R_{23,i}(t)$) in Eq. (3).

### 2.3.3 Evapotranspiration

We adopt the representation of sparse vegetation proposed by (Bohn and Vivoni, 2016) for the VIC model and referred to as 'clumped' vegetation scheme. Each model compartment is divided into a vegetated and a non-vegetated fraction using a canopy cover fraction coefficient $f_c(t)$ [−]. The uptake of soil moisture for transpiration and soil evaporation is coupled in a way that, for each model compartment, we evaluate an overall water balance over the two fractions. Using such a coupled approach facilitates the representation of the seasonal variations in vegetated and non-vegetated fractions compared to an uncoupled 'tile' approach, in which a separate soil moisture state is represented for vegetated and bare soil fractions. Consistently with other existing large-scale models, aerodynamic interactions between both fractions are neglected to keep the number of parameters to a minimum (Table A3).

The canopy coefficient $f_c(t)$ is estimated in V2Karst using the Beer-Lambert's law as in (Van Dijk and Bruijnzeel, 2001; Ruiz et al., 2010). This law has been originally used to separate the fraction of incident radiation (and by extension of net radiation) absorbed by the canopy from the fraction penetrating the canopy (Kergoat, 1998; Ross, 1975; Shuttleworth and Wallace, 1985). The canopy cover fraction at time $t$ is expressed as a function of the cell average leaf area index $LAI(t)$ [$m^2 . m^{-2}$] and an extinction coefficient $k$ [−], which is understood to vary across vegetation type since it accounts for leaf architecture (Ross, 1975):

$$f_c(t) = 1 - e^{-kLAI(t)}. \tag{4}$$

Notice that Eq. (4) allows to describe the seasonal variations in canopy cover fraction without introducing additional parameters in the model, given that they will simply follow the seasonal variations in $LAI$.

### Canopy interception

It has been shown that a simple parameterization of daily interception can give reasonable simulation results (Gerrits, 2010; De Groen, 2002; Savenije, 1997). Following these studies, in V2Karst, interception is represented by a daily threshold model. Our formulation is as follows:

$$Ec_{act}(t) = f_c(t) min\left(Ec_{pot}(t), P(t), V_{can,max}(t)\right), \tag{5}$$

where $Ec_{pot}(t)$ [mm] is the potential evaporation from canopy interception (Eq. (12)), $P(t)$ [mm] is the precipitation and $V_{can,max}(t)$ [mm] is the interception storage capacity over the vegetated fraction of the cell (Eq. (6)). The factor $f_c(t)$ in Eq. (5) accounts for the fact that evaporation from canopy occurs over the vegetated fraction only. We note that the potential rate $Ec_{pot}(t)$ was not accounted for in the original formulation by (Gerrits, 2010; De Groen, 2002; Savenije, 1997). The





interception storage capacity over the vegetated fraction $V_{can,max}$ [mm] depends (1) on the leaf area index over the vegetated fraction, which is estimated by rescaling cell average leaf area index $LAI(t)$ using the vegetation cover fraction $f_c(t)$ following (Bohn and Vivoni, 2016), and (2) on the canopy storage capacity per unit of leaf area index, denoted by $V_{can}$, which is understood to depend on the vegetation type since it accounts for leaf architecture (Gerrits, 2010)It is expressed as:

$$V_{can,max}(t) = V_{can}\left(\frac{\text{LAI}(t)}{f_c(t)}\right). \tag{6}$$

The model does not account for the carry-over of interception storage from one day to the next, which means that all precipitation which is not evaporated from the interception store reaches the ground as throughfall $T_f$ [mm]. This assumption can be justified by the fact that the interception process is highly dynamic at a sub-daily time scale, because the canopy can go through several wetting-drying cycles within a day (Gerrits, 2010). Therefore, when evaporation from canopy interception is estimated with a daily time step as in V2Karst, the canopy layer must be interpreted as a conceptual layer,

whose storage capacity does not exactly correspond to the physical storage capacity of the canopy (i.e. the amount of water that can be hold at a given time), but to the cumulative amount of water that can be hold by the canopy over a day (Gerrits, 2010).

**Bare soil evaporation**

It is assumed that soil evaporation is a faster process than transpiration consistently with general knowledge on ET processes

(Wang and Dickinson, 2012). Therefore, soil moisture can be first evaporated and then transpired if some available moisture remains for plant water uptake. Soil evaporation is withdrawn for the first soil layer as a function of the potential rate and soil moisture, similar to the previous version of VarKarst:

$$Es_{act,i}(t) = \min\left((1-f_c(t))Es_{pot}(t)\frac{V_{soi,i}^1(t-1)}{V_{S,i}^1}, V_{soi,i}^1(t-1) + T_f(t)\right), \tag{7}$$

where $Es_{pot}(t)$ is the potential soil evaporation (Eq. (12)). The factor $(1-f_c(t))$ in Eq. (7) accounts for the fact that soil evaporation occurs from the non-vegetated fraction only and therefore the potential rate has to be weighted by the bare soil

cover fraction. The right term of the equation $(V_{soi,i}^1(t-1) + T_f(t))$ is not weighted because we assume that the soil moisture is uniform over the fractions of each model compartment (we compute a unique water balance) and therefore the total moisture present in the first soil layer is available to soil evaporation because the vegetated fraction can supply moisture to the bare soil fraction.



**Transpiration from vegetated soil**

Transpiration mainly occurs in the first and second soil layers, and it switches to the third soil layer when the first two layers are depleted. The extraction of water by the roots below the root zone is documented in (Penman, 1950) and we account for this process by representing a soil layer below the root zone, which can provide water to the root zone through capillary rise as in the ISBA model (Boone et al., 1999). In V2Karst, the rate at which transpiration occurs in the two first soil layers $T_{rate,i}^{12}(t)$ [mm] and in the third soil layer $T_{rate,i}^{3}(t)$ [mm] are assessed as follows:

$$T_{rate,i}^{12}(t) = \left(1 - t_{wet}(t)\right) f_c(t) T_{pot}(t) \frac{V_{soi,i}^1(t-1) + V_{soi,i}^2(t-1)}{V_{S,i}^1 + V_{S,i}^2},$$

$$T_{rate,i}^{3}(t) = \left(1 - t_{wet}(t)\right) f_c(t) T_{pot}(t) \frac{V_{soi,i}^3(t-1)}{V_{S,i}^3} f_{red}. \tag{8}$$

Where $T_{pot}(t)$ is the potential transpiration (Eq. (12)), $t_{wet}(t)$ [−] is the fraction of the day with wet canopy (Eq. (11)) and $f_{red}$ [−] is a reduction factor which accounts for the fact that moisture below the root zone is less easily accessible to the roots than moisture in the root zone (Penman, 1950), and which is expected to vary across soil type since it is linked to the soil capability to supply water to the root zone . It is assumed that transpiration occurs in the two first soil layers when $T_{rate,i}^{12}(t)$ is higher than $T_{rate,i}^{3}(t)$, and that transpiration is drawn from the third soil layer otherwise. The actual transpiration in the two first soil layers $T_{act,i}^{12}(t)$ [mm] and in the third soil layer $T_{act,i}^{3}(t)$ [mm] are therefore calculated as follows:

when $T_{rate,i}^{12}(t) \geq T_{rate,i}^{3}(t)$:

$$\begin{cases} T_{act,i}^{12}(t) = \min\left(T_{rate,i}^{12}(t), V_{soi,i}^1(t-1) + V_{soi,i}^2(t-1) + T_f(t) - Es_{act,i}(t)\right), \\ \qquad\qquad\qquad T_{act,i}^3(t) = 0, \end{cases}$$

when $T_{rate,i}^{12}(t) < T_{rate,i}^{3}(t)$: $\tag{9}$

$$\begin{cases} \qquad\qquad T_{act,i}^{12}(t) = 0, \\ T_{act,i}^3(t) = \min\left(T_{rate,i}^3(t),\ V_{soi,i}^3(t-1) + R_{23,i}(t)\right). \end{cases}$$

Actual transpiration in the upper two layers $T_{act,i}^{12}(t)$ is partitioned between the two soil layers within the root zone as is used in the PCR-GLOBWB model (Van Beek, 2008). In V2Karst, the transpiration is attributed to the two first soil layers proportional to their storage content. This simple representation assumes that the roots can equally access the moisture stored in the first and second layer. Actual transpiration from the first layer $T_{act,i}^{1}(t)$ [mm] and the second layer $T_{act,i}^{2}(t)$ [mm] are computed as follows:

$$T_{act,i}^1(t) = \frac{V_{soi,i}^1(t-1) + T_f(t) - Es_{act,i}(t)}{V_{soi,i}^1(t-1) + T_f(t) - Es_{act,i}(t) + V_{soi,i}^2(t-1)} T_{act,i}^{12}(t), \tag{10}$$


$$T_{act,i}^2(t) = \frac{V_{soi,i}^2(t-1)}{V_{soi,i}^1(t-1) + T_f(t) - Es_{act,i}(t) + V_{soi,i}^2(t-1)} T_{act,i}^{12}(t).$$

In V2Karst, it is assumed that transpiration occurs when the canopy is dry only, as it is typically done in the other large-scale models. The fraction of the day with wet canopy $t_{wet}(t)$ [−] is estimated by assuming that the actual rate of evaporation from interception is constant throughout the day and is equal to the potential rate similar to (Kergoat, 1998):

$$t_{wet}(t) = \frac{Ec_{act}(t)}{f_c(t)Ec_{pot}(t)} \tag{11}$$

**Potential evapotranspiration**

We replace the Priestley-Taylor potential evaporation equation used in the previous version of the model by the Penman-Monteith equation (Monteith, 1965). Potential transpiration rate over the vegetated fraction of the cell $T_{pot}(t)$ [mm] is estimated from the canopy aerodynamic resistance $r_{a,can}(t)$ [s.m$^{-1}$] and surface resistance $r_{s,can}(t)$ [s.m$^{-1}$], potential evaporation from interception over the vegetated fraction of the cell $Ec_{pot}(t)$ [mm] is assessed assuming that the surface resistance is equal to 0 following e.g. (Shuttleworth, 1993), while potential bare soil evaporation rate over the bare soil

fraction of the cell $Es_{pot}(t)$ [mm] is calculated from the soil aerodynamic resistance $r_{a,soi}(t)$ [s.m$^{-1}$] and surface resistance $r_{s,soi}$ [s.m$^{-1}$], using the following equations:

$$T_{pot}(t) = \frac{\Delta(t)R_n(t) + K_t\rho_a(t)c_p\dfrac{e_s(t)-e_a(t)}{r_{a,can}(t)}}{\lambda(t)\left(\Delta(t) + \gamma(t)\left(1 + \dfrac{r_{s,can}(t)}{r_{a,can}(t)}\right)\right)},$$

$$Ec_{pot}(t) = \frac{\Delta(t)R_n(t) + K_t\rho_a(t)c_p\dfrac{e_s(t)-e_a(t)}{r_{a,can}(t)}}{\lambda(t)\left(\Delta(t) + \gamma(t)\right)}, \tag{12}$$

$$Es_{pot}(t) = \frac{\Delta(t)R_n(t) + K_t\rho_a(t)c_p\dfrac{e_s(t)-e_a(t)}{r_{a,soi}(t)}}{\lambda(t)\left(\Delta(t) + \gamma(t)\left(1 + \dfrac{r_{s,soi}}{r_{a,soi}(t)}\right)\right)}.$$

where $\lambda(t)$ [MJ.kg$^{-1}$] is the latent heat of vaporization of water, $\Delta(t)$ [kPa.°C$^{-1}$] is the gradient of the saturated vapour pressure-temperature function, $\gamma(t)$ [kPa.°C$^{-1}$] is the psychrometric constant $\rho_a(t)$ [kg.m$^{-3}$] is the air density, $c_p$ [MJ.kg$^{-1}$.°C$^{-1}$] is the specific heat of the air and is equal $1.013.10^{-3}$MJ.kg$^{-1}$.°C$^{-1}$, $e_s(t)$ [kPa] is the saturation vapor

pressure, $e_a(t)$ [kPa] is the actual vapor pressure, and $K_t$ [s.d$^{-1}$] is a time conversion factor which corresponds to the





number of seconds per simulation time step equal to $86{,}400 \, \text{s.d}^{-1}$. We neglect ground heat flux, which seems to be reasonable for daily calculations (see e.g. (Allen et al., 1998; Shuttleworth, 2012)).

The aerodynamic resistances of canopy ($r_{a,can}(t)$) and of the soil ($r_{a,soi}(t)$), that depend on the properties of the land cover and the soil respectively, are computed using the formulation of (Allen et al., 1998). To assess $r_{a,can}(t)$), roughness lengths

and zero displacement plane for the canopy are estimated from the vegetation height $h_{veg}$ [m] (Allen et al., 1998). To calculate $r_{a,soi}(t)$), the zero plane displacement height is equal to zero ($d = 0$) and the roughness length for momentum and for heat and water vapor transfer are assumed to be equal, as in (Šimůnek et al., 2009), and denoted as $z_0$ [m].

Finally, the canopy surface resistance is computed by scaling the stomatal resistance $r_{st}$ [s. m$^{-1}$] to canopy level using the leaf area index over the vegetated fraction (as in Eq. (6) to assess canopy interception capacity), and therefore assuming a

homogeneous response across all stomata in the canopy (Allen et al., 1998; Liang et al., 1994):

$$r_{s,can}(t) = \frac{r_{st}}{\left(\frac{LAI(t)}{f_c(t)}\right)}. \tag{13}$$

In other large-scale models, $r_{s,can}$ is also often expressed as a function of $LAI$, which allows to directly represents its seasonality following the variations in $LAI$.

**Seasonality of vegetation**

We represent the seasonality of vegetation by describing the seasonal variation of the cell average leaf area index $LAI$. We

use two parameters, the maximum $LAI_{max}$ [m$^2$. m$^{-2}$], which is the annual maximum value of $LAI$ during the growing season (assumed to be from June to August) and $LAI_{min}$ [%], which is the percentage of reduction in $LAI$ during the dormant season (assumed to be from December to February). The monthly value of leaf area index $LAI_m$ [m$^2$. m$^{-2}$] for the $m^{th}$ month is computed using a continuous, piecewise linear function of $LAI_{max}$ and $LAI_{min}$, which allows for a smooth transition between dormant and growing seasons and is similar to the function proposed by (Allen et al., 1998) to assess the seasonality

in crop factors:

$$
\begin{aligned}
LAI_m &= \frac{LAI_{min}}{100} LAI_{max} & \text{when } m = 1, 2, 12 \\
LAI_m &= \frac{LAI_{min}}{100} \frac{LAI_{max}}{4} (6 - m) + \frac{LAI_{max}}{4} (m - 2) & \text{when } m = 3, 4, 5 \\
LAI_m &= LAI_{max} & \text{when } m = 6, 7, 8 \\
LAI_m &= \frac{LAI_{min}}{100} \frac{LAI_{max}}{4} (m - 8) + \frac{LAI_{max}}{4} (12 - m) & \text{when } m = 9, 10, 11.
\end{aligned}
\tag{14}
$$

The advantage of using this simple parameterisation is that it permits to easily analyse the effect of vegetation seasonality by studying the sensitivity of the model predictions to parameter $LAI_{min}$, which captures the strength of the seasonal variation in $LAI$.



### 2.3.4 Water storage in the epikarst

Epikarst water storage $V_{epi,i}(t)$ [mm] for the $i$th compartment is updated at the end of each time step $t$ as follows:

$$V_{epi,i}(t) = V_{epi,i}(t-1) + R_{epi,i}(t) - Q_{epi,i}(t) - Q_{lat,i \to i+1}(t) \qquad \text{when } i < n_c,$$

$$V_{epi,n_c}(t) = V_{epi,n_c}(t-1) + R_{epi,n_c}(t) - Q_{epi,n_c}(t) - Q_{surf,n_c}(t) \text{ when } i = n_c. \tag{15}$$

where $Q_{epi,i}(t)$ [mm] is the potential recharge to the groundwater (Eq. (16)), $Q_{lat,i \to i+1}(t)$ [mm] is the lateral flow from the $i$th to the $(i+1)$th model compartment and $Q_{surf,n_c}(t)$ [mm] is the surface runoff generated by the $n_c$th compartment.

When soil and epikarst layers are saturated, the concentration flow component of the model is activated. The $i$th model compartment generates lateral flow towards the $(i+1)$th compartment $Q_{lat,i \to i+1}(t)$ [$mm$] equal to its saturation excess. Lateral flow from the $n_c$th compartment is lost from the cell as surface runoff while the other model compartments do not produce any surface runoff. The epikarst is simulated as a linear reservoir (Rimmer and Hartmann, 2012) with outflow coefficient $K_{E,i}$ [d]:

$$Q_{epi,i}(t) = \min\left(\frac{V_{epi,i}(t-1)}{K_{E,i}}, V_{epi,i}(t-1) + R_{epi,i}(t)\right). \tag{16}$$

## 3. Site and data for model testing

### 3.1 Site description

We test the model with plot scale measurements from sites of the FLUXNET network (Baldocchi et al., 2001). We identified four FLUXNET sites across European and Mediterranean carbonate rock areas for which sufficient data are available to force V2Karst and to test the model (see Sect. 3.2). A short summary of the sites' characteristics is provided in Figure 2 and
more detailed information can be found in Table B1.

The sites have different climate and land cover properties. The first site (Hainich site, referred to as 'German site') is located in the protected Hainich National Park, Thuringia, central Germany, and is characterised by a suboceanic-submountain climate and a tall and dense deciduous broadleaf forest. The second site (Llano de los Juanes site referred to as 'Spanish site') is located on a plateau of the Sierra de Gádor mountains, south-eastern Spain, has a semi-arid mountain
Mediterranean climate and is an open shrubland. The third site (Font-Blanche site, referred to as 'French 1 site') is located in south-eastern France, has a Mediterranean climate and its land cover is medium-height mixed evergreen forest. The fourth site (Puéchabon site, referred to as 'French 2 site') is located in southern France and is characterised by a Mediterranean climate with a short evergreen broadleaf forest. Overground vegetation properties are well characterised at all sites, but subsurface properties are more uncertain. In particular, the rooting depth water capacity was only well investigated at the
French 2 site. The four sites are appropriate for testing V2Karst since they satisfy the model assumptions, namely a karstified



or fissured and fractured bedrock, overall high infiltration capacity with limited surface runoff and high clay content in the soil (Table B1).

## 3.2 Data description and preparation

Data available at the four FLUXNET sites include measurements of precipitation, temperature, net radiation, relative humidity and wind speed to force the model, and eddy-covariance measurements of latent heat and at the German and Spanish sites measurements of soil moisture to estimate the model parameters (Sect. 4.1). Specifically, at the German site, soil moisture was measured in one vertical soil profile at three different depths (5, 15  and 30 cm) with Theta-probes (Knohl et al., 2003). We selected the measurement at 30 cm depth, which we deem to be most representative of the entire soil column which has a depth between 50 and 60 cm. At the Spanish site, soil moisture was assessed at a depth of 15 cm using a water content reflectometer (Pérez-Priego et al., 2013).

Regarding the data processing, data to force the model were gap-filled and aggregated from 30 min to daily time scale. V2Karst output observations, namely latent heat and soil moisture measurements, were aggregated from 30 min to monthly time scale and we discarded the months when more than 20 % of 30 min data were missing. We also removed monthly aggregated latent heat measurements when the mismatch in the energy balance closure was higher than 50% similar to (Miralles et al., 2011). Additionally, we discarded the monthly observations of latent heat and soil moisture for months in which the forcing data contain many gaps, and therefore the impact of the gap-filling of the data on the simulation results is likely to be too significant to sensibly compare simulated and observed soil moisture and latent heat. Further details on the data processing and is reported in Section S4 of our Supplementary material.

Table 2 reports the simulation period and the number of monthly latent heat and soil moisture observations that were used to estimate the model parameters at the four FLUXNET sites. We extracted a continuous time series of forcing data covering about 10 years at the German site, 7 years at the Spanish site, 3 years at the French 1 site and 8 years at the French 2 site, while latent heat and soil moisture measurements are not available over the entire simulation time series. All model simulations were performed using a one-year warmup period, which we found to be sufficient to remove the impact of the initial conditions on the simulation results (see Sect. S5 of our supplementary material).

Moreover, we corrected latent heat measurements and analysed their uncertainty. We derived two corrected estimates of actual ET, obtained by forcing the closure in the energy balance following (Foken et al., 2012; Twine et al., 2000), namely:

1. a corrected value that assumes that latent heat ($LE$ [$\mathrm{MJ. m^{-2}. month^{-1}}$])  and sensible heat ($H$ [$\mathrm{MJ. m^{-2}. month^{-1}}$]) have similar errors (referred to as Bowen ratio estimate):

$$E_{act,bow} = \frac{R_n}{\lambda. \left(1 + \frac{H}{LE}\right)} \; [\mathrm{mm. month^{-1}}], \tag{17}$$





2.  a corrected value that assumes errors in latent heat only (referred to as residual estimate):

$$E_{act,res} = \lambda.(R_n - H) \ [mm.month^{-1}]. \tag{18}$$

An additional analysis showed that the two corrected estimates of Eq. (17-18) and the uncorrected measure of actual ET are well correlated at the FLUXNET sites, which gives us some confidence regarding the temporal variations in actual ET measurements, while relative errors between corrected and uncorrected estimates can be quite high (see Section S4 of our

Supplementary material). We chose to use the Bowen ratio estimate (Eq. (17)) to calibrate the model. In fact, it is not clear whether one of the two turbulent fluxes may be more uncertain than the other (Foken et al., 2012).

## 4. Methods

In this study, we estimate V2Karst parameters and test the plausibility of model realisations at the FLUXNET sites (Sect. 4.1), we conduct a global sensitivity analysis of the model parameters at the FLUXNET sites to identify the model dominant

controls and inform model calibration for future applications (Sect. 4.2) and we last, perform a set of virtual experiments to learn about the mechanism of recharge production in the model and its sensitivity to precipitation characteristics and land cover type (Sect. 4.3). All the analyses were performed using the SAFE toolbox for global sensitivity analysis (Pianosi et al., 2015).

## 4.1 Parameter estimation at the FLUXNET sites using soft rules

We investigate whether it is possible to estimate parameter values that produce plausible simulations based on information available at each FLUXNET site. To this end, and similarly to (Hartmann et al., 2015), we use 'soft rules' to accept or reject parameter combinations based on the consistency between monthly model simulations on one side, and monthly observations and a priori information on model fluxes on the other side. Using soft rules instead of 'hard rules' (i.e. minimisation of the mismatch between observations and simulations) allows to identify a set of plausible model simulations

and accounts for the fact that (1) the observed soil moisture is not strictly commensurate with simulated soil moisture, (2) observations are affected by uncertainties (see Sect. 3.2) and (3) it is not expected that V2Karst simulations closely match site-specific data, since the model structure is based on general understanding of karst systems for large-scale applications and may not account for some site specificities. We define five soft rules to identify acceptable ('behavioural') parameter combinations:

1.  **The bias between observed and simulated actual ET is below 20**%:

$$Bias = \left| \frac{\sum_{t \in M_{ET}}(E_{act,sim}(t) - E_{act,bow}(t))}{\sum_{t \in M_{ET}} E_{act,bow}(t)} \right| < 20\%, \tag{19}$$

where $E_{act,sim}(t)[mm]$ is the simulated actual ET for month $t$ (sum of transpiration, soil evaporation and evaporation from canopy interception), $E_{act,bow}(t)[mm]$ is the Bowen ratio correction of observed actual ET (Eq.



(17)), and $M_{ET}$ is the set of months for which latent heat measurements are available. This rule allows to constrain the simulated water balance.

2. **The correlation coefficient ($\rho_{ET}$) between observed monthly actual ET ($E_{act,bw}$) and simulated total actual ET ($E_{act,sim}$) is above 0.6.** This rule ensures that the temporal pattern of simulated ET follows the observed pattern.

3. **The correlation coefficient ($\rho_{SM}$) between observed monthly soil moisture ($SM_{obs}$ [% soil saturation]) and simulated monthly soil moisture ($SM_{sim}$ [m$^3$. m$^{-3}$ soil volume]) is above 0.6.** Simulated soil moisture $SM_{sim}$ for month $t$ is calculated as the average soil moisture within the root zone over all model compartments. This rule guarantees that soil moisture variations are consistent with observations.

4. **Total simulated surface runoff ($Q_{surf}$) is less than 10% of precipitation**, in accordance with a priori information on the carbonate rock sites, which attests that runoff is negligible (see section 3.1).

5. **Soil and vegetation parameter values are consistent with a priori information**, i.e. they fall within constrained (site-specific) ranges. This rule applies to the parameters for which a priori information is available at the FLUXNET sites, namely $h_{veg}, r_{st}, LAI_{min}, LAI_{max} V_r$ and $V_{soi}$ and the constrained ranges are reported in Table 3. This rule ensures that acceptable model outputs are produced using plausible parameter values.

For each site, we derived a sample of size 100,000 for the 15 parameters of V2Karst using latin hypercube sampling and unconstrained (wide) ranges for the model parameters to explore a large range of soil and vegetation types, and we applied the above rules in sequence to either reject or accept the sampled parameter combinations. We sampled more densely the constrained parameter ranges used in rule 5 so that a sufficiently large number of parameterisations remain after applying rule 5. Similarly to (Hartmann et al., 2015), a priori information on parameter ranges (rule 5) is applied last so that we can first assess the constraining of the parameter space based on information on model output only (rules 1 to 4), and then the consistency of this constraining with a priori information (rule 5).

We also note that the thresholds used in rules 1 to 3 are stricter compared to the study by (Hartmann et al., 2015), in which the threshold for the bias rule (1) was set to 75% and for the correlation rules (2 and 3) was set to 0. The reason is that in (Hartmann et al., 2015) behavioural parameter sets had to be consistent with observations at all sites within each climate zone defined in the study, while here we perform the parameter estimation for each site separately and therefore we expect better model performances.

## 4.2 Parameter global sensitivity analysis

We use the Elementary Effect Test (Saltelli et al., 2008), or method of Morris (Morris, 1991). This is a global sensitivity analysis method, and therefore it permits to analyse sensitivity across the entire parameter variability space, it is well suited for identifying uninfluential parameters (Campolongo et al., 2007; Saltelli et al., 2008) and it can be applied to dependent parameters (in V2Karst it is assumed that $V_e \leq V_r \leq V_{S,n}$ as explained in Sect. 2.3.1). The method requires the computation





of the Elementary Effects (EEs) of each parameter in $n$ different baseline points in the parameter space. The EE of the $i$th parameter $x_i$ at given baseline point $\left(x_1^j, x_2^j, ..., x_{i-1}^j, x_i^j, ..., x_M^j\right)$ and for a predefined perturbation $\Delta$ is assessed as follows:

$$EE_i^j = \frac{y\left(x_1^j, x_2^j, ..., x_{i-1}^j, x_i^j + \Delta, ... x_M^j\right) - y\left(x_1^j, x_2^j, ..., x_{i-1}^j, x_i^j, ... x_M^j\right)}{\Delta}, \tag{20}$$

where $M$ is the number of parameters and $y$ is the model output (simulated recharge in our case). The sensitivity indices analysed in the present study are the mean of the absolute values of the EEs (denoted by $\mu_i^*$) introduced in (Campolongo et al., 2007), which is a measure of the total effect of the $i$th parameter, and the standard deviation of the EEs ($\sigma_i$) proposed in (Morris, 1991), which is an aggregate measure of the intensity of the interactions of the $i$th parameter with the other parameters and of the degree of non-linearity in the model response to changes in the $i$th parameter.

The total number of model evaluations required to compute these two sensitivity indices is $n(M + 1)$, where $n$ is the number of baseline points chosen by the user. The baseline points and the perturbation $\Delta$ of Eq. (20) were determined following the radial design proposed by (Campolongo et al., 2011). The baseline points were randomly selected using latin hypercube sampling for the 15 parameters of V2Karst, and dropping the parameter sets that did not meet the condition $V_e \leq V_r \leq V_{S,n}$. In our application, we used $n = 500$ points, which means that we needed 8000 model evaluations for each sensitivity analysis for each of the four FLUXNET sites. We derived confidence intervals on the sensitivity indices via bootstrapping using 1000 bootstrap resamples, and checked the convergence of the results at the chosen sample size, as in (Sarrazin et al., 2016).

## 4.3 Virtual experiments to analyse sensitivity to climate and land cover change

Our last analysis consists of a set virtual experiments to investigate the sensitivity of recharge and actual ET simulated by V2Karst to changes in (1) the precipitation properties (specifically precipitation average amount and temporal distribution) and (2) land cover (specifically from forest to shrub and vice versa).

Virtual experiments using numerical models permit full control on experimental conditions, and thus to unequivocally attribute changes in model outputs to changes in model inputs (see e.g. (Pechlivanidis et al., 2016; Weiler and McDonnell, 2004)). Several studies have used virtual experiments to analyse the impact of precipitation spatial and temporal variability on hydrologic model outputs. In fact, using historical precipitation time series or future projections only allow to explore a limited range of possible realisations, which makes it difficult to disentangle the effects of different precipitation properties on model outputs. Instead, synthetic precipitation time series can be tailored to analyse the impact of specific precipitation characteristics, for instance precipitation spatial distribution (Pechlivanidis et al., 2016; Van Werkhoven et al., 2008b) and precipitation temporal distribution, namely frequency and intensity (Jothityangkoon and Sivapalan, 2009; Porporato et al., 2004), storminess (Jothityangkoon and Sivapalan, 2009) and seasonality (Botter et al., 2009; Jothityangkoon and Sivapalan, 2009; Laio et al., 2002; Yin et al., 2014).





In this study, we create a synthetic precipitation time series where the same precipitation event is periodically repeated. The precipitation time series is characterized by the intensity of precipitation events $I_p$ [mm. d$^{-1}$] and the interval between two wet days $H_p$ [d]. The duration of each precipitation event here is set to one day. The average monthly precipitation

$P_m$ [mm. month$^{-1}$] for an average month with 30 days is therefore equal to:

$$P_m = 30.\frac{I_p}{1 + H_p} \tag{21}$$

To determine the possible range of variation of the three variables, $P_m$, $I_p$ and $H_p$, we analysed their distributions at the four FLUXNET sites and over all European and Mediterranean carbonate rock areas using GLDAS data (Rodell et al., 2004) (distributions are reported in section S6 of our supplementary material). We found that wide but plausible ranges are: $P_m$ varies between 0 and 500 mm.month$^{-1}$, $I_p$ varies between 0 and 200 m.d$^{-1}$ and $H_p$ varies between 0 and 89 d (note that $H_p =$

0 means that it rains every day). We then derived a set of 2266 precipitation time series by deterministically sampling $P_m$, and $H_p$ within those ranges (and consequently deriving a sampled value of $I_p$ from Eq. (21)). We sampled more densely closer to the lower bound of the ranges since lower values of $P_m$ and $H_p$ are more likely to occur.

For each of the precipitation time series so obtained, we ran the V2Karst model until the simulated fluxes reached a steady-state (i.e. periodic oscillations of all state and flux variables) and we analysed the steady-state monthly average of

recharge, transpiration, soil evaporation and evaporation from interception.

The experiments are conducted at two virtual sites that are designed based on the characteristics of the FLUXNET sites. Specifically, we use a virtual 'forest site' that has the characteristics of the German site (i.e. its behavioural parameterisations for the soil, epikarst and vegetation parameters) and a virtual 'shrub site' that has the characteristics of the Spanish site. The forest site also inherits the suboceanic-submountain climate characteristics of the German site (i.e. we force the model by the

average values of air temperature, net radiation, humidity and wind speed measured at that site), while the shrub site inherits the semi-arid climate of the Spanish site. To investigate the impact of a change in land cover at these virtual sites, we swapped the vegetation parameters (indicated in Table 1) between the two virtual sites.

We do not investigate the effects of varying temperature, net radiation, relative humidity and wind speed characteristics as we did for precipitation, because these weather variables are correlated (see e.g. (Ivanov et al., 2007)) and therefore they

cannot be varied independently. Instead, we account for their overall combined effect in a simple way by analysing the changes in sensitivity when these variables are set to winter (low energy for ET) and summer (high energy for ET) conditions. Table 4 reports the values of the parameters and weather variables used at the two virtual sites.



# 5. Results

## 5.1 Parameter estimation

In this section, we present the results of the parameter estimation at FLUXNET sites. We analyse the impact of the application of the soft rules defined in Sect. 4.1 on the reduction in acceptable ('behavioural') parameterisations (Sect. 5.1.1)

and we examine V2Karst outputs (Sect. 5.1.2)

### 5.1.1 Analysis of the constraining of the parameter space

Figure 3 shows that behavioural parameterisations consistent with all rules can be identified at all sites, but their number is very different from one site to another. Specifically, out of the initial 100,000 randomly generated parameter samples, we found 36,838 behavioural parameterisations at the German site, 147 at the Spanish site, 6354 at the French 1 site and 4077 at

the French 2 site. From Fig. 3, we also see that the application of each rule reduces the number of behavioural parameterisations, except for rule 4 (value of total surface runoff < 10% of precipitation), since all model simulations produce less than 7% of surface runoff at all sites. This can be explained by the fact that V2Karst gives priority to recharge production over surface runoff. Therefore, the latter only occurs under extremely wet conditions when all model compartments are saturated.

Figure 4 reports a parallel coordinate plot of the behavioural parameter sets and associated values of the output metrics after sequential application of the soft rules. The application of rules 1 to 4 does not significantly reduce the parameter ranges, but it only allows to discard low values of parameters $V_r$ and $V_{soi}$ at all sites (dark blue lines in Fig. 4). Instead, the application of rule 5 (a priori parameter ranges, red lines in Fig. 4) permits a significant reduction in parameter ranges, not only for the parameters that are directly constrained by this rule ($h_{veg}$, $r_{st}$, $LAI_{min}$, $LAI_{max}$, $V_r$ and $V_{soi}$) but also for the

spatial variability coefficient $a$. Specifically, behavioural values of parameter $a$ are found to be between 0 and 3.2 at the French 1 site, between 0 and 2.8 at the French 2 site. At the Spanish site, we also observe that the behavioural simulations (red lines) cover more densely some portions of the ranges, specifically higher values of parameters $r_{s,soi}$ and $a$, and lower values of $z_0$ and $V_e$. This means that the value for these parameters is more likely to be within these sub-ranges.

### 5.1.2 Analysis of model simulations

In this paragraph, we analyse the repartition of the water fluxes simulated by the V2Karst using the behavioural parameterisations. Figure 5a compares the total simulated recharge and the total actual ET, expressed in percentage of total precipitation at the four FLUXNET sites (mean and 95% confidence interval across the behavioural parameterisations). At the Spanish site, we present the results over two different time periods that have very different precipitation amounts, namely a drier period from 1 January 2006 to 31 December 2008 and a wetter period from 1 January 2009 to 30 December 2011 (see

Fig. 2). Figure 5 shows that, apart from extremely wet periods at Spanish site, in all other cases the fraction of recharge



($Q_{epi}$) is significantly lower than the fraction actual ET ($ET_{act}$). Figure 5b shows the partitioning of ET among its different components (transpiration, soil evaporation and interception). We observe that transpiration ($T_{act}$) is the largest component at all sites, while the relative importance of evaporation from canopy interception ($Ec_{act}$) and soil evaporation ($Es_{act}$) varies across sites. In particular, at the German site, $Ec_{act}$ is on average particularly high compared to the other sites, which may be

partly explained by the fact that summer $LAI$ (parameter $LAI_{max}$) is higher at this densely forested site compared to the other sites, and therefore the summer canopy storage capacity is higher as well.

Finally, Fig. 6 presents the time series of monthly precipitation input ($P$), simulated monthly recharge ($Q_{epi}$), total actual ET ($E_{act}$) and soil moisture in the root zone ($SM_{sim}$) at the four FLUXNET sites. Observation of soil moisture and actual ET are also reported and the blue lines correspond to the Bowen ratio corrected estimate used in rules 1-2 for parameter

estimation (see Sect. 4.1). We see that the soft rules allow to significantly reduce the uncertainty in model outputs at all sites. In fact, the width of the behavioural ensemble, i.e. the ensemble of simulations obtained by application of the rules (black lines), is much narrower than the non-behavioural ensemble (grey lines). Simulated actual ET ($E_{act}$) is also closer to the observations (blue line) in the behavioural ensemble compared to the non-behavioural one. This means that the application of the soft rules and a priori information on parameter ranges allows not only to improve the precision of the simulated states

and fluxes (reduced uncertainty ranges of the simulations), but also the accuracy of simulated actual ET (simulations close to observations). Moreover, the model structure is flexible enough to capture most corrected and uncorrected ET observations, since the non-behavioural model ensemble (grey) includes most corrected and uncorrected ET values.

From Fig. 6, we also observe that the seasonal variations in model predictions are consistent with our understanding of the sites over the entire simulation horizon and not only over the months for which ET and soil moisture observations are used to

estimate the parameters (blue and red areas in the plot). Specifically, at the German site we find a marked seasonality of simulated $E_{act}$ and $SM_{sim}$, with low $E_{act}$ and high $SM_{sim}$ in winter, and high $E_{act}$ and low $SM_{sim}$ in spring and summer. In fact, in winter, the energy available for ET is low and the deciduous vegetation is not able to transpire or intercept large amounts of precipitation, while in spring and summer more energy is available for ET and the vegetation has a higher value of $LAI$, and therefore ET losses can occur and deplete the soil moisture. At the other sites we observe a similar pattern for

$SM_{sim}$, while $E_{act}$ tends to peak in spring and to be lower in summer when the ET fluxes are more water-limited than at the German site.

**5.2 Parameter global sensitivity analysis**

The sensitivity analysis results refer to the sensitivity of total simulated recharge (expressed as a percentage of total precipitation) to the 15 parameters of the V2Karst model. For each parameter, the plots in Fig. 7 report on the horizontal axis

the absolute mean ($\mu^*$) of the Elementary Effects and on the vertical axis their standard deviation ($\sigma$). In all plots, we observe that the bootstrap confidence intervals of the sensitivity indices are narrow and show little overlap, which gives confidence





that the sensitivity results are robust. Similarly to the analysis of the simulated fluxes in Sect. 5.1.2 (Fig. 5), at the Spanish site we present the results for two different time periods with different precipitation amounts.

### 5.2.1 Global sensitivity analysis with constrained parameter ranges

We first examine the left panels in Fig. 7, which show the sensitivity results when ($h_{veg}$, $r_{st}$, $LAI_{min}$, $LAI_{max}$ and $V_r$) and the soil storage capacity $V_{soi}$ are sampled within constrained ranges to inform model calibration in future model applications, since such parameter ranges capture the uncertainty in parameter values left after considering site-specific information. We first note that $\mu^*$ and $\sigma$ take a non-zero value for all parameters at all sites, which means that all parameters are influential and have a non-linear effect on recharge, possibly through interactions with other parameters. The existence of parameter interactions can explain the limited reduction in some parameter ranges during our parameter estimation (Sect. 4.1).

We observe that the spatial variability coefficient $a$ has by far the largest influence, followed by parameters $V_{soi}$ and $V_r$. In fact, their value of $\mu^*$ is significantly higher than the other parameters at all sites. The implication for model calibration in future applications of V2Karst is that efforts should primarily seek to reduce the uncertainty in parameters $a$, $V_{soi}$ and $V_r$. These three parameters also have a significantly large value of $\sigma$, which indicates non-linearities in the model response to variations in these parameters and which is coherent with the nature of Eq. (1-2). Interestingly, parameter $V_{can}$, that controls evaporation from interception, and $r_{s,soi}$, that controls soil evaporation, have an impact on recharge at most sites and at the Spanish site during wet years respectively. This shows that the processes of evaporation from interception and soil evaporation can be important for recharge simulations.

Moreover, we observe that parameters $LAI_{min}$, $z_0$ $k$ and $V_e$ have a very small impact on total recharge at all sites ($\mu^* <$ 3 %). However, Section S7 of our supplementary material reports additional sensitivity analysis results for other model outputs and shows that the most influential parameters that should be the focus of the calibration strategy vary depending on the output of interest. In particular, parameter $V_e$ has a significant impact on the fraction of actual transpiration in total ET, and therefore on the partitioning of ET among its different components.

### 5.2.2 Global sensitivity analysis with unconstrained parameter ranges

The right panels of Fig. 7 show the sensitivity indices when sampling parameters within unconstrained ranges. This analysis allows to test the plausibility of the model structure through the assessment of the model sensitivity across a large spectrum of soil and vegetation conditions.

The most apparent difference with respect to the previous SA results is that vegetation parameters ($h_{veg}$, $r_{st}$, $LAI_{min}$ $LAI_{max}$ and $V_r$) now have a much higher value of the sensitivity indices (both $\mu^*$ and $\sigma$). More specifically, $LAI_{max}$ has a very high sensitivity index at all sites ($\mu^* > 10.5\%$), which can be attributed to the fact that this parameter is used to calculate different model components. Interestingly, the seasonality of leaf area index appears to play an important role in V2Karst since $\mu^*$ for $LAI_{min}$, although always lower than $\mu^*$ for $LAI_{max}$, stands out at all sites.



When comparing parameter sensitivities across sites, we see some significant differences, that we can interpret by considering their climatic differences. In fact, we would expect transpiration to be mainly energy-limited at the German site, given that it has a suboceanic-submountain climate and mainly water-limited at the French sites, which have a Mediterranean climate, and at the Spanish site, which has a semi-arid Mediterranean climate. Specifically, the most influential parameter at the Spanish site is by far parameter $a$ (high $\mu^*$), which has an impact on the water storage in the soil and therefore on the amount of water available to sustain ET between rain events, while at the German site parameter $LAI_{max}$, which is used to calculate PET, has the largest effect on recharge (high $\mu^*$).We also notice that parameters $r_{st}$ and $h_{veg}$, that control PET, are more influential at the German site compared to the other sites.

Finally, we observe that, the parameters that specifically control the volume of transpiration ($r_{st}$ and $V_r$) have a significantly higher value of $\mu^*$ than the parameters that specifically control soil evaporation ($z_0$, $r_{s,soi}$ and $V_e$) and evaporation from interception ($V_{can}$). Moreover, $z_0$, $r_{s,soi}$ and $V_e$ have a very small impact ($\mu^* < 3$ %), while parameter $V_{can}$ can have an important effect at the German site ($\mu^* = 5.7$ %). This suggests that transpiration is overall dominating the ET fluxes at these sites when exploring a wide range of soil and land cover properties and that interception is an important process under the climate of the German site. Additionally, we see that parameter $f_{red}$, that controls transpiration from the third soil layer, has an impact on recharge simulated at the Spanish site.

### 5.3 Virtual experiment

After showing that the V2Karst model behaves reasonably at the four FLUXNET sites, in this section we use virtual experiments to further learn about the sensitivity of simulated recharge to precipitation characteristics and land cover using virtual sites (see Sect. 4.3).

### 5.3.1 Sensitivity of simulated fluxes to precipitation characteristics

Figure 8 shows the monthly average value of simulated recharge $Q_{epi}$, for different values of the precipitation monthly amount $P_m$ (x-axis) and the interval between rainy days $H_p$ (y-axis) at the virtual forest and shrub sites. We do not report $Q_{epi}$ values in the top right of the plots because this region corresponds to very intense precipitation events (higher than 200 mm.d$^{-1}$) that have a very low probability of occurrence (see Sect. 4.3).

From the top left panel of Fig. 8, we see that winter $Q_{epi}$ is mostly sensitive to $P_m$, in fact simulated recharge increases when moving along the horizontal direction from left to right, but shows little variations along the vertical direction (when $H_p$ is varied). This resuls is due to the fact that actual ET is very limited in winter because of the low energy available. We indeed estimated that the maximum value of total ET across the different precipitation inputs is 13 mm.month$^{-1}$ at the forest site and 35 mm.month$^{-1}$ at the shrub site. Therefore, a large part of precipitation becomes recharge rather independently of its temporal distribution.




From the right panel of Fig. 8, we observe a systematic reduction in summer $Q_{epi}$ compared to winter at both virtual sites. Moreover, summer recharge is overall highly sensitive not only to $P_m$ but also to $H_p$, since it increases when moving along the vertical direction from bottom to top, i.e. when the same amount of monthly precipitation falls in less frequent but more intense events. This result can be explained by the fact that in summer potential ET is larger and therefore, if events are less

intense, a larger part of the precipitation is lost via ET, while if instead events are more intense, the canopy and soil stores reach saturation and precipitation generates a saturation excess flow to the epikarst and hence more recharge and less ET. Moreover, in summer, $Q_{epi}$ shows a limited sensitivity to $P_m$ and $H_p$ when these quantities take low values (brown and red dots on the left of the plots), because only few soil compartments reach saturation under drier conditions and therefore little recharge can be generated. We also see that at the shrub site, $Q_{epi}$ is a significant flux ($Q_{epi} > 5mm$) for smaller values of

$P_m$ and $H_p$ compared to the forest site, which may be due to the fact that at the shrub site, the soil water capacity ($V_{soi}$) is much smaller and therefore the soil compartments can reach saturation under drier conditions.

### 5.3.2 Sensitivity of simulated fluxes to land cover change

Figure 9 reports the results of another virtual experiment similar to Fig. 8 but focusing on the impact of land cover change. Specifically, the panels in Fig. 9 show the variation in simulated recharge when land cover is changed from forest to shrub at

the virtual forest site (and vice versa at the virtual shrub site), and more specifically, Fig.9 reports $\Delta Q_{epi} = Q_{epi}^{shrub} - Q_{epi}^{forest}$. We see that in all plots $\Delta Q_{epi}$ is positive, which means that recharge is larger and therefore actual ET is lower under shrub compared to forest land cover for both sites. From the left panels of Fig. 9, we observe that $\Delta Q_{epi}$ is very limited in winter, which is expected since ET fluxes are small in winter as explained in Sect. 5.3.1.

Instead, the right panels of Fig. 9 show that summer $\Delta Q_{epi}$ is much higher compared to winter conditions. The value of

summer $\Delta Q_{epi}$ is largest when the monthly precipitation $P_m$ is high and the interval between wet days $H_p$ is low (green dots at the virtual forest site and dark blue dots at the virtual shrub site), because under these precipitation conditions the amount of moisture available for ET is maximum. Interestingly, for both virtual sites, summer $\Delta Q_{epi}$ is sensitive to both $P_m$ and $H_p$, but its sensitivity is highly variable across the different precipitation inputs, and more specifically an increase in $H_p$ can have a different effect on $\Delta Q_{epi}$ depending on the value of $P_m$ (no variation, increase or decrease in $\Delta Q_{epi}$). In fact, when $P_m$ is

low, $\Delta Q_{epi}$ is always low and does not vary sensibly when $P_m$ and $H_p$ are varied (brown area in the left end of the plot), since recharge is always low under these precipitation conditions as shown in Fig. 8. For intermediate values of $P_m$, $\Delta Q_{epi}$ has a similar pattern at both sites and increases when either $H_p$ or $P_m$ increases. Instead, for high values of $P_m$, we see that for both sites $\Delta Q_{epi}$ decreases when $H_p$ increases and that at the virtual forest site, $\Delta Q_{epi}$ increases when $P_m$ increases.



Importantly, our results also show that the impact of a change in land cover can vary greatly across sites, since at the virtual shrub site summer $\Delta Q_{epi}$ reaches much higher values and is sensitive to $P_m$ and $H_p$ over a larger range of values of $P_m$ and $H_p$ compared to the virtual forest site.

## 6 Discussion

### 6.1 Plausibility of V2Karst simulations

We tested the model by evaluating its ability to reproduce observations at four carbonate rock FLUXNET sites, which is a standard approach to model testing, used for instance to test the previous version of the model Varkarst (Hartmann et al., 2015) and large-scale ET products (Martens et al., 2017; McCabe et al., 2016; Miralles et al., 2011). We demonstrated that V2Karst is able to produce behavioural simulations consistent with observations and a priori information at FLUXNET sites, and additionally the time series of the model outputs are coherent with our understanding of the sites. A different number of behavioural parameterisations was identified at the different sites, because we used the same constrains across sites. The fact that the highest number of behavioural parameterisations was found at the more humid German site and the lowest at the semi-arid Spanish site is coherent with previous findings that higher fit-to-observation can be obtained at wetter locations (Atkinson et al., 2002; Bai et al., 2015).

Interestingly, for the French 1 site, the results of the parameter estimation allow to corroborate the hypothesis that root water uptake is likely to extent below the physical soil layer as communicated by Guillaume Simioni (investigator of the site). In fact, we found here that behavioural values of parameter $V_r$ are higher than 59 mm, while site-specific information indicates that the physical soil layer has a storage capacity of 49 mm (Table B1). This result further attests to the realism of V2Karst structure.

Moreover, the global sensitivity analysis using constrained parameter ranges, that are representative of a wide range of different land cover and soil types, showed a set of sensitivities that are interpretable in light of the different climatic conditions at the four FLUXNET sites. This suggests that the model behaves sensibly and consistently with our understanding of the key vegetation-recharge processes we aim at reproducing.

### 6.2 Sensitivity of simulated groundwater recharge to changes in climate and vegetation characteristics in karst areas

In this study, we investigated the sensitivity of simulated recharge to both climate and land cover change, through a global sensitivity analysis of the model parameters at the FLUXNET sites, and through virtual experiments using a simple synthetic periodic precipitation input.

Firstly, the results of Elementary Effect Test using unconstrained (wide) ranges showed that the vegetation parameters ($h_{veg}$, $r_{st}$, $LAI_{min}$, $LAI_{max}$ and $V_r$ and additionally $V_{can}$ at the German site) have a significant impact on simulated recharge at the FLUXNET sites, which means that simulated recharge is sensitive to changes in land cover properties. More





specifically, the maximum leaf area index ($LAI_{max}$) was highly influential at all sites, and to a lesser extent the parameter controlling the seasonality in $LAI$ ($LAI_{min}$). This is consistent with the findings of previous studies, since (Tesemma et al., 2015) found that assimilating year-to-year monthly $LAI$ in the VIC model can significantly improve runoff simulations compared to using long-term average $LAI$ and (Rosero et al., 2010) determined that $LAI$ has a large influence on simulated

latent heat in the Noah land surface model. Therefore, the future potential increasing trend in global $LAI$ documented by (Zhu et al., 2016) could have a significant impact on the partitioning between green and blue water, including in karst areas.

Our results are also comparable to the sensitivity analysis results obtained for the WaterGap model in (Güntner et al., 2007) and (Werth et al., 2009) with respect to continental water storage and additionally runoff for the latter study. These two studies are the only ones to the author knowledge that performed a parameter global sensitivity analysis including land

cover parameters for the large-scale models of Table A1. Similar to our results, both studies found that highly influential parameters are parameters that control PET (Priestley-Taylor empirical coefficient in WaterGap, which is replaced by parameters $r_{st}$ and $h_{veg}$ in V2Karst), the water storage capacity in the root zone (denoted as $V_r$ in V2Karst) and at a few sites the interception capacity per unit of $LAI$ (denoted as $V_{can}$ in V2Karst). We note that the impact of parameter $LAI$ was not reported and vegetation seasonality was not considered in these two studies.

Secondly, the results of our virtual experiment showed that simulated recharge is sensitive not only to changes in the precipitation amount but also in the precipitation temporal distribution (interval between wet days) and in land cover, and that its sensitivity is highly dependent on the precipitation properties and on the value of the other weather variables that are used to calculate PET (temperature, net radiation, relative humidity and wind speed). These findings indicate that it is critical

to assess the combined impact of changes in all these variables on karst groundwater recharge to gain insights on future water availability in karst areas. A previous study by (Hartmann et al., 2017) also found that recharge simulated with VarKarst is sensitive to the precipitation amount and temporal distribution (specifically intensity of heavy precipitation events), using historical weather time series. Here we complemented the study of (Hartmann et al., 2017) by unequivocally attributing the changes in recharge to changes in precipitation properties using virtual experiments. Our results are also

consistent with past studies for non-karst areas that established dependencies between hydrological fluxes on one side and precipitation properties on the other side, using synthetic precipitation inputs (Jothityangkoon and Sivapalan, 2009; Porporato et al., 2004) and observations of recharge in a semi-arid tropical region (Taylor et al., 2013) and in a seasonally humid tropical region (Owor et al., 2009), and comparing different approaches for the temporal disaggregation of projected monthly precipitation to daily values to force the WaterGap model (Weiß and Alcamo, 2011). However, to the author

knowledge, no previous study had systematically examined the combined impact of changes in specific precipitation characteristics and in land cover on the water balance.

Although, precipitation patterns are more complex than simple periodic variations and the steady state conditions may never be reached in practice, we believe that performing virtual experiments similar to the ones proposed in the present study



is a complementary approach to application of climate projections provided by Global Circulation Models (GCMs) and future land cover change scenarios (e.g. (Holman et al., 2017; Hurtt et al., 2011)), to understand the sensitivity of a model to changes in input characteristics and to determine which aspects of a model input would be worth further investigating.

**6.3 Applying V2Karst over larger domains**

In this section, we first discuss the importance for large-scale applications of the new processes that we introduced in V2Karst and second the strategy to estimate the model parameters over large domains.

The results of our global sensitivity analyses suggest that all newly introduced processes into V2Karst (transpiration, soil evaporation, evaporation from canopy interception, vegetation seasonality and contribution of the water stored below the
root zone to transpiration) are relevant for applications over large domains because all of them can affect simulated recharge, depending on the climatic, soil and land cover conditions. Specifically, the results of our sensitivity analyses across a large range of soil and land cover conditions (wide unconstrained ranges) showed that overall transpiration and vegetation seasonality are important processes under the climate of the four FLUXNET sites, and additionally evaporation from canopy interception and the contribution of water stored below the root zone are also important model components under the climate
of the German site and Spanish site respectively. Moreover, the sensitivity analysis using site-specific constrained ranges revealed that the process of evaporation from canopy interception has an effect on simulated recharge at all forested sites (German site and two French sites) and that the process of soil evaporation has an impact on simulated recharge at the semi-arid site with sparse and short vegetation (Spanish site). The importance of representing canopy interception, in particular for forested land covers, was already mentioned in previous studies (Gerrits, 2010; Savenije, 2004) and the significance of
separating transpiration and soil evaporation was reported in (Maxwell and Condon, 2016; Wang and Dickinson, 2012).

Regarding the estimation of V2Karst parameters, in this study, we showed that the application of the soft rules based on the comparison between observed and simulated variables and on a priori information on parameter ranges (Sect. 4.1) allowed to estimate V2Karst parameters and constrain the model predictions at the four FLUXNET sites. Therefore, to confine V2Karst parameter ranges over a large modelling domain, future studies will investigate the application of an
approach similar to the strategy presented in this study and in (Hartmann et al., 2015) for the VarKarst model, based on soft rules and on the grouping of the model grid cells across the application domain into typical karst-vegetation landscapes. In addition to a priori information on the value of the soil water capacity $V_{soi}$ used in (Hartmann et al., 2015), a priori information on the vegetation parameters will also need to be derived from large-scale databases of vegetation properties (more details on these databases in Sect. S1 of our supplementary material). We can anticipate that the estimation of the
parameters that characterise sub-surface properties ($a$, $V_{soi}$, $V_r$) may be particularly critical, since our sensitivity analyses using site-specific constrained parameter ranges showed that these parameters have the largest impact on simulated recharge.



In addition, unlike above-ground vegetation properties that can be more easily observed (e.g. *LAI*, vegetation height), sub-surface properties are not often well investigated.

One question that we think is still insufficiently addressed in large-scale hydrological modelling is the issue of which parameters should be varied during parameter estimation and uncertainty analysis, and instead which parameters can be

reasonably fixed to a constant value across the modelling domain to simplify the analyses. Other studies have reported on the issue, and in particular a study by (Cuntz et al., 2016) showed that some constant parameters of the Noah-MP land surface model can be highly influential for some model outputs. Likewise, in this study, we found that parameters $V_{can}$ and $r_{s,soi}$, that are typically fixed in the other large-scale models of Table A1, do have an impact on total recharge at least one FLUXNET site. Moreover, as mentioned in Sect. 2.3, $V_{can}$ and $r_{s,soi}$ are understood to vary across land cover type and soil

type respectively, even if no clear ranges of these parameters have been established across land cover and soil types respectively. Therefore, fixing these two parameters could potentially introduce large uncertainties in V2Karst simulations.

The reason for the modellers' decision to fix a given parameter could for example have been based on the finding that the parameter might not have been influential for a particular site at which sensitivity was analysed. However, it might be that the same parameter is influential for other systems with different characteristics since parameter sensitivity can show a high

variability across places as suggested by this study and as further demonstrated in (Güntner et al., 2007; Van Werkhoven et al., 2008a). It is therefore particularly important to assess the sensitivity of model parameters across the modelling domain to test the suitability of fixing model parameters, as done in this study at FLUXNET sites.

## 7. Conclusions

The objectives of the present study were (1) to develop and test an ET component with explicit representation of land cover

processes for the large-scale karst recharge model VarKarst, so that the model can be used for climate and land cover change impact assessment, (2) to evaluate the mechanisms of recharge production in the model as well as the model's sensitivity to temporal precipitation patterns and land cover using virtual experiment.

Many different approaches are used to represent ET in large-scale hydrologic models, and the lack of in-situ ET observations makes it difficult to assess and compare the performance of these different formulations. Moreover, some

models use a large number of parameters that can be only poorly constrained by the few available observations. High model complexity also makes Monte Carlo simulation computationally expensive and hampers uncertainty and sensitivity analysis. The new version of the VarKarst model developed here, V2Karst (V1.0), is the first large-scale model to include explicit representation of both karst and land cover processes. We sought to include parsimoniously processes that are understood to be relevant for climate and land cover impact assessment, namely, (1) a representation of the three ET components

(transpiration, soil evaporation in presence of sparse canopy and evaporation from canopy interception) and (2) a physically-based PET equation (Penman-Monteith). The model also comprises a parsimonious representation of vegetation seasonality.





We showed that V2Karst was able to produce plausible simulations at four carbonate rock FLUXNET sites, since its simulations were consistent with observations of latent heat and soil moisture and a priori information at the sites, and the parameters that dominate the model sensitivity were in accordance with our perception of expected controls on recharge. Additionally, it was also shown that all newly introduced processes in V2Karst can have an impact on simulated recharge

depending on the climate, the soil properties and the land cover.

Virtual experiments, using synthetic periodic precipitation inputs to force the model, allowed to characterise the sensitivity of simulated recharge to the precipitation temporal distribution, the precipitation amount, the seasonal conditions of the other climate variables and the land cover. This had been little examined in previous studies in karst areas. Our results call for a large-scale assessment of the combined impact of future changes in climate (and more specifically the precipitation amount

and temporal distribution) and in land cover on groundwater recharge in karst areas.

Importantly, our study demonstrate that global sensitivity analysis can provide valuable insights for model development, since it can help to determine which processes should be included in models and which parameters can be fixed to constant values with little impact on the simulations. Moreover, global sensitivity analysis, allows to characterise a model sensitivity to changes in climate and land cover. We therefore believe that large-scale hydrology would benefit from a more exhaustive

evaluation of the models' sensitivities over their application domain, since so far sensitivity analyses of large-scale models are very few and many of them explore a limited ranges of possible parameter combinations only.



## Appendix A. Review of ET component in large-scale hydrological models

| Model | $\Delta t$ | Sub-grid variability of soil moisture [a] | Energy balance | ET processes | | | | | Number of parameters for ET estimation [b] | Reference |
|---|---|---|---|---|---|---|---|---|---|---|
| | | | | $T_{act}^{over}$ | $T_{act}^{under}$ | $Es_{act}$ | $Ec_{act}$ | Carbon cycle | | |
| WBM | daily | no | no | **yes** | no | no | no | no | 3 (minimum) | (Federer et al., 2003; Vörösmarty et al., 1989; Vörösmarty et al., 1998) |
| LaD | sub-daily | no | **yes** | **yes** | no | no | no | no | 5 | (Milly and Shmakin, 2002) |
| WaterGap V2.2 | daily | **implicit** | no | **yes** | no | no | **yes** | no | 7 | (Döll et al., 2003; Müller Schmied et al., 2014) |
| LPJ | daily | no | no | **yes** | no | **yes** | **yes** | **yes** | 14 | (Gerten et al., 2004; Sitch et al., 2003) |
| Model of (Kergoat, 1998) | daily | no | no | **yes** | no | **yes** | **yes** | no | 15 | (Kergoat, 1998) |
| PCR-GLOBWB | daily | **implicit** | no | **yes** | no | **yes** | **yes** | no | 15 | (Van Beek and Bierkens, 2008; Van Beek, 2008; Sperna Weiland et al., 2015; Sutanudjaja et al., 2011) |
| Mac-PDM | daily | **implicit** | no | **yes** | **yes** | no | **yes** | no | 16 [c] | (Arnell, 1999; Gosling and Arnell, 2011; Smith, 2016) |
| ISBA | sub-daily | **implicit** | **yes** | **yes** | no | **yes** | **yes** | no | 17 | (Boone et al., 1999; Decharme and Douville, 2006; Noilhan and Planton, 1989) |
| GLEAM V3 | daily | no | no | **yes** | no | **yes** | **tall land cover** | no | 18 [d] | (Martens et al., 2017; Miralles et al., 2010, 2011) |
| VIC V4.2 | daily/ sub-daily | **implicit** | **optional** | **yes** | no | **yes** | **yes** | no | 22 | (Bohn and Vivoni, 2016; Liang et al., 1994) |

**Table A1**. Characteristics of selected large-scale models: simulation time step ($\Delta t$), representation of sub-grid variability of soil moisture, solving of the energy balance, ET processes represented (Overstory transpiration $T_{act}^{over}$, understory transpiration $T_{act}^{under}$, soil evaporation $Es_{act}$, evaporation from canopy interception $Ec_{act}$, and carbon cycle i.e. vegetation dynamic model), and number of parameters for ET estimation. The models were selected based on the following criteria: (1) explicit representation of land cover properties, (2) calculation of ET and soil water balance at a daily or sub-daily time step, and (3) applications in previous studies over a wide range of climate and land cover types. Tables A2 and A3 present the parameterisations used in these models.

[a] None of these models account for karst processes as done by the VarKarst model (Hartmann et al., 2015).
[b] Number of parameters for a given land cover type, excluding parameters used in the representation of vegetation seasonality, carbon cycle (vegetation dynamic), sublimation from snowpack and snowmelt evaporation to make models more comparable.
[c] This number includes the parameters used for the computation of both understory and overstory (grass) transpiration.
[d] Number of parameters assuming tall vegetation (interception is considered for tall vegetation only).

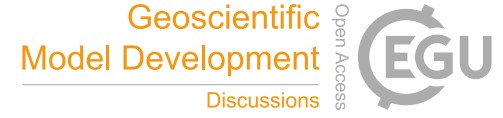



| Model | Potential evapotranspiration (PET) | | | Stress model for actual ET calculation from PET |
| | Formulation [a] | Surface resistance $r_s$ | Number of parameters | |
| --- | --- | --- | --- | --- |
| WBM | T, SW | constant when considered | 0 (minimum) | function of soil moisture which multiplies PET |
| LaD | PM | constant | 2 | function of soil moisture which multiplies PET |
| WaterGap V2.2 | PT | not included | 1 | demand-supply model (Federer, 1982) |
| LPJ | empirical formula based on PT | function of $CO_2$ and photosynthesis | 4 | demand-supply model for transpiration (Federer, 1982) and function of soil moisture which multiplies PET for soil evaporation |
| Model of (Kergoat, 1998) | PM | Jarvis type (Jarvis, 1976; Stewart, 1988) | 10 | function of soil moisture which multiplies $r_s$ |
| PCR-GLOBWB | PM | empirical reference crop scheme (Allen et al., 1998) [b] | 2 | function of soil moisture and soil hydraulic properties which multiplies PET |
| Mac-PDM | PM | constant | 8 [c] | function of soil moisture which multiplies PET |
| ISBA | PM | Jarvis type (Jarvis, 1976; Stewart, 1988) | 8 | function of soil moisture which multiplies $r_s$ for transpiration and PET for soil evaporation |
| GLEAM V3 | PT | not included | 3 [d] | function of soil moisture and vegetation optical depth which multiplies PET |
| VIC V4.2 | PM | Jarvis type (Jarvis, 1976; Stewart, 1988) | 12 | function of soil moisture which multiplies $r_s$ for transpiration and PET for soil evaporation |

**Table A2**. Representation of potential evapotranspiration (PET) and stress model for actual evapotranspiration (ET) calculation from PET in the large-scale models of Table A1.

[a] T: Thornthwaite (Thornthwaite, 1948); PT: Priestley-Taylor (Priestley and Taylor, 1972); PM: Penman-Monteith (Monteith, 1965); SW: Shuttleworth-Wallace (Shuttleworth and Wallace, 1985).

5   [b] This approach consists of calculating a value of PET for a reference grass surface with known properties and to adjust this potential rate using land cover specific empirical crop factors. This formulation avoids the specification of the stomatal resistance whose value is largely uncertain (see Sect. S1 of our supplementary material). Tabulated values of the crop factors for agricultural crops are provided in (Allen et al., 1998). However, the origin of the crop factor formulation for non-agricultural crops is not clear.

10   [c] This number includes the parameters used for the computation of PET for both understory and overstory (grass).

[d] Number of parameters assuming tall vegetation (interception is considered for tall vegetation only).



| Model | Sparse vegetation formulation [a] | Soil layers [b] | Evaporation from canopy interception ($Ec_{act}$) | | Seasonality of vegetation |
|---|---|---|---|---|---|
| | | | model | Number of parameters | |
| WBM | not included | 1 layer | not included | 0 | not included |
| LaD | not included | 1 layer | not included | 0 | not included |
| WaterGap V2.2 | not included | 1 layer | overflow store | 3 | empirical *LAI* growth model |
| LPJ | uncoupled (vegetated and bare soil tiles) | 3 layers ($Es_{act}$ from shallow layer and $T_{act}$ from all layers depending on their relative root fractions) | empirical: fraction of precipitation (Kergoat, 1998) | 2 | vegetation dynamic model |
| Model of (Kergoat, 1998) | coupled moisture uptake | 1 layer | empirical: fraction of precipitation (Kergoat, 1998) | 2 | *LAI* set to zero during leaf-off season |
| PCR-GLOBWB | coupled moisture uptake | 2 layers ($Es_{act}$ from shallow layer and $T_{act}$ from all layers depending on their relative root fractions) | overflow store | 2 | monthly values of crop factors and *LAI* |
| Mac-PDM | uncoupled (overstory and understory tiles) | 1 layer for each tile | Calder (Calder, 1990) | 3 [c] | not included |
| ISBA | coupled moisture uptake | 3 layers ($Es_{act}$ from two shallower layers and $T_{act}$ from middle layer and capillary rise from deeper layer) | overflow store | 3 | monthly values of vegetation parameters |
| GLEAM V3 | uncoupled (vegetated and bare soil tiles) | 3 layers ($Es_{act}$ from shallower layer and $T_{act}$ in wettest layer) | Gash (Gash, 1979; Valente et al., 1997) | 7 [d] | assimilation of vegetation optical depth |
| VIC V4.2 | coupled moisture uptake | 2 layers ($Es_{act}$ from shallower layer and $T_{act}$ from all layers depending on their relative root fractions) | overflow store | 3 | monthly values of *LAI* and assimilation of daily NDVI |

**Table A3**. Representation of sparse vegetation, soil layers, evaporation from canopy interception and seasonality of vegetation in the large-scale models of Table A1.

[a] Uncoupled approaches consist of assessing separately the water balance for the vegetated and bare soil fractions (overstory and understory fractions for Mac-PDM). Therefore, this approach is based on the simplifying assumption that the vegetation roots do not extent beyond the surface area covered by the vegetation canopy. Instead, coupled approaches evaluate the overall water balance over both fraction, thus allowing for interactions for soil moisture uptake between vegetated and bare soil fractions. All models neglect aerodynamic interactions between vegetation and bare soil. This can be accounted for using for instance the Shuttleworth-Wallace PET equation (Shuttleworth and Wallace, 1985), which requires the specification of further resistance parameters compared to the Penman-Monteith equation. The Shuttleworth-Wallace equation was used anecdotally in the WBM model for a few applications.

[b] $Es_{act}$: actual soil evaporation; $T_{act}$: actual vegetation transpiration.

[c] This number includes the parameters used for the computation of PET for both understory and overstory (grass).

[d] Number of parameters assuming tall vegetation (interception is considered for tall vegetation only).





## Appendix B. Additional information on the four carbonate rock FLUXNET sites

| Site name | | Hainich (German site) | Llano de los Juanes (Spanish site) | Puéchabon (French 1 site) | Font-Blanche (French 2 site) |
|---|---|---|---|---|---|
| General information | Coordinates | 51°04′45″N, 10°27′07″E | 36°55′56′′N, 2°44′55′′W | 43°14′27″N, 5°40′45″E | 43°44′29′′N, 3°35′45′′E |
| | Elevation | 430 m a.s.l. | 1600 m a.s.l | 420 m a.s.l | 270 m a.s.l |
| Vegetation | Type | Deciduous broadleaf trees | Shrubs, herbs, bare soil, rock outcrops | Evergreen trees (30% broadleaf and 70% needleleaf) | Evergreen broadleaf trees |
| | Maximum *LAI* | 5 m$^2$.m$^{-2}$ | 2.71 m$^2$.m$^{-2}$ | 2.2 m$^2$.m$^{-2}$ | 2.9 ± 0.4 m$^2$.m$^{-2}$ |
| | Height | Around 33 m | 0.5 m (average) - 1.2 m (maximum) | 6 m (broadleaf) and 12 m (needleleaf) | 5.5 m |
| | Seasonality | Leaves from May to Octobe | 1.31 m$^2$.m$^{-2}$ (annual minimum) | Not available | Not available |
| | Rooting depth | Not available | Roots probably access water below the soil | Roots probably access water below the soil | 4.5 m (150 mm available water capacity) |
| Soil | Texture | Silty clay | Silt loam and clay loam | Sandy clay loam | Silty clay loam and clay loam |
| | Depth | 0.5 - 0.7 m | 0.1 – 0.3 m (occasionally up to 1.5 m) | 0.6 m (maximum) | No clear limit between soil and epikarst |
| | Available water capacity [a] | 0.13 m$^3$.m$^{-3}$ | 0.25 m$^3$.m$^{-3}$ | 49 mm | No clear limit between soil and epikarst |
| | Other properties | Permeable loess layer of 10 -50 cm between soil and bedrock | Rocky soil | Rocky soil | Rocky soil |
| Bedrock | | Fissured and fractured limestone | Karstified dolomite and dolines | Karstified limestone | Karstified limestone |
| Hydrology | Surface runoff | Low | Low | Low | Inexistent |
| | Recharge | Large part of the water balance | Diffuse and concentrated, high temporal variability | Not available | Not available |
| Measurements | Height for humidity and temperature | 43.5 m | 1.5m | 16 m | 12.2 m |
| | Height for wind speed | 43.5 m | 2.5 m | 16 m | 12.2 m |
| | Depth for soil moisture | 0.05, 0.15, 0.3 m | 0.15 m | Not measured | Not measured |
| References | | (Knohl et al., 2003; Mund et al., 2010; Pinty et al., 2011), personal communication from Martina Mund and Manfred Fink | (Alcalá et al., 2011; Cantón et al., 2010; Contreras et al., 2008; Li et al., 2007, 2011; Pérez-Priego et al., 2013; Serrano-Ortiz et al., 2007) | (Ecofor, n.d.; Gea-Izquierdo et al., 2015; Simioni et al., 2013), personal communication from Guillaume Simioni, | (Rambal, 1992, 2011; Rambal et al., 2003; Reichstein et al., 2002) |

**Table B1**. Description of the four carbonate rock FLUXNET sites. [a] between wilting point and field capacity.





**Supplementary material**

**Code availability**

The code of the V2Karst model is open source and freely available under the terms of the GNU General Public License version 3.0. The model code is written in matlab and is provided through a Github repository:
https://github.com/fannysarrazin/V2Karst_model

**Author contribution**

F.S., A.H. and T.W. contributed to the design of the model equations of V2Karst. F.S. developed the code of the V2Karst model building on the code of the previous version of the model (VarKarst), which was developed by A.H.. F.S. performed the numerical experiments. F.S., A.H., F.P. and T.W. contributed to the methodology and the analysis of the results. The
manuscript was prepared by F.S. and edited by all the authors.

**Competing interest**

The authors declare that they have no conflict of interest.

**Acknowledgements**

F.S. was supported by a PhD Scholarship Programme from the University of Bristol Development and Alumni Relations
Office. Support to A.H. was provided by the Emmy Noether-Programme of the German Research Foundation (DFG; grant number HA 8113/1-1; project 'Global Assessment of Water Stress in Karst Regions in a Changing World'). F.P. was partially supported by a UK Engineering and Physical Sciences Research Council fellowship (EPSRC; grant number EP/R007330/1). We are grateful to Rafael Rosolem for advice on FLUXNET data processing and ET modelling. We thank Timothy Foster for advice on ET modelling. We thank Yoshihide Wada for providing information on ET representation in
the PCR-GLOBWB model. We thank the investigators of the four FLUXNET sites (DE-Hai, Es-Lju, FR-FBn and FR-Pue) for allowing us to use their data, and in particular Martina Mund and Manfred Fink (De-Hai) Penelope Serrano Ortiz, Francisco Domingo Poveda and Andrew Kowalski (ES-Lju), Richard Joffre and Serge Rambal (FR-Pue), and Guillaume Simioni (FR-FBn) for sharing information on the characteristics of the FLUXNET sites. We appreciate the support of Eleonora Canfora and Dario Papale from the FLUXNET European Fluxes Database Cluster.



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





| Parameter | Description | unit | Lower limit | Upper limit | Category |
|---|---|---|---|---|---|
| $h_{veg}$ | Vegetation height | [m] | 0.2 | Site specific | vegetation |
| $r_{st}$ | Stomatal resistance | [s.m$^{-1}$] | 20 | 600 | vegetation |
| $LAI_{min}$ | Reduction in leaf area index during the dormant season | [%] | 5 | 100 | vegetation |
| $LAI_{max}$ | Annual maximum leaf area index | [m$^2$.m$^{-2}$] | 0.5 | 8 | vegetation |
| $V_r$ | Maximum storage capacity of the root zone | [mm] | 20 | 500 | vegetation |
| $V_{can}$ | Canopy storage capacity per unit of $LAI$ | [mm LAI] | 0.1 | 0.5 | vegetation |
| $k$ | Beer-Lambert's law extinction coefficient | [-] | 0.4 | 0.7 | vegetation |
| $f_{red}$ | Reduction factor for transpiration below the root zone | [-] | 0 | 0.15 | soil |
| $z_0$ | Soil roughness length | [m] | 0.0003 | 0.013 | soil |
| $r_{s,soi}$ | Soil surface resistance | [s.m$^{-1}$] | 0 | 100 | soil |
| $V_e$ | Maximum storage capacity of the first soil layer | [mm] | 5 | 45 | soil |
| $a$ | Spatial variability coefficient | [-] | 0 | 6 | soil and epikarst |
| $V_{soil}$ | Mean soil storage capacity | [mm] | 20 | 800 | soil |
| $V_{epi}$ | Mean epikarst storage capacity | [mm] | 200 | 700 | epikarst |
| $K_{epi}$ | Mean epikarst outflow coefficient | [d] | 0 | 50 | epikarst |

**Table 1**. Description of V2Karst parameters, unconstrained ranges used in the application at the four FLUXNET sites to capture the variability across soil, epikarst and vegetation types, category of the parameters (which indicated whether the parameters depend on soil, epikarst or vegetation properties). Parameters $a$, $V_{soil}$, $V_{epi}$ and $K_{epi}$ were already present in the previous version of the model (VarKarst). More information on how the ranges were determined is provided in Sect. S3 of our supplementary material.



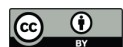

| Site | Simulation period (including a one-year warm-up period) | | Number of months with latent heat measurement for calibration | Number of months with soil moisture measurement for calibration |
|---|---|---|---|---|
| | Start | End | | |
| German site | 1 Jan. 2000 | 17 Dec. 2009 | 62 | 74 |
| Spanish site | 1 Jan. 2005 | 30 Dec. 2011 | 12 | 12 |
| French 1 site | 2 Jan. 2009 | 30 Dec. 2011 | 13 | Not measured |
| French 2 site | 18 Apr. 2002 | 29 Jun. 2009 | 37 | Not measured |

**Table 2**. Simulation period at the four FLUXNET sites, and number of months where latent heat measurements and soil moisture measurements are available to calibrate the model. Soil moisture measurements are not provided at the two French sites.





| Parameter | Unit | German site (deciduous forest) | | Spanish site (shrubland) | | French 1 site (evergreen forest) | | French 2 site (evergreen forest) | |
|---|---|---|---|---|---|---|---|---|---|
| | | Lower limit | Upper limit | Lower limit | Upper limit | Lower limit | Upper limit | Lower limit | Upper limit |
| $h_{veg}$ | [m] | 23.1 | 42.9 | 0.35 | 0.85 | 7.1 | 13.3 | 3.9 | 7.2 |
| $r_{st}$ | [s.m$^{-1}$] | 275 | 400 | 195 | 350 | 320 | 455 | 320 | 455 |
| $LAI_{min}$ | [%] | 5 | 20 | 34 | 63 | 80 | 100 | 80 | 100 |
| $LAI_{max}$ | [m$^2$.m$^{-2}$] | 3.5 | 6.5 | 1.9 | 3.5 | 1.5 | 2.9 | 2.0 | 3.8 |
| $V_r$ | [mm] | 60 | 300 | 30 | 200 | 30 | 200 | 30 | 200 |
| $V_{soi}$ | [mm] | 60 | 400 | 30 | 300 | 30 | 300 | 30 | 300 |

**Table 3**. Site-specific constrained parameter ranges at the four FLUXNET sites for the vegetation parameters ($h_{veg}$, $r_{st}$, $LAI_{min}$, $LAI_{max}$, $V_r$) and for the soil storage capacity ($V_{soi}$). More information on how the ranges were determined is provided in Sect. S3 of our supplementary material. Parameters are defined in Table 1.





| V2Karst input | | Unit | Virtual forest site | Virtual shrub site |
|---|---|---|---|---|
| Vegetation parameter | $h_{veg}$ | [m] | 32.1 | 0.4 |
| | $r_{st}$ | [s.m$^{-1}$] | 390 | 291 |
| | $LAI_{min}$ | [%] | 16 | 38 |
| | $LAI_{max}$ | [m$^2$.m$^{-2}$] | 5.0 | 2.0 |
| | $V_r$ | [mm] | 289 | 151 |
| | $V_{can}$ | [mm LAI] | 0.29 | 0.35 |
| | $k$ | [-] | 0.53 | 0.45 |
| Soil and epikarst parameter | $f_{red}$ | [-] | 0.010 | 0.080 |
| | $z_0$ | [m] | 0.0110 | 0.0045 |
| | $r_{s,soi}$ | [s.m$^{-1}$] | 56 | 61 |
| | $V_e$ | [mm] | 11 | 8 |
| | $a$ | [-] | 1.8 | 1.9 |
| | $V_{soil}$ | [mm] | 373 | 174 |
| | $V_{epi}$ | [mm] | 396 | 519 |
| | $K_{epi}$ | [d] | 33 | 15 |
| Weather input (winter) | $R_n$ | [MJ.m$^{-2}$.d$^{-1}$] | -0.0 | 2.2 |
| | $T$ | [°C] | 0.1 | 4.9 |
| | $RH$ | [%] | 89 | 61 |
| | $WS$ [a] | [m.s$^{-1}$] | 3.5 | 4.0 |
| Weather input (summer) | $R_n$ | [MJ.m$^{-2}$.d$^{-1}$] | 10.5 | 12.1 |
| | $T$ | [°C] | 16.6 | 20.4 |
| | $RH$ | [%] | 72 | 43 |
| | $WS$ [a] | [m.s$^{-1}$] | 2.6 | 3.4 |

**Table 4**. Values of V2Karst parameters and weather variables used in the virtual experiment. Values for the virtual forest site and the virtual shrub site are based on the characteristics of the German FLUXNET site and Spanish FLUXNET site respectively. Values of the model parameters (parameters are defined in Table 1) correspond to behavioural parameterisations obtained when calibrating the model and values of the weather variables ($R_n$ net radiation, $T$ temperature, $RH$ relative humidity, $WS$ wind speed) correspond to the average values calculated at FLUXNET sites.

[a] At the virtual shrub site, $WS$ was recalculated at a height of 43.5 m because the original measurement provided at a height of 2.5 m at the Spanish site was too low to simulate a change of land cover to tall vegetation (forest). More details on this are reported in Section S4 of our supplementary material.

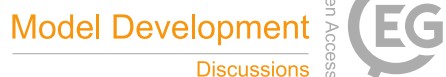

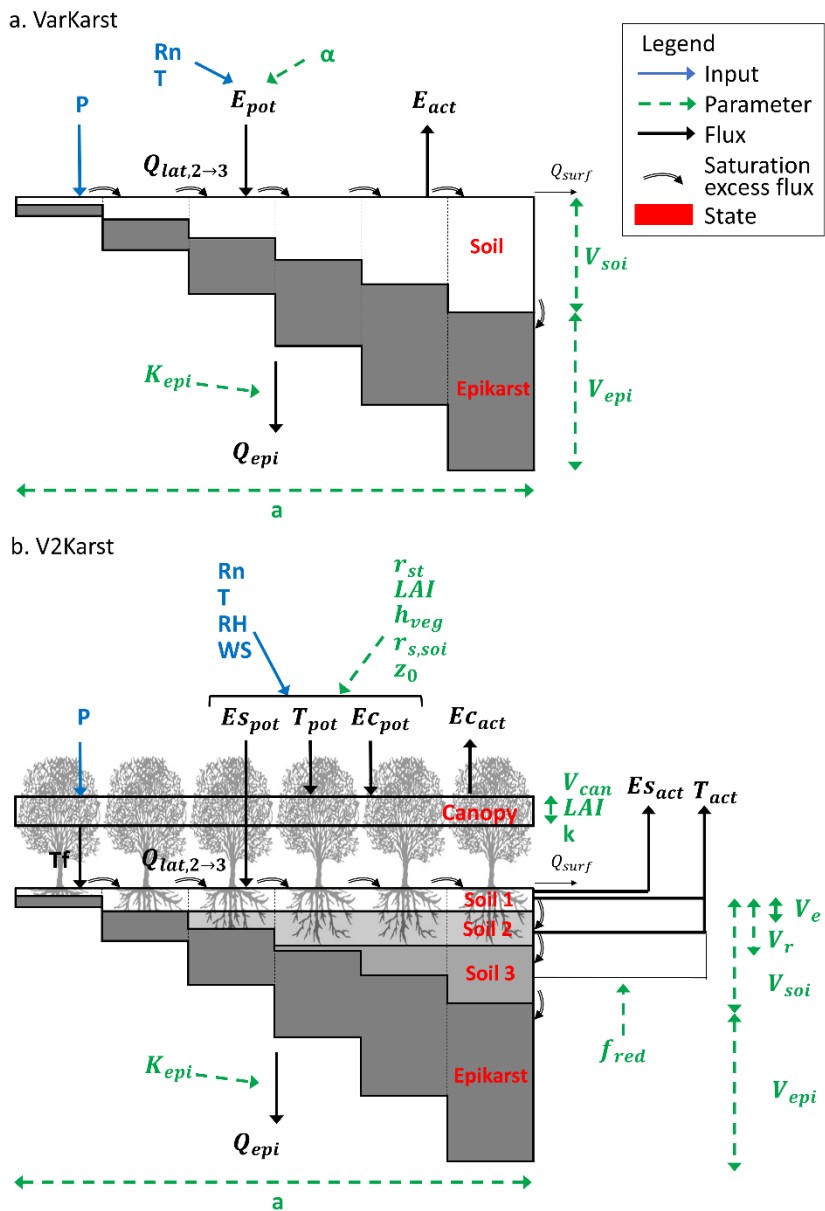

**Figure 1**. Schematic representation of (a) the VarKarst model (Hartmann et al., 2015) and (b) the new version of the model V2Karst using six vertical compartments. Model parameters are in green (see their definition in Table 1)**,** inputs are in blue ($P$ precipitation, $R_n$ net radiation, $T$ temperature, $RH$ relative humidity, $WS$ wind speed), model fluxes are in black ($E_{pot}$ potential total evapotranspiration, $T_{pot}$ potential transpiration, $Ec_{pot}$ potential evaporation from canopy interception, $Es_{pot}$ potential soil evaporation, $E_{act}$ total actual ET, $T_{act}$ actual transpiration, $Ec_{act}$ actual evaporation from canopy interception, $Es_{act}$ actual bare soil evaporation, $Tf$ throughfall, $Q_{lat,2 \to 3}$ lateral flow from the second to the third compartment, $Q_{surf}$ surface runoff and $Q_{epi}$ recharge) and state variables are in red.





**German site**
- Fractures and fissured bedrock
- Tall and dense deciduous broadleaf forest
- Suboceanic-submountain climate
$\bar{P} = 762 mm. year^{-1}$
$\bar{T} = 8.4\ °C$
Elevation = 430 $m$

**French 2 site**
- Karstified bedrock
- Short evergreen broadleaf forest
- Mediterranean climate
$\bar{P} = 892\ mm. year^{-1}$
$\bar{T} = 13.4\ °C$
Elevation = 270 $m$

**Spanish site**
- Karstified bedrock
- Open shrubland
- Semi-arid mountain Mediterranean climate
$\bar{P} = 486\ mm. year^{-1}$ (dry years 2005-2008)
$\bar{P} = 912 mm. year^{-1}$ (wet years 2009-2011)
$\bar{T} = 11.7\ °C$
Elevation = 1600 $m$

**French 1 site**
- Karstified bedrock
- Medium height mixed evergreen forest
- Mediterranean climate
$\bar{P} = 571\ mm. year^{-1}$
$\bar{T} = 13.6\ °C$
Elevation = 420 $m$

**Figure 2**. Four carbonate rock FLUXNET sites selected for the analyses. Mean annual precipitation $\bar{P}$ and mean annual temperature $\bar{T}$ were estimated over the period 1 January 2001-17 December 2009 for the German site, 1 January 2006-31 December 2008 for the Spanish site (dry years), 1 January 2009-30 December 2011 for the Spanish site (wet years), 1 January 2010-30 December 2011 for the French 1 site and 1 April 2003-31 March 2009 for the French 2 site.

Sources of the photos: (Pinty et al., 2011) for the German site, (Alcalá et al., 2011) for the Spanish site, http://www.gip-ecofor.org/f-ore-t/fontBlanche.php for the French 1 site, http://puechabon.cefe.cnrs.fr/ for the French 2 site. Source of the carbonate rock and country map: (Williams and Ford, 2006) (country map obtained from Terraspace, Russian space agency).





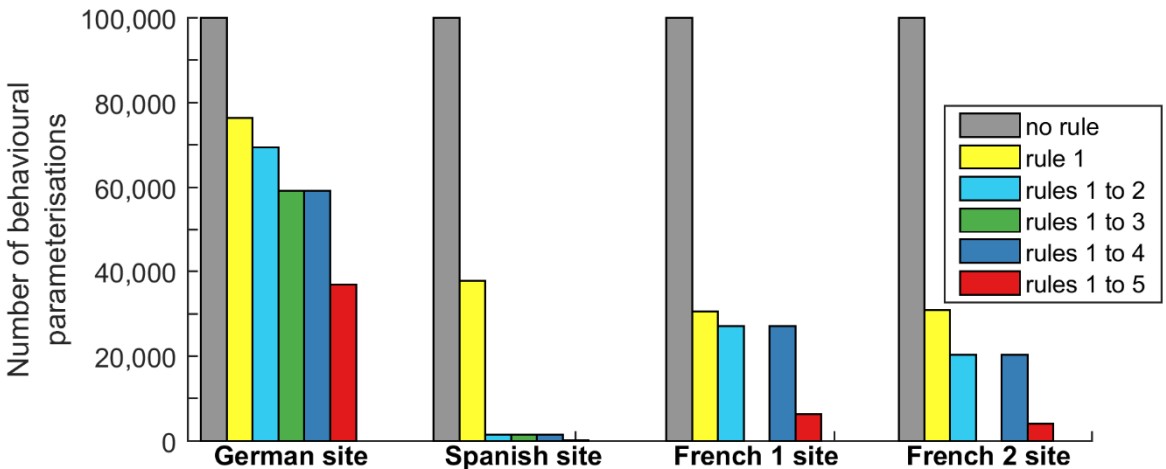

**Figure 3**. Reduction in the number of behavioural parameterisations of the V2Karst model at the four FLUXNET sites, when applying sequentially the five soft rules defined in Sect. 4.1 (no rule: initial sample; rule 1: ET bias; rule 2: ET correlation; rule 3: soil moisture correlation; rule 4: runoff; rule 5: a priori information). Rule 3 could not be applied to the French sites where soil moisture observations are not available.





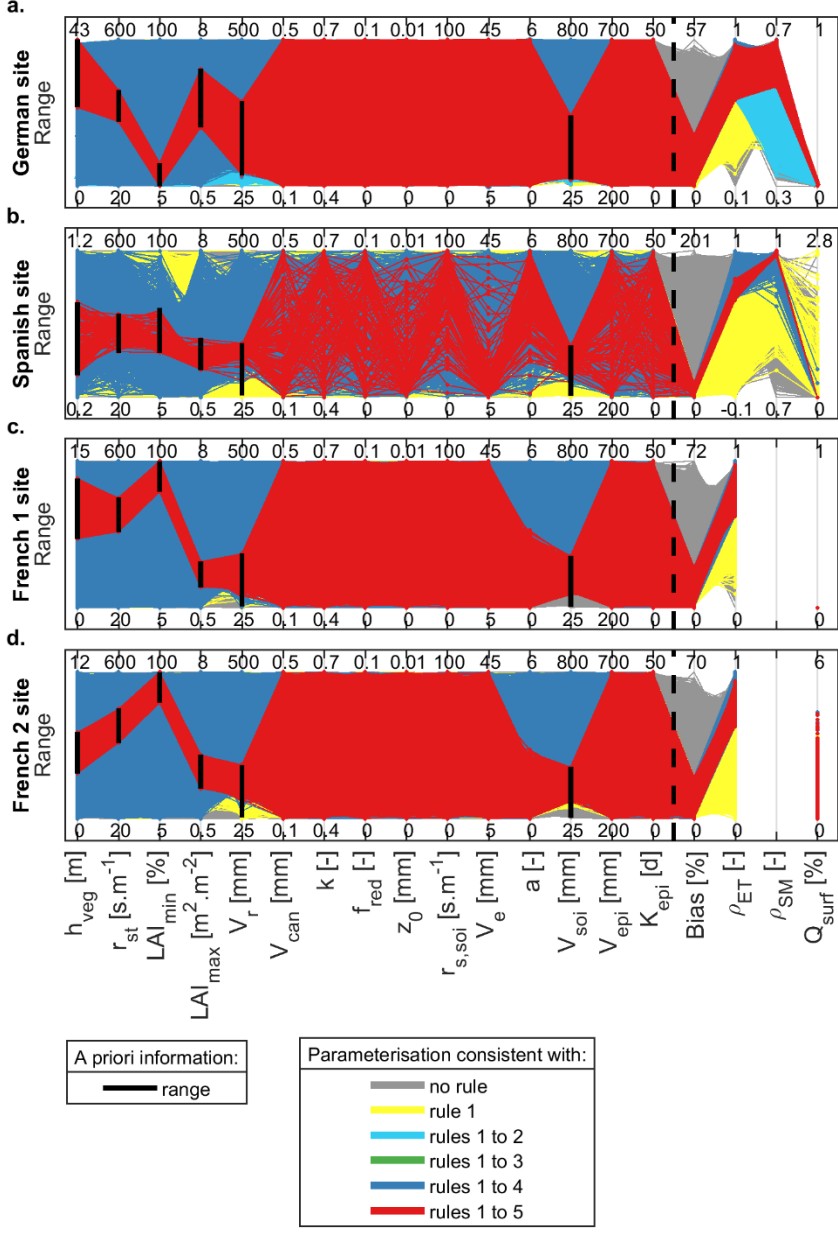

**Figure 4**. Parallel coordinate plots representing V2Karst behavioural parameterisations, and their corresponding simulated output values, identified when sequentially applying the five soft rules defined in Sect. 4.1 at (a) the German site, (b) the Spanish site, (c) the French 1 site and (d) the French 2 site. Parameters are defined in Table 1. *BIAS* absolute mean error between observed and simulated total actual ET (rule 1), $\rho_{ET}$ correlation coefficient between observed and simulated total actual ET (rule 2), $\rho_{SM}$ correlation coefficient between observed and simulated soil moisture (rule 3), $Q_{surf}$ surface runoff (rule 4). Rule 5 corresponds to application of a priori information on parameter ranges (black vertical bars).





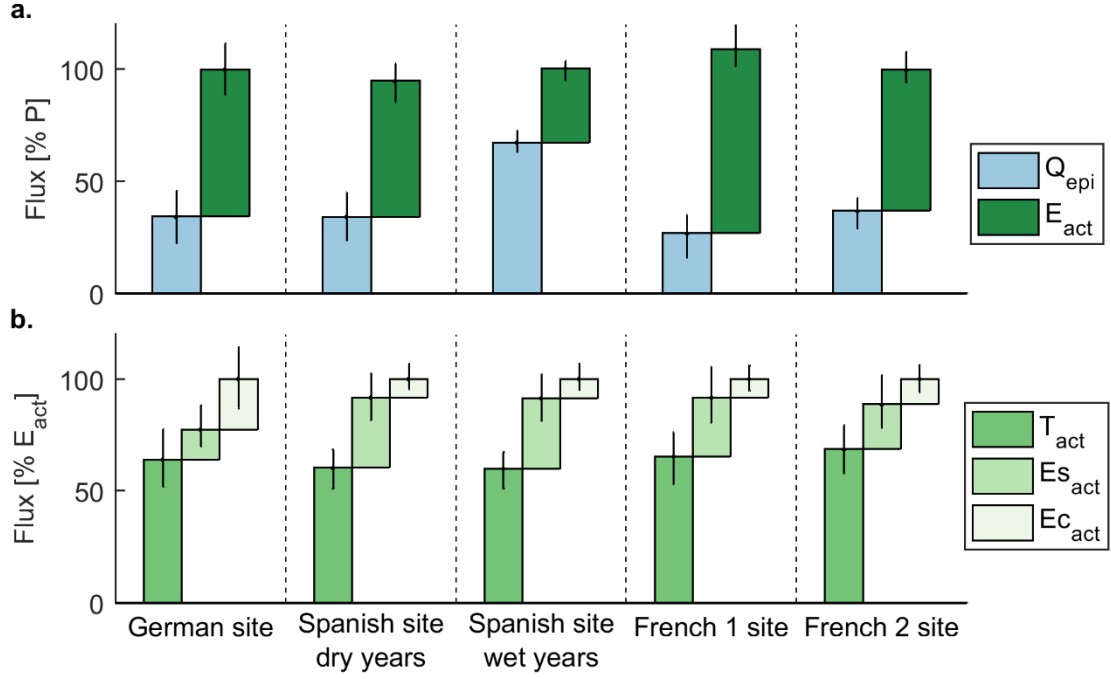

**Figure 5**. (a) Simulated recharge ($Q_{epi}$) and actual ET ($E_{act}$) expressed as a percentage of total precipitation and (b) simulated actual transpiration ($T_{act}$), actual soil evaporation ($Es_{act}$) and actual evaporation from interception ($Ec_{act}$) expressed as a percentage $E_{act}$. The figure reports the ensemble mean and 95% confidence intervals calculated over the behavioural simulation ensemble of the V2Karst model at the four FLUXNET sites. Simulated fluxes were evaluated over the period 1 January 2001-17 December 2009 for the German site, 1 January 2006-31 December 2008 for the Spanish site (dry years), 1 January 2009-30 December 2011 for the Spanish site (wet years), 1 January 2010-30 December 2011 for the French 1 site and 1 April 2003-31 March 2009 for the French 2 site.









**Figure 6**. Monthly time series of precipitation input ($P$), simulated recharge ($Q_{epi}$), simulated actual ET ($E_{act}$, which is the sum of evaporation from canopy interception, transpiration and soil evaporation), simulated soil moisture within the root zone ($SM_{sim}$), and monthly observations of actual ET and soil moisture at (a) the German site, (b) the Spanish site, (c) the French 1 site and (d) the French 2 site. Blu and red shaded areas correspond to the periods in which observation of ET and soil moisture respectively were selected to apply the soft rules of Sect. 4.1 (further details on data processing in Sect. 3.2).







**Figure 7**. Sensitivity indices of the V2Karst parameters ($\mu^*$ is the mean of the absolute Elementary Effects and $\sigma$ is the standard deviation of the Elementary Effects) for total simulated recharge (expressed as a percentage of total precipitation) at the four FLUXNET sites, when constrained (site-specific) parameter ranges are used (ranges of Table 3) and when unconstrained ranges are used (ranges of Table 1). Sensitivity indices were computed over the period 1 January 2001-17 December 2009 for the German site, 1 January 2006-31 December 2008 for the Spanish site (dry years), 1 January 2009-30 December 2011 for the Spanish site (wet years), 1 January 2010-30 December 2011 for the French 1 site and 1 April 2003-31 March 2009 for the French 2 site.





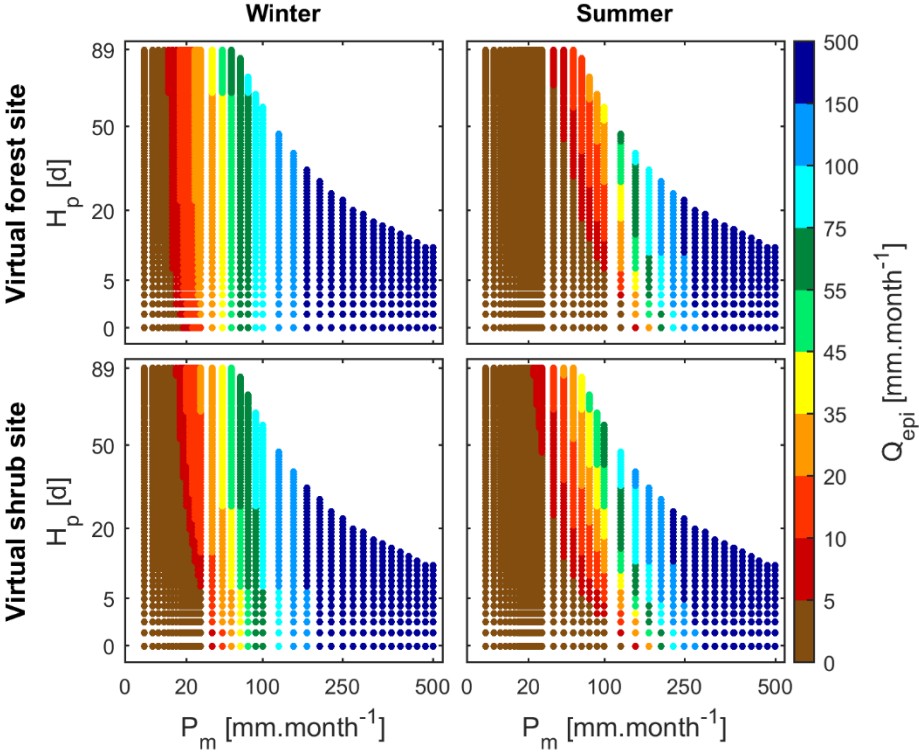

**Figure 8**. Average monthly recharge ($Q_{epi}$) simulated with V2Karst for different values of the average monthly precipitation amount $P_m$ [mm. month$^{-1}$] and the interval between wet days $H_p$ [d] of the synthetic periodic precipitation input used to force the model at the virtual forest and shurb sites and under winter and summer conditions.

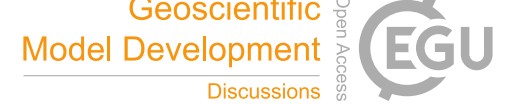

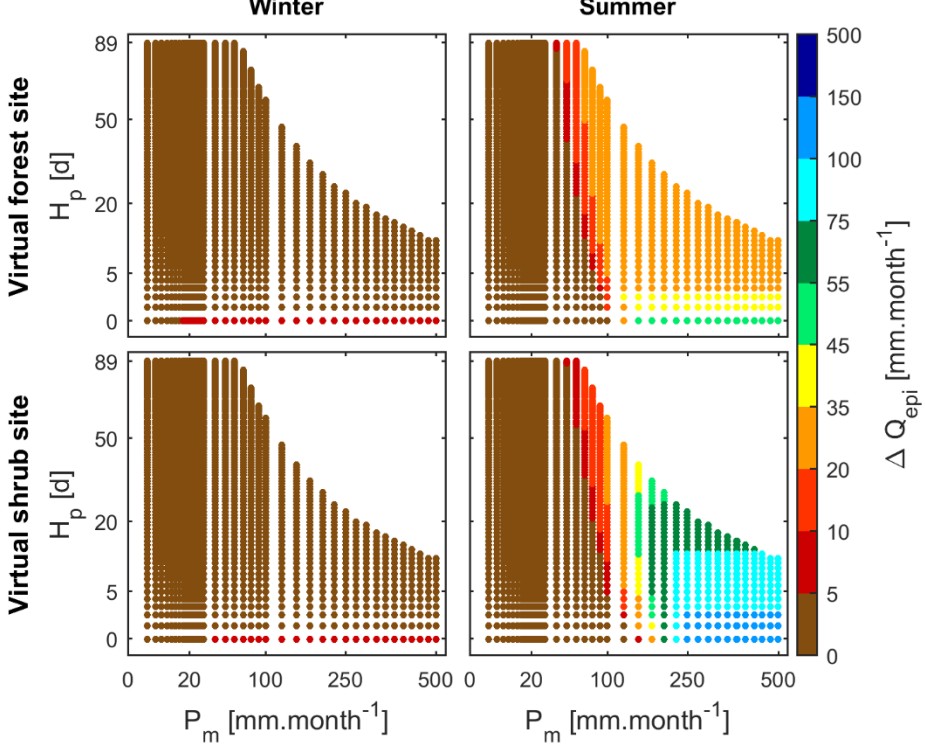

**Figure 9.** Change in monthly recharge ($\Delta Q_{epi} = Q_{epi}^{shrub} - Q_{epi}^{forest}$) simulated with V2Karst when the land cover is set to shrub compared to forest for different values of the average monthly precipitation amount $P_m$ [mm.month$^{-1}$] and the interval between wet days ($H_p$ [$d$]) of the synthetic periodic precipitation input used to force the model at the virtual forest and shurb sites and under winter and summer conditions.