# Peer review of "V2Karst V1.1: A parsimonious large-scale integrated vegetationrecharge model to simulate the impact of climate and land cover change in karst regions"

_Geoscientific Model Development, 2017_

## Referee Comment (RC1) · Anonymous Referee #1 · 30 Apr 2018

The manuscript presents a modified version of the large-scale karst recharge model VarKarst. The here presented model (V2Karst V1.0) replaces the simplified evapotranspiration (ET) component (empirical Priestley-Taylor equation) by the physical based Penman-Monteith equation (for potential evapotranspiration). The authors also include a separate calculation of the different evaporation processes in order to use the model for climate and land cover change impact studies. The model extension increases the number of parameters. The general functioning as well as the influence of the new parameters are tested by applying the new model to four study sides, different in climate and vegetation. The manuscript is a novel extension of previous work published by the research group. The conceptual description and the numerical adaptation of the processes are sound. The results of the model application on the four test sides prove the general functioning of the new model. However, the manuscript also has weak points, which are mainly related to the presentation of the method and the results. The manuscript can easily be shortened by 10-20% without losing important information. The presentation of the results needs to be improved, especially since it is difficult to distinguish between observed values and modeled results. My detailed comments are listed below.

Main Comments

The purpose of V2Karst V1.0 is to predict recharge in karst regions. The authors mention that "a large part of the groundwater recharge occurs as concentrated and fast flow in large apertures and the other part as diffuse and slow flow in the matrix (Hartmann and Baker, 2017)." Especially concentrated recharge, e.g. fast infiltration into sinkholes, can be considered as a short-term process and is entirely uncoupled from soil and/or vegetation properties (overland flow -> percolation). I assume that your model, calculating the water balance, underestimates the recharge in karst regions dominated by concentrated recharge. Do you think your model is able to equally represent both recharge processes?

I am aware of the fact that the manuscript is focused on the implementation and the testing of the new evapotranspiration component. Since soil layers in karst regions can be thin or even totally absence the authors should consider this fact in the interpretation of the results. The manuscript lacks a description/characterization of the four karst regions (e.g. by describing dominant karst features or by the interpretation of spring hydrographs). In general, a differentiation between different karst systems and therefore the wide variety of hydraulic properties dominating the recharge pattern (see above) is missing here.

As already mentioned, the current manuscript is too long and needs to be shortened: - (Almost) every section starts with a short introduction on the section. Most of them are redundant. - The authors use wordy descriptions instead of clear words for describing their work. Is a "virtual experiment with synthetic data to assess the sensitivity" (Page 1, Line 19) not simply a "sensitivity analysis"? - Discussion chapter: Consists of sentences/paragraphs, which can be defined as general knowledge (e.g. Page 26. Line 16; Page 26, Line 24) or which should be familiar by the readers of the journal (e.g. Page 28, Line 12).

Secondary Comments

- Page 1, Line 21: "... and they suggest that simulated recharge is sensitive to both precipitation (overall amount and temporal distribution) and land cover." Is this one of the main results of your work and is it really a new finding? - Page 2, Line 30: The sentence is difficult to understand. - Page 4, Line 17: Please, rephrase the long sentence (and consider deleting the first part of the sentence). - Page 6, Line 12: Could you please add a bit more information on how diffuse and concentrated recharge is considered by the model. - Page 12, Line 12/13/14: Please, consider using SI-Units. - Page 13 Seasonality of vegetation: Are you using the same seasonality on every study site irrespective of the local climate and vegetation type? - Page 15, Line 4: Please consider splitting the sentence. - Equation 17/18: Please remove the units from the equation and mention both parameters in the text, e.g. "1. E$_{act,bow}$ [mmmonth-1], a corrected value that assumes that latent heat ($ðİŘ£ðİŘÿ$ [MJ.m$-2$.month$-1$]) and sensible heat ($ðİŘż$ [MJ.m$-2$.month$-1$]) have similar errors (referred to as Bowen ratio estimate): - Page 17, Line 29: Please, rephrase the sentence. - Page 24, Line 4: Please, rephrase the sentence. - Figure 4: The Figure presents the results in a confusing way and some of the values exceed the constrained parameter ranges according to Table 3.

Minor comments and typographical errors
- Please, use a consistent citation style. - Please, use a consistent style for figure references. Two different versions exist: Fig. and Figure - Units -> Replace the dots by multiplication sign or even better delete them. - Page 2, Line 2: . . .world. . . - Page 2, Line 2: For instance, . . . - Page 5, Line 4: . . . (Hartmann et al., 2015). This . . . (space missing) - Page 5, Line 33: . . . to represent. . . - Page 7, Line 13: . . . the following formulas . . . - Page 12, Line 13: . . . is the psychrometric constant, . . . - Page 15, Line 18: . . . data processing are reported . . . - Page 20, Line 18: red lines -> the a priori information are indicated by black lines in Figure 4! - Page 29, Line 14: We, therefore, . . . - Figure 5, Line 4: . . . percentage of Eact . . . - Figure 6: Line 5: Blue . . . - Figure 9: Line 4: remove the open bracket

---

## Referee Comment (RC2) · Anonymous Referee #2 · 8 May 2018

The paper proposed by Sarrazin et al. aims at adding a new evaporation formulation to the recharge model VarKarst which specialises on the hydrology of karst systems. The aim of this development is to make the model suitable for exploring the impact of climate and land surface changes on these very sensitive hydrological structures. The main themes of these improvements are to be applicable at the large scale and to be parsimonious.

I believe this model fails on both accounts for a simple reason, the authors have neglected the fact that evaporation is strongly controlled by the diurnal cycle of radiation

and atmospheric processes. One of the main consequences of climate change is to modify the diurnal cycle at the surface and in the atmosphere. Thus the application of V2Karst to climate change is bound to produce unrealistic sensitivities. The model would be more parsimonious and more robust (because based on stronger physical grounds) if it would explicitly represent the diurnal cycle. Furthermore this enhancement of VarKarst neglects 30 years in the developments of land surface models. These models do not represent hydrological processes and even less karst systems, and are rightfully criticized for this. But they have specialized on the surface/atmosphere exchanges and in particular the simulation of evaporation, vegetation processes and infiltration. At no moment do the authors refer to developments in one of the three leading land surface models ( JULES, ORCHIDEE and CLM) or their application to the 4 FLUXNET stations used here. A simple Google search would have shown to the authors that these open-access codes (Thanks in great part to GMD !) perform much better on these sites and do not require the tuning of so many parameters. Furthermore they are designed to be applicable at the large scale.

I would recommend to reject the paper and encourage the authors to download one of the above mentioned land surface models and couple it to VarKarst. This would produce a model for these sensitive hydrological regions which is much more robust and produces more credible result for the impact of climate and land-cover changes.

I am sorry to have to make such a harsh recommendation to GMD and in the following I will detail where I believe the basic assumptions of the authors to be wrong and where the usage of developments made for land surface models would help.

Rational to explicitly represent land cover properties :

It is laudable for the authors to use the Penman-Monteith formulation for potential evaporation. But should they have paid attention to its derivation, they would have noted that it only provides potential evaporation over a infinitesimal time intervals as it assumes that atmospheric variables and surface states do not evolve through other processes.

[Figure]

A constant Rn(t) or ra,can(t) over the course of the day is a very unsatisfactory assumption, especially under a changing climate. Because of very contrasted impact of changing atmospheric composition on long-wave and short-wave radiation, we can encounter the same Rn but with very different radiation balance, turbulent fluxes and surface temperatures. The authors will find in the literature a number of paper which examine the impact of climate change on the different potential evaporation formulation. They all recommend to use sub-diurnal solutions because of the modified diurnal dynamics.

The parsimony of our representation of nature if not for us to choose. We have to prove that certain simplifications in the representation of surface processes are valid for the application we envisage. The authors aim to develop a model valid at the large scale, for climate and land surface change. Is it then reasonable to assume that over the course of a day ra,can does not change ? I think the development of land surface models has shown that one cannot neglect the diurnal dynamic of the opening of the stomata, the soil moisture stress or the dependence of stomatal resistance to atmospheric $CO_2$ concentration. If the authors believe that they have found a way to represent with a single daily value these complex processes and their interaction with the environment they should let the world know as it would allow land surface models to be simplified.

In their rational for their modelling strategy they only mention one land surface model : ISBA in its 1998 version. This is not up to date. Even ISBA has evolved since then and does not use any more a Jarvis type parametrisation. It now also uses a Ball-Berry type formulation which balances carbon uptake and transpiration. Please note that ISBA operates at sub-diurnal time steps.

Soil water balance :

The explanations of the evolution of moisture in the unsaturated zone is not very clear to me. It looks to me like a superposition of buckets with the addition of lateral flows. It

has been the experience in the land surface model community that this simple representation of soil moisture limits the ability to simulate the impact of stresses on transpiration. This is particularly critical in semi-arid those encountered at 3 of the selected FLUXNET stations. What is the reasoning of the authors behind this simplification in the treatment of the unsaturated zone, apart from "parsimony" ?

Evapotranspiration :

Only one vegetation type seems to be allowed per grid-box, is this correct ? Because of the strong heterogeneity of the distribution of vegetation, it has been the experience of the community that a larger number of plant functional types is needed per grid box. The strict minimum has been found to be a low and a high vegetation. This simplification will be critical for the application to larger domains and in particular in semi-arid regions where the competition of the various vegetation types for water is critical. Please explain here as well why the literature on vegetation modelling is not relevant for this model.

Canopy interception :

This is another topic where the community has acquired a rich experience which could benefit the authors. The representation of canopy interception at different temporal and spatial scales has been fiercely debated in the early 90s. Thus a number of parametrisations were developed to take into account the spatial and temporal variability of interception. This would be relevant here.

Do the authors believe that a rainfall event in the evening or in the morning produces the same interception loss ? Does a rainfall event of 10mm/h and 100mm/h produce the same interception ? So does the assumption of treating these processes averaged over the day have any implication on the sensitivity of V2Karst to climate change ? We know that rainfall intensity and possibly also the time of day at which precipitation will occur will change in a warmer climate.
May I point out at this stage that precipitation intensification has been observed at the sub-diurnal range. Daily mean rainfall has not yet been too much affected by climate change. On the other hand, hourly precipitation rates have been increasing faster than expected from the Clausius-Capleyron relation. Thus, the virtual experiments experiments proposed in section 4.3 are not relevant for climate change. The authors are referred to the wealth of literature published on this topic in the last few years.

Transpiration from vegetated soil :

Transpiration does not occur from the soils (as written in the paper) but from the stomata in the vegetation. This is not a negligible detail. Firstly the stomata only open when daylight is present and thus photosynthesis can occur. During the early afternoon, once the water in contact with the roots and within the plant has been evaporated, transpiration declines. This is caused by the slower diffusion of water within the soil which limits the supply. This is known to be a critical process for transpiration and which will be affected by higher $CO_2$ concentration which will lead plants to reduce the opening of their stomata. I guess these processes are neglected in the proposed model, why ? It would be a very interesting topic to see how this early afternoon depression of transpiration is affected by climate change for plants on karstic soils. It is bound to be different than on loamy soils for instance.

Sorry, the assumption "... evaporation from interception is constant throughout the day ..." is not valid and will change with climate and land surface type.

Parameter estimation :

The proposed parameter estimation is difficult to interpret in view of the strong hypothesis made in the basic equations of the model. The 15 parameters of this model are so conceptual, i.e. far away from first physical principles, that indeed they can all be tuned. But given the large number of "tunable parameters" can it not be expected that the model can be made to match any dataset ? To me hveg, LAI(min,max) or z0 are not "tunable parameters" as they can either be measured or derived from turbulence

theory. Furthermore I find that the range of values explored for these parameters (Table 3 does not provide the limits for all 15 parameters) is much wider than realistic values I have observed.

Land surface models also use the FLUXNET observations to "tune parameters". But fewer parameters are adjusted and only those where the definition itself includes processes which are not modelled, i.e. are conceptual. Furthermore these parameters are specific to the plant functional type present at the FLUXNET station and then then transferred to the larger scale. This is the value of using vegetation classes in land surface models.

A simple internet search for FLUXNET and the name of one of the leading land surface models, returns a large number of papers. Some where the models are simply validated and others where the observations are used to refine some vegetation parameters. The authors should have done that search during the development of their model.

Conclusion :

I am very sorry to have to write this review about the development of V2Karst. I know what a huge effort it is to develop a complex numerical model. As the authors are working in Britain, I would recommend that they look into the JULES land surface model. It is freely available and could be coupled to VarKarst to produce a very innovative tool which could indeed allow to explore the consequences of climate and land surface changes on water resources of karst aquifers. This need, to initiate a convergence between hydrological and land surface modelling, has been recognized by NERC and lead to the initiation of the HydoJULES program. The authors should contact the leaders of this program to obtain help.

---

## Author Comment (AC1) · 19 Jun 2018

**COMMENT 1:** *The manuscript presents a modified version of the large-scale karst recharge model VarKarst. The here presented model (V2Karst V1.0) replaces the simplified evapotranspiration (ET) component (empirical Priestley-Taylor equation) by the physical based Penman-Monteith equation (for potential evapotranspiration). The authors also include a separate calculation of the different evaporation processes in order to use the model for climate and land cover change impact studies. The model extension increases the number of parameters. The general functioning as well as the influence of the new parameters are tested by applying the new model to four study sides, different in climate and vegetation. The manuscript is a novel extension of previous work published by the research group. The conceptual description and the numerical adaptation of the processes are sound. The results of the model application on the four test sides prove the general functioning of the new model. However, the manuscript also has weak points, which are mainly related to the presentation of the method and the results. The manuscript can easily be shortened by 10-20% without losing important information. The presentation of the results needs to be improved, especially since it is difficult to distinguish between observed values and modeled results. My detailed comments are listed below.*

> **REPLY 1:** We wish to thank the Reviewer for appreciating the novelty and soundness of our work and for the detailed comments, especially regarding the karst aspect of our work.
>
> We realise that the manuscript is rather long in parts, and we will shorten our revised version.
>
> We will also clarify the distinction between observed and simulated values. In brief, the parameter estimation approach (Sect. 4.1) uses measurements of weather variables to force the model and measurements of model output to define the soft rules. The global sensitivity analysis (Sect 4.2) uses measurements of weather variables to force the model, but no measurements of model output. Finally, the virtual experiments (Sect 4.3) do not use any measurements. In the revised version of the manuscript, we will clarify the point by changing the sentence that introduces the method section p16 L8-12, for instance as: "To test the plausibility of V2Karst realisations at FLUXNET sites, we estimate the model parameters by constraining model simulations with actual ET and soil moisture observations (Sect. 4.1), and we perform two sensitivity analyses using measured data to force the model (Sect. 4.2), and synthetic forcing data and land cover change scenarios (Sect. 4.3)."
>
> Part of the confusion regarding the distinction between observed and simulated values may also come from the fact that Figure 6 includes a lot of information. We will simplify Figure 6 by removing the yellow and green lines corresponding to the two additional estimates of measured evapotranspiration (no correction and residual correction) and report the additional information on these two quantities in our supplementary material. These two additional estimates of evapotranspiration were not directly used in applying the soft rules but were only used to assess the uncertainty in latent heat measurements. Likewise, we will shorten the paragraph on the assessment of uncertainty in latent heat measurements (Sect 3.2 p15 L26-p16 L6), to improve readability.

**MAIN COMMENTS**

**COMMENT 2:** *The purpose of V2Karst V1.0 is to predict recharge in karst regions. The authors mention that "a large part of the groundwater recharge occurs as concentrated and fast flow in large apertures and the other part as diffuse and slow flow in the matrix (Hartmann and Baker, 2017)." Especially concentrated recharge, e.g. fast infiltration into sinkholes, can be considered as a short-term process and is entirely uncoupled from soil and/or vegetation properties (overland flow -> percolation).*

*I assume that your model, calculating the water balance, underestimates the recharge in karst regions dominated by concentrated recharge. Do you think your model is able to equally represent both recharge processes?*

**REPLY 2:** We agree with the need for representing concentrated recharge processes. In karst systems, infiltration and recharge can be slow and diffuse in the matrix and fast and concentrated in large conduits or fissures that act as preferential flow pathways. Lateral flow at the surface and in the epikarst is an important mechanism that concentrates the infiltrating water into the preferential flow pathways (Jeannin and Grasso, 1997; Williams, 1983, 2008). In particular, the epikarst plays a role of temporary storage that can redistribute fast and concentrated recharge (Williams, 1983, 2008). Figure R1.c below provides a conceptual model of the soil and epikarst processes of a real karst system.

V2karst's representation of concentrated and diffuse infiltration and recharge, which is the same as in the VarKarst model (Hartmann et al., 2015), follows this conceptual model (Figure R1.a). V2karst represents the spatial variability of subsurface properties observed in karst systems by dividing each model simulation unit into a number of vertical compartments that have different soil and epikarst properties. Parameters for each model compartment (soil and epikarst storage capacities and epikarst outflow coefficient) are estimated using a distribution function. The daily water balance is explicitly evaluated for each model vertical compartment. Additionally, when a given compartment saturates (both soil and epikarst stores), its saturation excess generates a surface lateral flow to the next unsaturated compartments that have higher storage capacities and higher permeabilities. In this representation, surface and subsurface lateral flow are thus lumped together. Conceptually, in V2Karst, the direct contribution of precipitation to infiltration and recharge can be associated with diffuse infiltration and recharge, while the contribution of lateral flow can be associated with concentrated infiltration and recharge. Hence, the V2karst structure allows to account for the interplay between diffuse and concentrated infiltration and recharge processes.

The representation of karst processes in V2Karst and VarKarst is based on a previous karst model developed for applications at the local scale introduced in Hartmann et al. (2012). The structure of this previous models explicitly represents lateral flow both at the surface and in the epikarst, in agreement with understanding of the flow mechanisms in the epikarst (e.g. Williams, 1983, 2008). It was tested using hydrodynamic and hydrochemical information at stalactite drips in a karstic cave in Hartmann et al. (2012). However, simplifications have been introduced for applications at the large-scale, given the limited information available to constrain the additional parameters required in the previous karst model to represent lateral flow in the epikarst.

We are aware of the fact that, in V2Karst's and VarKarst's representation, concentrated recharge is not entirely uncoupled from the soil and vegetation properties as it is observed in real karst systems. However, a previous study by Hartmann et al. (2017) compared simulated recharge with VarKarst and independent estimates of recharge and showed that there was no systematic bias in the simulations (Figure R2 below). Moreover, the study by Hartmann et al. (2017) showed that recharge values simulated with VarKarst were significantly higher than recharge values simulated with models that do not include karst processes (Figure R2 below).

As stated in REPLY 9, we will briefly explain how diffuse and concentrated recharge is represented in VarKarst and V2karst p6 L12.

[Figure]

**Figure R1**. (a) Schematic description of the VarKarst model for one model grid cell including the soil (yellow) and epikarst storages (grey) and the simulated fluxes, (b) its gridded discretization over karst regions and (c) the subsurface heterogeneity that its structure represents for each grid cell.
Figure taken from Hartmann et al. (2015, Figure 1)

[Figure]

**Figure R2**. Comparison of simulated and observed recharge. **In blue are reported values simulated with the VarKarst model (heterogeneous representation)**, in yellow values simulated with the PCR-GLOBWB model (homogeneous representation, i.e. absence of karst processes in the model representation), and in green the Varkarst model with subsurface heterogeneity processes turned off. Whiskers indicate the simulation uncertainty (1 SD) for simulations with VarKarst. **No statistical difference (5% significance level) was found between simulations with Varkarst and the observations**. mm/a, millimeters per year.
Figure taken from Hartmann et al. (2017, Figure 2)

**COMMENT 3:** *I am aware of the fact that the manuscript is focused on the implementation and the testing of the new evapotranspiration component. Since soil layers in karst regions can be thin or even totally absence the authors should consider this fact in the interpretation of the results.*

**REPLY 3:** Large differences in soil depth can indeed be observed across different karst landscapes. To apply the VarKarst model over Europe and the Mediterranean, Hartmann et al. (2015) have identified four main karst landscapes with different soil depths and degrees of karstification based on climate and topography (Humid, Mountain, Mediterranean and Desert landscapes). Different values of the parameter $V_{soi}$ (mean soil water capacity) have been applied in the different landscapes. In particular, very small values of $V_{soi}$ have been used in arid areas (i.e. desert landscape in Hartmann et al. (2015)),where soils tend to be very thin.

We are also aware of the fact that soils may be absent in some karst areas (e.g. karren field in high mountain areas, see e.g. Hartmann et al. (2014a)), and that these areas may consequently produce very high recharge amounts. The model does not account for the fact that the soil may be absent and always includes a soil layer, although the soil layer can be very thin and therefore can have a limited impact on recharge. This assumption seems reasonable given the large extent of the simulation units for large-scale applications (0.25°x0.25° or 0.5°x0.5° cell in previous applications). In fact, we can assume that soil layers can always be found in such large simulation units. However, for model applications at high resolutions, we recognise the fact that an explicit consideration of bare rock regions should be included in the model.

We will briefly discuss this point in Sect. 6.3 that discusses the application of V2karst over large domains.

**COMMENT 4:** *The manuscript lacks a description/characterization of the four karst regions (e.g. by describing dominant karst features or by the interpretation of spring hydrographs).*

**REPLY 4:** In this study, we focus on groundwater recharge, which is a key component of the water balance. Recharge characterises the amount of renewable groundwater, and therefore the amount of groundwater available to human consumption and ecosystems (e.g. Scanlon et al., 2006; Döll and Fiedler, 2008; Wada et al., 2012). We do not model groundwater flow and storage nor spring discharge. Therefore, to test the model, we focus on datasets that can be related to the fluxes and states simulated by V2Karst, i.e. soil moisture and evapotranspiration as in Hartmann et al. (2015). No datasets providing time series of recharge are available. We do not use spring hydrographs, because spring discharge depends on groundwater routing and is not commensurate with groundwater recharge. Spring discharge measurements were used to extensively test a previous versions of the VarKarst model including groundwater storage and routing to the spring, at different locations in Europe and the Mediterranean (Hartmann et al., 2013a, 2013b, 2014b).

Table B1 describes the four sites, and more specifically the soil depth and bedrock. We have realised that we have exchanged the name of the two French sites in Table B1 (Puéchabon is actually the French 2 site and Font-Blanche the French 1 site) and we will correct this mistake in the revised version of the manuscript. We did not add more details on the sites in the main text to limit the length of the manuscript.

**COMMENT 5:** *In general, a differentiation between different karst systems and therefore the wide variety of hydraulic properties dominating the recharge pattern (see above) is missing here.*

> **REPLY 5**: A large variability in hydraulic properties and in recharge patterns can indeed be observed across karst systems ( e.g. Klimchouk and Ford, 2000; Hartmann et al., 2014a). For applications over large-scale domain, Hartmann et al. (2015) identified four typical karst landscapes with different hydraulic properties, based on climate and topography (as mentioned in REPLY 3). This simplified classification of the simulation domain in four landscapes was introduced to enable large-scale applications of the model.

> In Sect. 6.3 (discussion of V2karst large-scale application), we will better highlight the fact that future large-scale applications of V2Karst will need to account for the variability observed across karst systems, for instance using a simplified classification as in Hartmann et al. (2015).

**COMMENT 6:** *As already mentioned, the current manuscript is too long and needs to be shortened:*
**1)** *(Almost) every section starts with a short introduction on the section. Most of them are redundant.*
**2)** *The authors use wordy descriptions instead of clear words for describing their work. Is a "virtual experiment with synthetic data to assess the sensitivity" (Page 1, Line 19) not simply a "sensitivity analysis"?*
**3)** *Discussion chapter: Consists of sentences/paragraphs, which can be defined as general knowledge (e.g. Page 26. Line 16; Page 26, Line 24) or which should be familiar by the readers of the journal (e.g. Page 28, Line 12).*

> **REPLY 6:** We thank the Reviewer for suggesting how to shorten the manuscript and we reply below to the three points raised by the Reviewer.

> **1)** We will revise the introductions of the sections to make them more informative (see REPLY 1) and we will remove them where appropriate (p4 L13-15, p20 L3-5 and p23 L17-19).

> **2)** The virtual experiments are indeed sensitivity analyses. However, an important aspect is that we used synthetic data, which has been done only in few studies (e.g. the studies reported p18 L20-29). The reason for using synthetic data is that it allows to explore conditions beyond what was historically observed, and it is therefore useful to better understand potential impact of changes in climate. It also permits unequivocal attribution of changes in model output to changes in model inputs.
> Therefore, we think that it is important to mention that we used synthetic data p1 L19. However, as discussed in REPLY 7, we will clarify this part of the abstract. We will also check for wordy descriptions in the manuscript and revise them where appropriate.

> **3)** We agree that the discussion section can be shortened and clarified and more specifically at the places indicated by the Reviewer.
> We will clarify the discussion of the virtual experiments (Sect. 6.2). Firstly, the virtual experiments confirm that the model behaves reasonably, since it shows sensitivities to both precipitation (overall amount and temporal distribution) and land cover, in agreement with previous studies and general understanding. Secondly, the virtual experiments allow to unequivocally and quantitatively characterise the relationship between simulated recharge and both precipitation (overall mean and temporal distribution) and land cover. Therefore, virtual

experiments are a complementary approach to model applications using site-specific data to assess the impact of climate and land cover changes on simulated recharge.

We will revise the discussion on large-scale application of the model (Sect. 6.3) and we will clarify in particular the points mentioned in REPLY 3 and 5.

**SECONDARY COMMENTS**

**COMMENT 7:** *- Page 1, Line 21: ". . . and they suggest that simulated recharge is sensitive to both precipitation (overall amount and temporal distribution) and land cover." Is this one of the main results of your work and is it really a new finding?*

> **REPLY 7:** We agree that we need to reformulate this sentence. We refer to REPLY 6 for a clarification of the objectives of the virtual experiments.
> We will replace the sentences p1 L19-24 ("Then, we use virtual experiments with synthetic data to assess the sensitivity of simulated recharge to precipitation characteristics and land cover. Results reveal how both vegetation and soil parameters control the model behaviour, and they suggest that simulated recharge is sensitive to both precipitation - overall amount and temporal distribution- and land cover") for instance by: "Then, virtual experiments with synthetic data confirms that the model has sensible sensitivities to precipitation (overall amount and temporal distribution) and land cover in light of previous studies In addition, results allow to quantify the relationship between changes in simulated recharge and changes in precipitation and land cover characteristics."

**COMMENT 8:**
*- Page 2, Line 30: The sentence is difficult to understand.*
*- Page 4, Line 17: Please, rephrase the long sentence (and consider deleting the first part of the sentence).*

> **REPLY 8:** We will reformulate these two sentences.

**COMMENT 9:** *Page 6, Line 12: Could you please add a bit more information on how diffuse and concentrated recharge is considered by the model.*

> **REPLY 9:** We refer to REPLY 2 for a detailed explanation of the conceptualisation of recharge processes in V2Karst and we will add a brief explanation on this p6 L12.

**COMMENT 10:** *Page 12, Line 12/13/14: Please, consider using SI-Units.*

> **REPLY 10:** We have adopted these units because they are typically used in the evapotranspiration literature (e.g. Allen et al., 1998; Shuttleworth, 1993, 2012).
> The revised version of the manuscript will include the unit of net radiation ($R_n$ [$MJ.m^{-2}.d^{-1}$]) which is missing.

**COMMENT 11:** *Page 13 Seasonality of vegetation: Are you using the same seasonality on every study site irrespective of the local climate and vegetation type?*

**REPLY 11**: Typically, in hydrological models, vegetation seasonality is represented using different schemes with different levels of complexity or is neglected completely (Table A3). We chose to implement a simple representation (piecewise linear function). In this way, we can test the impact of vegetation seasonality by assessing the sensitivity of recharge to the seasonality parameter LAImin. We applied the same function at all sites, but we used different values of the seasonality parameter LAImin as indicated in Table 3. The timings of the four phases reported in Eq. (14) are appropriate for study sites that are located in the northern hemisphere and that have natural vegetation.

We will add a sentence p13 to clarify the fact that the timings of the four phases of the seasonality model should be adapted to the application domain.

**COMMENT 12:**
- *Page 15, Line 4: Please consider splitting the sentence.*
- *Equation 17/18: Please remove the units from the equation and mention both parameters in the text, e.g. "1. Eact,bow [mmmonth1], a corrected value that assumes that latent heat (ð˙IR£ð ˘ ˙IRÿ [MJ.m ˘ −2.month−1]) and sensible heat (ð˙IR˘ z [MJ.m ˙ −2.month−1]) have similar errors (referred to as Bowen ratio estimate):*
- *Page 17, Line 29: Please, rephrase the sentence.*
- *Page 24, Line 4: Please, rephrase the sentence.*

**REPLY 12:** We will apply these changes in the revised version of the manuscript.

**COMMENT 13:** *Figure 4: The Figure presents the results in a confusing way and some of the values exceed the constrained parameter ranges according to Table 3.*

**REPLY 13**: We realise that the plot we used – a parallel coordinate plot – is not familiar to all readers in earth sciences. A parallel coordinate plot is a two-dimensional plot that allows to visualise a multidimensional space (here the space of the model parameters and outputs). In Figure 4, each line represents a combination of model parameters values (normalised) and the corresponding model output values (normalised). Parallel coordinate plots are increasingly used (e.g. Inselberg, 2009; Kasprzyk et al., 2013; Pianosi et al., 2017), and are implemented for instance in the Matlab Statistics and Machine Learning Toolbox (function "parallelcoords") and in the SAFE toolbox for sensitivity analysis we utilised in our paper (Pianosi et al., 2015). Initially, we sampled the parameter space within wide ranges (Table 2), and we applied a priori information on parameter ranges (Table 3) only in rule 5. Therefore, prior to application of rule 5, parameter values can exceed the ranges of Table 3 (yellow, light blue, green and dark blue lines in Figure 4). Instead, posterior to application of rule 5, all parameter values should be within the ranges of Table 3 (red lined in Figure 4).

The Reviewer may be referring to the fact that some red lines slightly exceed the black vertical lines in Figure 4. This is a plotting issue and it will be corrected in the revised version of the manuscript.

**MINOR COMMENTS AND TYPOGRAPHICAL ERRORS**

**COMMENT 14***: Please, use a consistent citation style.*

   **REPLY 14**: We will check the citation style throughout the manuscript.

**COMMENT 15***: Please, use a consistent style for figure references. Two different versions exist: Fig. and Figure.*

   **REPLY 15:** We will replace 'Figure' by 'Fig.' p6 L22, p6 L28, p14 L14 so that the referencing of the figures is consistent with guidelines of GMD available at (https://www.geoscientific-model-development.net/for_authors/manuscript_preparation.html): *The abbreviation "Fig." should be used when it appears in running text and should be followed by a number unless it comes at the beginning of a sentence, e.g.: "The results are depicted in Fig. 5. Figure 9 reveals that...".*

**COMMENT 16:**
*- Units -> Replace the dots by multiplication sign or even better delete them.*
*- Page 2, Line 2: . . .world. . .*
*- Page 2, Line 2: For instance, . . .*
*– Page 5, Line 4: . . . (Hartmann et al., 2015). This . . . (space missing)*
*- Page 5, Line 33: . . . to represent. . .*
*- Page 7, Line 13: . . . the following formulas . . .*
*- Page 12, Line 13: . . . is the psychrometric constant, . . .*
*- Page 15, Line 18: . . . data processing are reported . . .*
*- Page 20, Line 18: red lines -> the a priori information are indicated by black lines in Figure 4!*
*- Page 29, Line 14: We, therefore, . . .*
*- Figure 5, Line 4: . . . percentage of Eact . . .*
*- Figure 6: Line 5: Blue . . .*
*- Figure 9: Line 4: remove the open bracket*

   **REPLY 16:** The revised version of the manuscript will include these corrections.

**References to support our reply to the Reviewer**

Allen, R. G., Pereira, L. S., Raes, D. and Smith, M.: Crop evapotranspiration: Guidelines for computing crop requirements, FAO Irrigation and Drainage Paper 56, Food and Agriculture Organization (FAO), Rome, Italy., 1998.

Döll, P. and Fiedler, K.: Global-scale modeling of groundwater recharge, Hydrol. Earth Syst. Sci., 12(3), 863–885, doi:10.5194/hess-12-863-2008, 2008.

Hartmann, A., Lange, J., Weiler, M., Arbel, Y. and Greenbaum, N.: A new approach to model the spatial and temporal variability of recharge to karst aquifers, Hydrol. Earth Syst. Sci., 16(7), 2219–2231, doi:10.5194/hess-16-2219-2012, 2012.

Hartmann, A., Weiler, M., Wagener, T., Lange, J., Kralik, M., Humer, F., Mizyed, N., Rimmer, A., Barberá, J. A., Andreo, B., Butscher, C. and Huggenberger, P.: Process-based karst modelling to relate hydrodynamic and hydrochemical characteristics to system properties, Hydrol. Earth Syst. Sci., 17(8), 3505–3521, doi:10.5194/hess-17-3305-2013, 2013a.

Hartmann, A., Barberá, J. A., Lange, J., Andreo, B. and Weiler, M.: Progress in the hydrologic simulation of time variant recharge areas of karst systems - Exemplified at a karst spring in Southern Spain, Adv. Water Resour., 54, 149–160, doi:10.1016/j.advwatres.2013.01.010, 2013b.

Hartmann, A., Goldscheider, N., Wagener, T., Lange, J. and Weiler, M.: Karst water resources in a changing world: Review of hydrological modeling approaches, Rev. Geophys., 52(3), 218–242, doi:10.1002/2013RG000443, 2014a.

Hartmann, A., Mudarra, M., Andreo, B., Marin, A., Wagener, T. and Lange, J.: Modelling spatiotemporal impacts of hydroclimatic extremes on groundwater recharge at a Mediterranean karst aquifer, Water Resour. Res., 50(8), 6507–6521, doi:10.1002/2014WR015685, 2014b.

Hartmann, A., Gleeson, T., Rosolem, R., Pianosi, F., Wada, Y. and Wagener, T.: A large-scale simulation model to assess karstic groundwater recharge over Europe and the Mediterranean, Geosci. Model Dev., 8(6), 1729–1746, doi:10.5194/gmd-8-1729-2015, 2015.

Hartmann, A., Gleeson, T., Wada, Y. and Wagener, T.: Enhanced groundwater recharge rates and altered recharge sensitivity to climate variability through subsurface heterogeneity, Proc. Natl. Acad. Sci., 114(11), 2842–2847, doi:10.1073/pnas.1614941114, 2017.

Inselberg, A.: Parallel coordinates: Visual multidimensional geometry and its applications, Springer, New York., 2009.

Jeannin, P.-Y. and Grasso, D. A.: Permeability and hydrodynamic behavior of karstic environment, in Karst Waters Environmental Impact, edited by G. Gunay and A. I. Johnson, pp. 335–342, A.A. Balkema, Rotterdam, Netherlands., 1997.

Kasprzyk, J. R., Nataraj, S., Reed, P. M. and Lempert, R. J.: Many objective robust decision making for complex environmental systems undergoing change, Environ. Model. Softw., 42, 55–71, doi:10.1016/j.envsoft.2012.12.007, 2013.

Klimchouk, A. B. and Ford, D. C.: Types of karst and evolution of hydrogeologic setting, in Speleogenesis. Evolution of Karst Aquifers, edited by A. B. Klimchouk, D. C. Ford, A. Palmer, and W. Dreybrodt, pp. 45–53, National Speleological Society, Huntsville, Alabama, U.S.A., 2000.

Pianosi, F., Sarrazin, F. and Wagener, T.: A Matlab toolbox for Global Sensitivity Analysis, Environ. Model. Softw., 70, 80–85, doi:10.1016/j.envsoft.2015.04.009, 2015.

Pianosi, F., Iwema, J., Rosolem, R. and Wagener, T.: A multimethod Global Sensitivity Analysis approach to support the calibration and evaluation of Land Surface Models, in Sensitivity Analysis in Earth Observation Modelling, edited by G. Petropoulos and P. Srivastava, pp. 125–144, Elsevier Inc., 2017.

Scanlon, B. R., Keese, K. E., Flint, A. L., Flint, L. E., Gaye, C. B., Edmunds, W. M. and Simmers, I.: Global synthesis of groundwater recharge in semiarid and arid regions, Hydrol. Prpcesses, 20(15), 3335–3370, doi:doi.org/10.1002/hyp.6335, 2006.

Shuttleworth, W. J.: Evapotranspiration, in Handbook of Hydrology, edited by D. R. Maidment, p. 4.1-4.53, McGraw-Hill inc., New York., 1993.

Shuttleworth, W. J.: Terrestrial Hydrometeorology, John Wiley & Sons, Ltd, Chichester, UK., 2012.

Wada, Y., Van Beek, L. P. H. and Bierkens, M. F. P.: Nonsustainable groundwater sustaining irrigation: A global assessment, Water Resour. Res., 48(6), W00L06, doi:10.1029/2011WR010562, 2012.

Williams, P. W.: The role of the subcutaneous zone in karst hydrology, J. Hydrol., 61(1–3), 45–67, doi:10.1016/0022-1694(83)90234-2, 1983.

Williams, P. W.: The role of the epikarst in karst and cave hydrogeology : a review, Int. J. Speleol., 37, 1–10, doi:10.5038/1827-806X.37.1.1, 2008.

---

## Author Comment (AC2) · 19 Jun 2018

**OVERALL REPLY:** We thank the Reviewer for the extensive review and take the opportunity to clarify some important points of our approach, while responding to the criticisms made. We provide detailed responses to specific comments below, but we thought it helpful to summarise the key point here: our manuscript introduces a new evapotranspiration component into a previously developed large-scale hydrological model for karst areas (Hartmann et al., 2015, GMD). We are therefore following an approach to large-scale hydrological modelling that has been widely used for climate change impact studies (e.g. Beyene et al., 2010; Sperna Weiland et al., 2012; Gosling et al., 2017). The Reviewer focuses on the use of land surface models to the same issue, which is an alternative approach currently taken to simulate climate and land cover change impacts on hydrological variables. However, we are not attempting to build a land surface model here, but rather to advance a hydrological model for large-scale applications. Indeed, we ourselves have argued in the past that these two communities should come closer together and learn from each other (Archfield et al., 2015, WRR).

We provide below a point-by-point reply to the comments of the Reviewer with additional references to previous studies that help us support our modelling choices (see list of references at the end of this report). We have found it difficult to address some of the Reviewers' specific comments, because he/she did not provide any reference to support his/her statements.

**COMMENT 1**: *The paper proposed by Sarrazin et al. aims at adding a new evaporation formulation to the recharge model VarKarst which specialises on the hydrology of karst systems. The aim of this development is to make the model suitable for exploring the impact of climate and land surface changes on these very sensitive hydrological structures. The main themes of these improvements are to be applicable at the large scale and to be parsimonious.*

*I believe this model fails on both accounts for a simple reason, the authors have neglected the fact that evaporation is strongly controlled by the diurnal cycle of radiation and atmospheric processes. One of the main consequences of climate change is to modify the diurnal cycle at the surface and in the atmosphere. Thus the application of V2Karst to climate change is bound to produce unrealistic sensitivities. The model would be more parsimonious and more robust (because based on stronger physical grounds) if it would explicitly represent the diurnal cycle.*

**REPLY 1**: Our model is in line with widely published approaches to climate change assessment using large-scale hydrological models (e.g. Beyene et al., 2010; Sperna Weiland et al., 2012; Gosling et al., 2017), that all neglect the diurnal cycle and apply hydrological models at a daily time step. We provide further details on this issue in the reply to the reviewer's COMMENT 3 (Penman-Monteith equation) and 9 (canopy interception). We agree that there is indeed a strong need for better comparison studies to understand how neglected processes (e.g. of diurnal cycles) affect climate change impact studies, but this is beyond the scope of the study presented here.

Moreover, the reviewer wrote *"One of the main consequences of climate change is to modify the diurnal cycle"*. We agree that global average radiative forcing is projected to increase (IPCC, 2013; Van Vuuren et al., 2011), and average land surface air temperature is documented to have already increased globally (IPCC, 2013, Chapter 2 p187-188 for a global assessment, 2014, Chapter 23 p1275-1276 for an assessment for Europe) and is projected to further increase (IPCC, 2013, Chapter 12 p1062-1064 for a global assessment, 2014, Chapter 23 p1276 for an assessment for Europe). However, to our knowledge, changes in the diurnal cycle appear to be much more uncertain. Changes in temperature are more documented, possibly because the historical temperature record is more accurate than the other climate variables (Allen and Ingram, 2002).

We are aware of multiple studies that analysed the past changes in diurnal temperature range (i.e. difference between minimum and maximum daily temperature). A summary of these studies is provided in the Fifth Assessment Report of the Intergovernmental Panel on Climate Change (IPCC, 2013). They overall indicate that decreases in diurnal temperature range have been observed, but these decreases were found to be smaller than changes in average temperature (IPCC, 2013, Chapter 2, p188). Moreover, some studies suggest that these apparent changes in diurnal temperature range may be attributed to non-climatic factors (IPCC, 2013, Chapter 2, p188). In this regard, a more recent study points out that the drivers of the past observed changes in diurnal temperature ranges are still not well understood (Davy et al., 2017). Additionally, it has been shown that climate models involved in the Coupled Model Intercomparison Project Phase 5 (CMIP5, scientific basis for the IPCC fifth Assessment Report) are reproducing poorly the past observed changes in diurnal temperature range (Lewis and Karoly, 2013), which suggests that future projections in diurnal temperature range have large uncertainties.

**COMMENT 2**:*Furthermore this enhancement of VarKarst neglects 30 years in the developments of land surface models. These models do not represent hydrological processes and even less karst systems, and are rightfully criticized for this. But they have specialized on the surface/atmosphere exchanges and in particular the simulation of evaporation, vegetation processes and infiltration. At no moment do the authors refer to developments in one of the three leading land surface models ( JULES, ORCHIDEE and CLM) or their application to the 4 FLUXNET stations used here. A simple Google search would have shown to the authors that these open-access codes (Thanks in great part to GMD !) perform much better on these sites and do not require the tuning of so many parameters. Furthermore they are designed to be applicable at the large scale.*
*I would recommend to reject the paper and encourage the authors to download one of the above mentioned land surface models and couple it to VarKarst. This would produce a model for these sensitive hydrological regions which is much more robust and produces more credible result for the impact of climate and land-cover changes. I am sorry to have to make such a harsh recommendation to GMD and in the following I will detail where I believe the basic assumptions of the authors to be wrong and where the usage of developments made for land surface models would help.*

**REPLY 2**: For clarity, we have structured our reply in three parts: (1) we explain why we chose to develop V2karst as an evolution of a parsimonious hydrological model, rather than using more complex land surface models, (2) we attempt to compare the performance of V2Karst and land surface models at the four FLUXNET sites used in our study and (3) we specify the changes that we intend to introduce in the revised version of the manuscript to clarify these points. We do not discuss V2karst representations here, as we will explain them more specifically in REPLY 3 (Penman-Monteith equation), REPLY 6 (soil water balance), and REPLY 9 (interception).

**1. Reasons for developing a parsimonious hydrological model**

Land Surface Models are undeniably crucial tools because they include state-of-the-art scientific understanding of moisture and energy processes. However, parallel to the development of land surface models, a wealth of studies have drawn attention to the problem of dealing with model complexity in the context of natural systems. In fact, in natural systems, controlled experimentation to ascertain model formulations is not possible and model components and parameters tend to be poorly defined, especially at large-scales, because of a lack of data and knowledge (e.g. Young et al., 1996; Abramowitz et al., 2008; Beven and Cloke, 2012; IPCC, 2013, Chapter 9, pp790-791; Hong et al., 2017; Haughton et al., 2018). For example, Beven and

Cloke (2012) highlight the fact that more complex models do not necessarily produce more robust predictions, presumably because of our lack of knowledge of natural processes and because of the uncertainty in estimates/observations of the variables needed to run such complex models. Therefore, regarding future modelling challenges, Beven and Cloke (2012) have argued that understanding which parameterisation may be more appropriate and assessing model uncertainties may be more relevant than further increasing the detail of process representation.

A recent study published in GMD (Haughton et al., 2018) argues that: "*In general, numerical LSMs* [Land Surface Models] *have become increasingly complex over the last 5 decades, expanding from basic bucket schemes to models that include tens or even hundreds of processes involving multiple components of the soil, biosphere, and within-canopy atmosphere. Model components may have been added on to existing models without adequate constraint on component parameters (Abramowitz, 2013) or without adequate system closure (Batty and Torrens, 2001). New component parameters may be calibrated against existing model components, leading to problems of equifinality (Medlyn et al., 2005), non-identifiability (Kavetski and Clark, 2011), and epistemological holism (Lenhard and Winsberg, 2010). These problems can often only be overcome by ensuring that each component is itself well constrained by data and numerically stable. As noted earlier, these conditions rarely exist for any given component.*" In fact, although land surface models have a strong physical basis, they also include many empirical functions that are typically difficult to constrain (Mendoza et al., 2015). More critically, the fact that land surface models include a large number of parameters (many of which hard-coded, as highlighted in Cuntz et al. (2016) or Mendoza et al. (2015)) hampers an exhaustive assessment of uncertainty and sensitivity of model predictions (Young et al., 1996). We further discuss the issue of parameter estimation in land surface models in REPLY 15. Moreover, the study by Haughton et al. (2018, GMD) shows that land surface models can be outperformed by simple empirical models, in line with the results of previous studies conducted within the Land Surface Model Benchmarking Evaluation Project (PLUMBER, Best et al., 2015).

Importantly, land surface models simulate a large range of different fluxes (e.g. sensible heat, latent heat, ground heat flux, radiation, runoff, $CO_2$) and the sensitivity of model parameters has been reported to vary depending on the simulated flux considered (Cuntz et al., 2016; Rosero et al., 2010; Rosolem et al., 2012). For instance, Cuntz et al. (2016) showed that a large number of parameters of the Noah land surface model are non-influential or have a very small influence on total simulated runoff. This means that parts of the land surface models may be simplified when the objective is to simulate hydrological variables. In this sense, Hong et al. (2017) highlighted the fact that model development should account for the model intended uses.

V2Karst aims to simulate seasonal and annual groundwater recharge, because these variables are appropriate to characterise the amount of renewable groundwater, and hence the amount of groundwater available to human consumption and ecosystems (e.g. Scanlon et al., 2006; Döll and Fiedler, 2008; Wada et al., 2012). As generally done in hydrological models (Table A1 of our manuscript), V2Karst focuses on solving the water balance, while it does not solve the energy balance, as land surface models do. In fact, V2Karst is not meant to be used for assessing the energy fluxes (radiation, sensible heat, latent heat and ground heat flux). An additional motivation for us not to solve the energy balance is that its inclusion increases model complexity and computational cost tremendously, which makes it difficult to perform a full uncertainty and sensitivity analysis to assess the adequateness of the different model components. By focusing on the water balance instead and by using parsimonious representations, we enable all

components of V2Karst to be subject to uncertainty and sensitivity analysis, as we have explained in Sect. 2.1 p4 L16- p5 L7 in our manuscript.

**2. Performance of Land Surface Models at the four FLUXNET sites**

We are not aware of studies that would allow us to directly infer that JULES, ORCHIDEE and CLM have better performance compared to V2Karst at the four FLUXNET sites, as stated by the reviewer. We found six studies in which the JULES, ORCHIDEE, CLM or Noah land surface models were applied at some of the FLUXNET sites we have used (Anav et al., 2010; Davin et al., 2011; Zhao et al., 2012; Kuppel et al., 2012; Van den Hoof et al., 2013; Chaney et al., 2016). In the following paragraphs, we explain in detail that either the results of these studies cannot be compared to our results, or that the performance of the land surface models appears to be similar to or slightly inferior to those of V2Karst.

The results of four of these studies cannot be compared with our results. The study by Chaney et al. (2016) does not specifically report performance results for any of the four FLUXNET sites. The study by Davin et al. (2011) analysed a full Regional Climate Model including the CLM model and did not present performance results regarding latent heat and soil moisture simulations, which are the two variables we analysed in our manuscript. The study by Kuppel et al. (2012) did not analyse the bias or the correlation coefficient between measured and simulated latent heat/evapotranspiration, while these two metrics were used in our analyses (Sect 4.1 of our manuscript). In fact, both metrics are important to characterise the hydrological performance of the recharge model, since the bias assesses the performance in reproducing the overall water balance, while the correlation coefficient assesses the consistency between the temporal pattern of simulated and observed latent heat/evapotranspiration. Finally, the study by Anav et al. (2010) did not report quantitative performance metrics for the individual sites.

In the two "comparable" studies ( Zhao et al., 2012; Van den Hoof et al., 2013), land surface models show similar or slightly inferior performance compared to V2Karst, although it is difficult to make a fair comparison because the time periods analysed were not the same as in our manuscript. Zhao et al. (2012) assessed the squared correlation coefficient $R^2$ between monthly observed and simulated latent heat at the French 2 site (Puéchabon) using the ORCHIDEE model and local forcing data. They found a value of $R^2$ of 0.59, i.e. a correlation coefficient of $\sqrt{0.59} = 0.77$ (Zhao et al., 2012, fourth row in Figure 6). This is comparable to the performance of the V2Karst model, since we identified simulations for which the correlation coefficient between monthly simulated and observed ET was higher than 0.77 (Figure 4 in our manuscript). We wish to mention that there appears to be a mistake in the labels in Figure 6 in Zhao et al. (2012) and therefore we are not completely sure we have read the figure correctly. Finally, in Van den Hoof et al. (2013), the JULES model was tested at the German site (Hainich) using local forcing data. The bias between observed and simulated evapotranspiration for the best performing version of the model is around 29% (this percentage is inferred from Figure 2.b in Van den Hoof et al. (2013), which reports that the mean annual observed ET is 280 mm.y[-1] and the simulated mean annual ET is 360 mm.y[-1]). The bias is therefore above the limit of acceptance of 20% that we have used in our manuscript. Van den Hoof (2013) did not provide the monthly correlation values for individual sites, but we can infer that, for the German site, the monthly correlation coefficient is between 0.7 and 1 (Van den Hoof et al., 2013, Figure 5.b), which is comparable to our study.

**3. Changes in the revised version of the manuscript**

Based on the Reviewer's comment, we realise that in our manuscript we have not stated clearly enough our philosophy in developing V2Karst in line with other large-scale hydrological models. We will clarify this point in the introduction p4 L3-11 (in the paragraph that states the objectives of the study). We agree that for completeness our manuscript should also refer to the literature on land surface modelling. In the revised version of our manuscript we will thus include a discussion of the ET components of land surface models in Sect. 2.1. However, we would avoid including a comparison between the performance of the V2Karst model and other land surface models at FLUXNET sites because, as shown in our point (2) above, it is difficult to make a fair comparison between different studies that used different metrics and different time periods to evaluate the models.

**COMMENT 3:** *Rational to explicitly represent land cover properties :*
*It is laudable for the authors to use the Penman-Monteith formulation for potential evaporation. But should they have paid attention to its derivation, they would have noted that it only provides potential evaporation over a infinitesimal time intervals as it assumes that atmospheric variables and surface states do not evolve through other processes. A constant Rn(t) or ra,can(t) over the course of the day is a very unsatisfactory assumption, especially under a changing climate. Because of very contrasted impact of changing atmospheric composition on long-wave and short-wave radiation, we can encounter the same Rn but with very different radiation balance, turbulent fluxes and surface temperatures. The authors will find in the literature a number of paper which examine the impact of climate change on the different potential evaporation formulation. They all recommend to use sub-diurnal solutions because of the modified diurnal dynamics.*

> **REPLY 3:** We are aware of the fact that the Penman-Monteith equation has been determined over an infinitesimal time step (e.g. Shuttleworth, 2012) and that, therefore, using sub-daily rather than daily time step for the calculation of the equation provides conditions that are closer to its theoretical derivation. Nevertheless, Penman-Monteith equation has been shown to be applicable at daily time step and is widely used at daily time step.
>
> In particular, Allen et al. (2006) commented on applications of the Penman-Monteith reference crop framework, that was designed by the Food and Agriculture Organization of the United Nations (FAO) and the American Society of Civil Engineers (ASCE). The framework has standardised the use of the Penman-Monteith equation for both hourly and daily time step (Allen et al., 1998; Walter et al., 2005) with a focus on agricultural crop. Allen et al. (2006) stated the following: "*The favorable performance of the PM equation in many studies, when applied with 24-h (and even monthly) time steps, is somewhat surprising, since the formulation of the combination equation (combined energy balance and aerodynamic components) theoretically requires weather inputs on a nearly instantaneous basis. The general consistency and accuracy of the PM method for 24-h time steps speaks to the combination equation's robustness in estimating evaporative behavior given a particular set of meteorological conditions.*" More recent papers still support both hourly and daily applications of the Penman-Monteith reference crop framework (e.g. Pereira et al., 2015). The daily framework has been included for instance in an R package for simulating evapotranspiration at daily time step (Guo et al., 2016).
>
> The Penman-Monteith equation is widely used at daily time step. Among its applications, we can cite the Moderate Resolution Imaging Spectroradiometer (MODIS) Evapotranspiration product

(Mu et al., 2011; Running et al., 2017). Furthermore, the Penman-Monteith equation is implemented in the large-scale hydrological models PCR-GLOBWB, MacPDM and VIC (these models are reviewed in Table A1-A3 of our manuscript). These models have been applied at daily time step in climate change impact studies. For instance, Gosling and Arnell (2016) used MacPDM, Sperna Weiland et al. (2012) used PCR-GLOBWB and Beyene et al. (2010) used VIC at daily time step. These three hydrological models are also included in the Inter-Sectoral Impact Model Intercomparison Project (ISI–MIP, Warszawski et al., 2014) that aims to assess the impact of climate change. In that context, daily simulations of PCR-GLOBWB, MacPDM and VIC are analysed for instance by Gosling et al. (2017).

Given all the above, we think that applying the Penman-Monteith equation at daily time step is one possible choice that is consistent with a wide range of published studies. The direct comparison between running models for hydrological projections at daily or sub-daily resolutions, while considering the uncertainties present in the different projections (as discussed in Beven and Cloke, 2012), would indeed be very interesting but beyond the scope of our study.

Our manuscript refers to other hydrological models that applied the Penman-Monteith equation at a daily time (Table A2). Additionally, in the revised version of the manuscript, p12 when discussing the Penman-Monteith equation, we will refer to Allen et al. (2006) (see above citation) to further justify the use of the Penman-Monteith equation at a daily time step.

**COMMENT 4**: *The parsimony of our representation of nature if not for us to choose. We have to prove that certain simplifications in the representation of surface processes are valid for the application we envisage. The authors aim to develop a model valid at the large scale, for climate and land surface change. Is it then reasonable to assume that over the course of a day ra,can does not change ? I think the development of land surface models has shown that one cannot neglect the diurnal dynamic of the opening of the stomata, the soil moisture stress or the dependence of stomatal resistance to atmospheric CO2 concentration. If the authors believe that they have found a way to represent with a single daily value these complex processes and their interaction with the environment they should let the world know as it would allow land surface models to be simplified.*

**REPLY 4**: We explain in REPLY 1 and 2 why we have chosen to include a parsimonious process representation in V2Karst, which is in line with well-established large-scale hydrological modelling studies on climate change impacts. An in-depth comparison with land surface models would be interesting, but it is beyond the scope of our work here.

The evaluation of the Penman-Monteith equation, which requires the specification of the aerodynamic resistance ra_can, is well documented at a daily times step (see REPLY 3).
Our representation does include soil moisture stress using a linear function of soil moisture which multiplies the potential evapotranspiration rate (Eq.(7) p10, and Eq.(8) p11 of our manuscript). This formulation is similar to the previous version of the model (VarKarst, Hartmann et al., 2015) and to other hydrological models (Table A2 of our manuscript).

V2Karst neglects the dependence of the stomatal resistance on atmospheric factors such as temperature, radiation, humidity and CO2 concentration, as implemented for instance in Ball-Berry (Sellers et al., 1996) and Jarvis-Stewart (Jarvis, 1976; Stewart, 1988) schemes. Stomatal resistance is assessed as a function of a minimum stomatal resistance (which is a parameter of the model) and the leaf area index (Eq.(13) p13). This formulation is similar to the hydrological

model MacPDM (Table A2) that has been applied in climate impact studies (e.g. Gosling and Arnell, 2016) as already reported in REPLY 3. Compared to MacPDM, V2Karst also includes the dependence of the stomatal resistance on the leaf area index, although we recognise that this approach is still a strong simplification of stomatal processes. The reason for our choice comes from the fact that the value of the stomatal resistance is poorly characterised for large-scale applications, because few ground measurements of stomatal resistance are available (some are reported in Breuer et al. (2003) and Körner et al. (1995)). Moreover, the temporal variability of stomatal resistance makes its measurements particularly difficult to interpret (Breuer et al., 2003) and therefore to use in modelling applications. We have discussed the lack of ground measurements of stomatal resistance (and more generally of vegetation properties) in Sect. 2.1 p4 L20 and in Sect S1 of our supplementary material. The Ball-Berry and Jarvis-Stewart parameterisations of stomatal resistance significantly increase model complexity and in particular introduce several empirical parameters, whose values are uncertain because of the lack of stomatal resistance measurements. The constant value of the stomatal resistance implemented in V2Karst allows us to lump all the uncertainties around stomatal resistance estimation into a single parameter, that we can include in our uncertainty and sensitivity analysis. In this way, we can easily analyse the impact of the value of the stomatal resistance on simulated recharge. We obviously do not claim to "*have found a way to represent with a single daily value these complex processes and their interaction with the environment*", but simply to have included in our model a simple approach to control the overall impact of this variable on recharge predictions.

The temporal behaviour of the stomatal resistance could be investigated in V2Karst in future studies. However, we believe that, as a first version of the model, such in-depth investigation is beyond the scope of this study.

**COMMENT 5**: *In their rational for their modelling strategy they only mention one land surface model : ISBA in its 1998 version. This is not up to date. Even ISBA has evolved since then and does not use any more a Jarvis type parametrisation. It now also uses a Ball-Berry type formulation which balances carbon uptake and transpiration. Please note that ISBA operates at sub-diurnal time steps.*

**REPLY 5**: The Reviewer is right, the description of the ISBA model is not up-to-date. In the revised version of the manuscript, we will refer to the new version of the ISBA model in Sect. 2.1 when briefly discussing the representation of ET in land surface models (as mentioned in REPLY 2).

**COMMENT 6**: *Soil water balance :*
*The explanations of the evolution of moisture in the unsaturated zone is not very clear to me. It looks to me like a superposition of buckets with the addition of lateral flows. It has been the experience in the land surface model community that this simple representation of soil moisture limits the ability to simulate the impact of stresses on transpiration. This is particularly critical in semi-arid those encountered at 3 of the selected FLUXNET stations. What is the reasoning of the authors behind this simplification in the treatment of the unsaturated zone, apart from "parsimony" ?*

**REPLY 6:** The evolution of moisture in the saturated zone in V2Karst is the same as in VarKarst (Hartmann et al., 2015, GMD). This representation is supported by physical reasoning and by previous testing of the VarKarst model.

As explained in Sect. 2.3.2 p8-9, percolation from a given soil layer ($Soi_i$) to the underlying soil layer ($Soi_k$) is equal to the saturation excess in $Soi_i$. Percolation from the deeper (third) soil layer to the epikarst layer is also equal to the third soil layer saturation excess. As explained p1 L10-11 and p8 L24 – p9 L3, the reasoning behind this simplification is that soils in karst areas typically have a high clay content (Blume et al., 2010), and therefore tend to have low unsaturated permeability (Clapp and Hornberger, 1978). Instead, saturated permeability is typically high because clay soil generally have cracks that can act as preferential flow pathways under wet conditions (Lu et al., 2016).

The representation of the process of lateral flow is a feature specific to the VarKarst model, which, together with the explicit representation of sub-surface sub-grid heterogeneity (using model vertical compartments), makes the model more appropriate for karst areas than other large models. The adequacy of the soil water balance representation in VarKarst has been tested in previous studies for previous versions of the model (Hartmann et al., 2012, 2015, 2017). In particular, Hartmann et al. (2015, 2017) showed that there is no systematic bias in the model predictions, and that recharge predictions produced by VarKarst are significantly higher than those produced by models that do not include karst processes. Further discussion of the representation of diffuse and concentrated flow in V2Karst and of the testing of the previous versions of the model are given in our response to the first review of this manuscript (REPLY 2 in 'Response to Referee #1' available at https://www.geosci-model-dev-discuss.net/gmd-2017-315/).

**COMMENT 7**: *Evapotranspiration :*
*Only one vegetation type seems to be allowed per grid-box, is this correct ? Because of the strong heterogeneity of the distribution of vegetation, it has been the experience of the community that a larger number of plant functional types is needed per grid box. The strict minimum has been found to be a low and a high vegetation. This simplification will be critical for the application to larger domains and in particular in semi-arid regions where the competition of the various vegetation types for water is critical.*

**REPLY 7:** We agree that, for large-scale applications, the model should be able to represent different vegetation types within a given simulation grid cell. The model can account for sub-grid heterogeneity in vegetation type using a 'tile' approach.
A 'tile' approach consists of subdividing each model grid cell in a number of independent units (tiles), each of which has a specific land cover (e.g. short or tall vegetation). The model can then be evaluated separately over each tile. The overall simulated fluxes for a given grid cell are computed as the area weighted average of the fluxes calculated over the tiles. The same approach is also used in other large-scale hydrological models, for instance in the Mac-PDM model (Gosling and Arnell, 2011) and in the VIC model (Bohn and Vivoni, 2016; Liang et al., 1994, http://vic.readthedocs.io/en/master/Overview/ModelOverview/).

We will add a sentence in Sect. 6.3 (that discusses large-scale application of V2Karst) to clarify the fact that sub-grid heterogeneity in vegetation types can be treated following a tile approach.

**COMMENT 8**: *Please explain here as well why the literature on vegetation modelling is not relevant for this model.*

> **REPLY 8**: We are not sure which specific literature the reviewer is referring to. In Sect. 2.1 and Tables A1-A3 of our manuscript we have provided a detailed review of ET and vegetation modelling in other large-scale hydrological models. As mentioned in REPLY 2, we will also add a brief discussion of the ET components of land surface models in Sect. 2.1 of our revised manuscript.
> We are aware of the fact that our manuscript is rather long and we have already been advised by another reviewer to shorten it, hence we would avoid adding further details on vegetation modelling besides Sect. 2.1.

**COMMENT 9**: *Canopy interception :*
*This is another topic where the community has acquired a rich experience which could benefit the authors. The representation of canopy interception at different temporal and spatial scales has been fiercely debated in the early 90s. Thus a number of parametrisations were developed to take into account the spatial and temporal variability of interception. This would be relevant here. Do the authors believe that a rainfall event in the evening or in the morning produces the same interception loss ? Does a rainfall event of 10mm/h and 100mm/h produce the same interception ? So does the assumption of treating these processes averaged over the day have any implication on the sensitivity of V2Karst to climate change ? We know that rainfall intensity and possibly also the time of day at which precipitation will occur will change in a warmer climate.*

> **REPLY 9**: V2Karst implements a parsimonious daily interception model as done in other large-scale hydrological models.
>
> In V2Karst, for each day, evaporation from the interception store is assessed over the vegetated fraction as the maximum value between daily precipitation, potential evapotranspiration and interception storage capacity. Our model is similar to the large-scale hydrological models WaterGap, PCR-GLOBWB and VIC (Table A3 in our manuscript). These models have been simulated at daily step to assess the hydrological impact of future changes in climate, for instance in Gosling et al. (2017) within the Inter-Sectoral Impact Model Intercomparison Project (ISI–MIP, Warszawski et al., 2014), and in Döll et al. (2018) for WaterGap, in Sperna Weiland et al. (2012) for PCR-GLOBWB and Beyene et al. (2010) for VIC. Past studies have shown that interception can be reasonably represented at a daily time scale (Savenije, 1997; De Groen and Savenije, 2006; Gerrits et al., 2009).
>
> We are aware of the fact that more sophisticated interception schemes have been developed, as reviewed for instance in (Muzylo et al., 2009). These more complex schemes include additional parameters compared to our formulation that includes only one parameter, namely the interception storage capacity. We chose to implement a simple interception scheme for large-scale applications because (1) the physical processes and atmospheric conditions driving evaporation from canopy interception are still poorly understood (Van Dijk et al., 2015), (2) ground measurements of evaporation from interception that would be necessary to constrain model parameters are not available (Fatichi and Pappas, 2017; Miralles et al., 2016), and (3) ground measurements of canopy resistance are also sparse and affected by large uncertainties (Van Dijk et al., 2015).

We have discussed the issues arising from lack of measurements in large-scale model applications in our model development rationale (Sect. 2.1 p4 L19-20) and in Sect S1 of our supplementary material. Similar to our representation of stomatal resistance (REPLY 4), since the representation of interception processes can only be poorly constrained for large-scale applications, we chose a simple interception scheme. In this way, we can easily include the uncertainty in interception representation in our analyses and gain a systematic understanding of the impact of this uncertainty on seasonal and annual recharge predictions (our variables of interest).

**COMMENT 10**: *May I point out at this stage that precipitation intensification has been observed at the sub-diurnal range. Daily mean rainfall has not yet been too much affected by climate change. On the other hand, hourly precipitation rates have been increasing faster than expected from the Clausius-Capleyron relation. Thus, the virtual experiments experiments proposed in section 4.3 are not relevant for climate change. The authors are referred to the wealth of literature published on this topic in the last few years.*

**REPLY 10**: We have actually found evidence suggesting that daily mean rainfall has been affected by climate change, and therefore analysing the impact of changes in daily precipitation (overall mean and temporal distribution), as done in our virtual experiment, is relevant.
The last IPCC assessment reported that many regions show a significant trend (positive or negative) in mean daily precipitation for the past period (IPCC, 2013 and particularly Fig. 2.33, reported as Figure R1 below). Additionally, a more recent study by Ye et al. (2016) found that an increase in mean daily precipitation intensity occurred in all seasons in the northern Eurasia region over the period 1966-2010. Regarding future climate, model projections indicate changes in seasonal precipitation and changes in the 95 percentile of daily precipitation, for instance in Europe (IPCC, 2014, Chapter 23, Section 23.2.2.2 and Fig. 23-2).

In the revised version of the manuscript, we will clarify the fact that daily precipitation is likely to change in the introduction section (Sect. 1).

[Figure]

**Figure R1**. Trend in daily precipitation intensity over the period 1951-2010. Trends were calculated only for grid boxes that had at least 40 years of data during this period and where data ended no earlier than 2003. Grey areas indicate incomplete or missing data. Black plus signs (+) indicate grid boxes where trends are significant (i.e., a trend of zero lies outside the 90% confidence interval).
Source of the Figure: (IPCC, 2013, Figure 2.33)

**COMMENT 11**: *Transpiration from vegetated soil :*
*Transpiration does not occur from the soils (as written in the paper) but from the stomata in the vegetation. This is not a negligible detail. Firstly the stomata only open when daylight is present and thus photosynthesis can occur. During the early afternoon, once the water in contact with the roots and within the plant has been evaporated, transpiration declines. This is caused by the slower diffusion of water within the soil which limits the supply. This is known to be a critical process for transpiration and which will be affected by higher CO2 concentration which will lead plants to reduce the opening of their stomata. I guess these processes are neglected in the proposed model, why ? It would be a very interesting topic to see how this early afternoon depression of transpiration is affected by climate change for plants on karstic soils. It is bound to be different than on loamy soils for instance.*

> **REPLY 11**: We agree with the reviewer that "*transpiration does not occur from the soil*". In the revised version of the manuscript, we will correct this mistake. More specifically, p11 L1 we will replace 'Transpiration from vegetated soil' by 'Transpiration over the vegetated fraction'.
>
> Regarding the reviewer's comment "*I guess these processes are neglected in the proposed model, why ?*", as explained in detail in REPLY 4, we neglect the dynamics of stomatal resistance because lack of observations does not allow constraining complex stomatal resistance schemes at a large-scale. We instead use a constant value of the stomatal resistance (scaled by the leaf area index) which enables us to easily analyse the effect of uncertain stomatal resistance on simulated recharge by including the stomatal resistance parameter in our sensitivity analysis.
>
> "*How this early afternoon depression of transpiration is affected by climate change for plants on karstic soils*" is a very interesting research question but is beyond the scope of our research. Again, the aim of our modelling activity here is to (reasonably, given all data and knowledge limitations) assess the effects of climate and land cover changes on groundwater recharge at the spatial and temporal scale of interest (large/regional; seasonal/annual). Our objective is not to assess the effect of climate variations on every specific process involved in plant transpiration.

**COMMENT 12**: *Sorry, the assumption "... evaporation from interception is constant throughout the day ..." is not valid and will change with climate and land surface type.*

> **REPLY 12**: We agree with the reviewer that the justification of Eq. (11) p12 L2-3 (fraction of the day with wet canopy) has been expressed incorrectly, ultimately misleading the interpretation of our proposed approach.
>
> We do not actually make the strong assumption that "evaporation from interception is constant throughout the day". We instead assess the fraction of the day with wet canopy as the fraction of available energy that was used to evaporate water from the interception store. This conceptual representation was proposed in Kergoat et al. (1998) and it was adopted in the dynamic vegetation model LPJ (Gerten et al., 2004; Murray, 2014) presented in Table A1-A3. In V2Karst we revised this formulation, which in the referenced studies uses the fraction of daytime with wet canopy (hence assuming that ET fluxes occur during daytime only), by using the fraction of the entire day, given that night time ET fluxes have been shown to be also important ( e.g. Pearce et al., 1980; Sugita and Brutsaert, 1991; Kelliher et al., 1992).

In the revised version of the manuscript, we will clarify the point by replacing the statement p12 L2-3 ("The fraction of the day with wet canopy $t_{wet}$(t) [−] is estimated by assuming that evaporation from interception is constant throughout the day ...") by "The fraction of the day with wet canopy $t_{wet}$(t) [−] is estimated as the fraction of available energy that was used to evaporate water from the interception store …".

**COMMENT 13**: *Parameter estimation :*

*The proposed parameter estimation is difficult to interpret in view of the strong hypothesis made in the basic equations of the model. The 15 parameters of this model are so conceptual, i.e. far away from first physical principles, that indeed they can all be tuned. But given the large number of "tunable parameters" can it not be expected that the model can be made to match any dataset ? To me hveg, LAI(min,max) or z0 are not "tunable parameters" as they can either be measured or derived from turbulence theory.*

**REPLY 13:** We agree with the Reviewer that (1) the ability to identify parameter sets that produce predictions close to observations is a necessary but not sufficient condition for the plausibility of the V2Karst model and (2) some parameters of V2Karst can be derived from a priori information. We think that we have accounted for these two points in our methodology, as we explain below.

Regarding (1), we agree that a model can produce reasonable predictions by 'incorrectly' activating processes to compensate for structural deficiencies. For this reason, in our evaluation of V2Karst, in addition to the comparison between observations and predictions (Sect. 4.1), we also performed two sensitivity analyses: one using measured forcing data (Sect. 4.2) and the other one using synthetic forcing data (Sect. 4.3). The aim of these analyses was precisely to assess whether the sensitivities of V2Karst outputs across sites are consistent with our understanding of the physical characteristics of those sites, and hence whether the model reproduces the observations for the right reasons by activating the appropriate controlling factors at the right place. In our manuscript, we have discussed the fact that the global sensitivity analysis appled to V2Karst showed a set of sensitivities that are interpretable in light of the different climatic conditions at the four FLUXNET sites (see Sect. 5.2.2 p22 and in Sect. 6.1 p25). Such use of sensitivity analysis as a tool for model diagnostic evaluation (verification of model structures) has been successfully applied already in several modelling studies (e.g. Rosero et al., 2010; Reusser and Zehe, 2011; Rosolem et al., 2012; Hartmann et al., 2013).

Regarding (2), we have actually used measurements at the FLUXNET sites to constrain the value of parameters hveg, LAI(min,max), Vr and Vsoi. As explained in Sect. 4.1, our parameter estimation strategy is based on the sequential application of 'soft rules' to identify plausible parameter values within an initial sample spanning over a wide range. Rules 1-4 select 'plausible' parameter sets based on their ability to produce model outputs consistent with observations, while the last rule (rule 5) selects the parameter sets that are consistent with a priori information about the sites. Constraining parameter values based on a priori information in the last step, instead of doing it in the first step as typical of other estimation strategies, provides us with a more stringent test of the adequacy of the model structure. In fact, if the parameter sets identified as plausible by Rules 1-4 (consistency with observations) are also plausible according to Rule 5 (a priori knowledge about the site), we can indeed conclude that the model 'gives the right response for the right reason'. If instead some parameters sets identified as 'plausible' by the comparison with

observations (Rules 1-4) are then ruled out when including a priori information (Rule 5), this indicates some deficiencies in the model structure that are being 'compensated for' by tunable parameters, and points us to the model components that need improvement. In our experiment, presumably because too few data are available to constrain the simulations, Rules 1-4 result in little constraining of the parameter ranges (Figure 4, Sect. 5.1.1). However, importantly, we can identify parameter sets that satisfy all rules simultaneously and that are therefore consistent with a priori information at the site, while also producing predictions consistent with the available observations.

A similar strategy was used in previous studies, and more specifically in Hartmann (2015) to estimate the parameters of the VarKarst model and in Rosero et al. (2010) to estimate soil and vegetation parameters of the Noah land surface model.

In future large-scale applications of V2Karst, we envisage that parameters hveg, LAI(min,max), Vr and Vsoi can be a priori constrained based on grid-cell specific information about vegetation and karst landscape type (Vsoi was constrained based on karst landscape type in Hartmann et al. (2015)).

In this sense, in Sect. 6.3, we explain that some model parameters will be estimated using a priori information for the different simulation grid-cells (p27 L21 – p28 L2). In the revised version of the manuscript, we will clarify Sect 6.3 to better convey this point.

**COMMENT 14:** *Furthermore I find that the range of values explored for these parameters (Table 3 does not provide the limits for all 15 parameters) is much wider than realistic values I have observed.*

**REPLY 14:** Table 2 provides the wide ranges for all 15 parameters that were used to derive the initial parameter sample for our parameter estimation strategy. Table 3 provides the a priori ranges that were used for applying our soft rule 5. These ranges are defined only for those parameters for which site-specific a priori information was available.

All these ranges were determined in accordance with values found in the literature. References and detailed explanations are given in Sect. S3 of our Supplementary material (Table S11 and S12 p13-15).

**COMMENT 15**: *Land surface models also use the FLUXNET observations to "tune parameters". But fewer parameters are adjusted and only those where the definition itself includes processes which are not modelled, i.e. are conceptual. Furthermore these parameters are specific to the plant functional type present at the FLUXNET station and then then transferred to the larger scale. This is the value of using vegetation classes in land surface models. A simple internet search for FLUXNET and the name of one of the leading land surface models, returns a large number of papers. Some where the models are simply validated and others where the observations are used to refine some vegetation parameters. The authors should have done that search during the development of their model.*

**REPLY 15**: We are aware of the fact that land surface models indeed typically tune a very limited number of model parameters. However, the results of recent studies suggest that excluding many parameters from calibration and uncertainty/sensitivity analysis raises a number of issues and that parameter estimation strategies of land surface models should be enhanced, as explained in the next paragraphs.

Among the parameters of land surface models that are not 'tuned', some are considered as 'physical' parameters and are commonly read from look-up tables (more specifically soil and

vegetation parameters are assigned based on soil texture and vegetation type respectively) or are set to site-specific values when measurements are available. Other parameters, which have an empirical basis, are set to constant values or are even hard-coded, i.e. are embedded in the model code as fixed values (e.g. coefficients to describe a particular shape for a curve in the model, such as the exponent used in the snow depletion curves for the melting season in the Noah land surface model as reported in Mendoza et al., (2015)). Hogue et al. (2006) highlighted the fact that many parameters in land surface models cannot be directly obtained from field measurements but through curve-fitting techniques, such as parameters controlling the soil hydraulic properties (derived for instance in Clapp and Hornberger, 1978).

Previous studies suggest that reading parameters from look-up table or setting them to fixed value are not satisfying strategies. Some studies have put into questions the physical meaning of vegetation and soil parameters in land surface models, and therefore the fact that they can be assigned from look-up tables or from site measurements. Rosero et al. (2010) analysed the sensitivity and optimal values of different vegetation and soil parameters for different version of the Noah land surface model along a precipitation gradient in the southern USA. Their results suggest that climate strongly controls the optimal soil and vegetation parameters values, which means that vegetation and soil parameters do not strictly represent vegetation and soil properties but also account for other properties. Hogue et al. (2006) analysed the differences in optimal values of different vegetation and soil parameters across five different land surface models (BUCKET, CHASM, BATS1e, BATS2, and Noah) and for five different vegetated sites in the USA. They found significant differences in optimal parameter values for the same parameters and the same site across models, which suggests that the vegetation and soil parameters have a different meaning across models. Therefore, they conclude that land surface models should be interpreted as simplified conceptual representations of natural systems, in which parameters are also conceptual representations of physical properties. The incommensurability between model parameters and physical properties can be explained by the scale mismatch between physical variables that can be actually measured and model parameters that represent cell average properties. An additional possible explanation for this incommensurability is that models implement governing equations that are well established to describe the behaviour of the system over small scales (e.g. Darcy's law), but that may not be valid for applications over larger domains (Kirchner, 2006).

Moreover, it has been shown that some empirical parameters that are typically fixed to constant values, and in particular some hard-coded parameters, have a significant impact on model predictions and therefore these parameters would need to be included in the parameter estimation strategy, i.e. to be 'tuned' (Cuntz et al., 2016; Mendoza et al., 2015).

The fact that land surface models include a large number of parameters that would potentially need to be 'tuned', and the limited availability of data to constrain model predictions at large-scales (as discussed in Sect. S1 of our supplementary material, and in REPLY 4 and 9 above) make the calibration of land surface models particularly challenging in large-scale applications. How to enhance parameter estimation of land surface models is indeed still an open question (e.g. Chaney et al., 2016).

**COMMENT 16**: *Conclusion :*
*I am very sorry to have to write this review about the development of V2Karst. I know what a huge effort it is to develop a complex numerical model. As the authors are working in Britain, I would recommend that they look into the JULES land surface model. It is freely available and could be coupled to VarKarst to produce a very innovative tool which could indeed allow to explore the consequences of climate and land surface changes on water resources of karst aquifers. This need, to initiate a convergence between hydrological and land surface modelling, has been recognized by NERC and lead to the initiation of the HydoJULES program. The authors should contact the leaders of this program to obtain help.*

> **REPLY 16:** The V2karst V1.0 model is an extension of a large-scale hydrological model previously published in GMD (VarKarst, Hartmann et al., 2015) that aims to simulate groundwater recharge in karst areas under changing environmental boundary conditions. Hydrological models are widely applied to study groundwater recharge at large-scales. As we have stressed in REPLY 2 and 15, land surface models simulate many more fluxes than hydrological models (e.g. sensible heat, latent heat, ground heat flux, radiation, CO2), which implies that they include many more parameters that are difficult to constrain in large-scale applications. Since our objective is to predict seasonal and annual recharge, which is the key variable of interest for water resources management in karst areas, and not to assess the other fluxes simulated by land surface models, we chose to build on and expand a hydrological model instead of using a land surface model. While we acknowledge that the approach proposed by the Reviewer may be an alternative route to achieve our goal, we hope we have provided a convincing rationale for our modelling choices and we would take the opportunity to revise our manuscript to clarify these points.

**References to support our reply to the Reviewer**

Abramowitz, G.: Calibration, compensating errors and data-based realism in LSMs, Presentation, 2013.

Abramowitz, G., Leuning, R., Clark, M. and Pitman, A.: Evaluating the performance of land surface Models, J. Clim., 21(21), 5468–5481, doi:10.1175/2008JCLI2378.1, 2008.

Allen, M. R. and Ingram, W. J.: Constraints on future changes in climate and the hydrologic cycle, Nature, 419(6903), 224–232, doi:10.1038/nature01092, 2002.

Allen, R. G., Pereira, L. S., Raes, D. and Smith, M.: Crop evapotranspiration: Guidelines for computing crop requirements, FAO Irrigation and Drainage Paper 56, Food and Agriculture Organization (FAO), Rome, Italy., 1998.

Allen, R. G., Pruitt, W. O., Wright, J. L., Howell, T. A., Ventura, F., Snyder, R., Itenfisu, D., Steduto, P., Berengena, J. and Yrisarry, J. B.: A recommendation on standardized surface resistance for hourly calculation of reference ETo by the FAO56 Penman-Monteith method, Agric. Water Manag., 81(1–2), 1–22, doi:10.1016/j.agwat.2005.03.007, 2006.

Anav, A., D'Andrea, F., Viovy, N. and Vuichard, N.: A validation of heat and carbon fluxes from high-resolution land surface and regional models, J. Geophys. Res. Biogeosciences, 115(G4), 13,281-13,235, doi:10.1029/2009JG001178, 2010.

Archfield, S. A., Clark, M., Arheimer, B., Hay, L. E., McMillan, H., Kiang, J. E., Seibert, J., Hakala, K., Bock, A., Wagener, T., Farmer, W. H., Andréassian, V., Attinger, S., Viglione, A., Knight, R.,

Markstrom, S. and Over, T.: Accelerating advances in continental domain hydrologic modeling, Water Resour. Res., 51(12), 10078–10091, doi:10.1002/2015WR017498, 2015.

Batty, M. and Torrens, P. M.: Modeling complexity: the limits to prediction, Cybergeo Eur. J. Geogr., doi:10.4000/cybergeo.1035, 2001.

Best, M. J., Abramowitz, G., Johnson, H. R., Pitman, A. J., Balsamo, G., Boone, A., Cuntz, M., Decharme, B., Dirmeyer, P. A., Dong, J., Ek, M., Guo, Z., Haverd, V., van den Hurk, B. J. J., Nearing, G. S., Pak, B., Peters-Lidard, C., Santanello, J. A., Stevens, L. and Vuichard, N.: The Plumbing of Land Surface Models: Benchmarking Model Performance, J. Hydrometeorol., 16(3), 1425–1442, doi:10.1175/JHM-D-14-0158.1, 2015.

Beven, K. J. and Cloke, H. L.: Comment on "hyperresolution global land surface modeling: Meeting a grand challenge for monitoring Earth's terrestrial water" by Eric F. Wood et al., Water Resour. Res., 48(1), W01801, doi:10.1029/2010WR010090, 2012.

Beyene, T., Lettenmaier, D. P. and Kabat, P.: Hydrologic impacts of climate change on the Nile River Basin: Implications of the 2007 IPCC scenarios, Clim. Change, 100(3), 433–461, doi:10.1007/s10584-009-9693-0, 2010.

Blume, H.-P., Brümmer, G. W., Horn, R., Kandeler, E., Kögel-Knabner, I., Kretzschmar, R., Stahr, K. and Wilke, B.-M.: Lehrbuch der Bodenkunde, Springer-Verlag, Berlin Heidelberg, doi: 10.1007/978-3-662-49960-3., 2010.

Bohn, T. J. and Vivoni, E. R.: Process-based characterization of evapotranspiration sources over the North American monsoon region, Water Resour. Res., 52(1), 358–384, doi:10.1002/2015WR017934, 2016.

Breuer, L., Eckhardt, K. and Frede, H. G.: Plant parameter values for models in temperate climates, Ecol. Modell., 169(2–3), 237–293, doi:10.1016/S0304-3800(03)00274-6, 2003.

Chaney, N. W., Herman, J. D., Ek, M. B. and Wood, E. F.: Deriving global parameter estimates for the Noah land surface model using FLUXNET and machine learning, J. Geophys. Res., 121(22), 13,218-13,235, doi:10.1002/2016JD024821, 2016.

Clapp, R. B. and Hornberger, G. M.: Empirical equations for some soil hydraulic properties, Water Resour. Res., 14(4), 601–604, doi:10.1029/WR014i004p00601, 1978.

Cuntz, M., Mai, J., Samaniego, L., Clark, M., Wulfmeyer, V., Branch, O., Attinger, S. and Thober, S.: The impact of standard and hard-coded parameters on the hydrologic fluxes in the Noah-MP land surface model, J. Geophys. Res., 121(18), 10,676-10,700, doi:10.1002/2016JD025097, 2016.

Davin, E. L., Stöckli, R., Jaeger, E. B., Levis, S. and Seneviratne, S. I.: COSMO-CLM2: A new version of the COSMO-CLM model coupled to the Community Land Model, Clim. Dyn., 37(9–10), 1889–1907, doi:10.1007/s00382-011-1019-z, 2011.

Davy, R., Esau, I., Chernokulsky, A., Stephen, O. and Zilitinkevich, S.: Diurnal asymmetry to the observed global warming, Int. J. Climatol., 31(1), doi:10.1002/joc.4688, 2017.

Van Dijk, A. I. J. M., Gash, J. H., Van Gorsel, E., Blanken, P. D., Cescatti, A., Emmel, C., Gielen, B., Harman, I. N., Kiely, G., Merbold, L., Montagnani, L., Moors, E., Sottocornola, M., Varlagin, A., Williams, C. A. and Wohlfahrt, G.: Rainfall interception and the coupled surface water and energy balance, Agric. For. Meteorol., 214–215, 402–415, doi:10.1016/j.agrformet.2015.09.006, 2015.

Döll, P. and Fiedler, K.: Global-scale modeling of groundwater recharge, Hydrol. Earth Syst. Sci., 12(3), 863–885, doi:10.5194/hess-12-863-2008, 2008.

Döll, P., Trautmann, T., Gerten, D., Muller-Schmied, H., Ostberg, S., Saeed, F. and Schleussner, C.-

F.: Risks for the global freshwater system at 1.5 °C and 2 °C global warming, Environ. Res. Lett., 13(4), 23, doi:10.1088/1748-9326/aab792, 2018.

Fatichi, S. and Pappas, C.: Constrained variability of modeled T:ET ratio across biomes, Geophys. Res. Lett., 44(13), 6795–6803, doi:10.1002/2017GL074041, 2017.

Gerrits, A. M. J., Savenije, H. H. G., Veling, E. J. M. and Pfister, L.: Analytical derivation of the Budyko curve based on rainfall characteristics and a simple evaporation model, Water Resour. Res., 25(4), W04403, doi:10.1029/2008WR007308, 2009.

Gerten, D., Schaphoff, S., Haberlandt, U., Lucht, W. and Sitch, S.: Terrestrial vegetation and water balance - Hydrological evaluation of a dynamic global vegetation model, J. Hydrol., 286(1–4), 249–270, doi:10.1016/j.jhydrol.2003.09.029, 2004.

Gosling, S. N. and Arnell, N. W.: Simulating current global river runoff with a global hydrological model: Model revisions, validation, and sensitivity analysis, Hydrol. Process., 25(7), 1129–1145, doi:10.1002/hyp.7727, 2011.

Gosling, S. N. and Arnell, N. W.: A global assessment of the impact of climate change on water scarcity, Clim. Change, 134(3), 371–385, doi:10.1007/s10584-013-0853-x, 2016.

Gosling, S. N., Zaherpour, J., Mount, N. J., Hattermann, F. F., Dankers, R., Arheimer, B., Breuer, L., Ding, J., Haddeland, I., Kumar, R., Kundu, D., Liu, J., van Griensven, A., Veldkamp, T. I. E., Vetter, T., Wang, X. and Zhang, X.: A comparison of changes in river runoff from multiple global and catchment-scale hydrological models under global warming scenarios of 1 ??C, 2 ??C and 3 ??C, Clim. Change, 141(3), 577–595, doi:10.1007/s10584-016-1773-3, 2017.

De Groen, M. M. and Savenije, H. H. G.: A monthly interception equation based on the statistical characteristics of daily rainfall., 2006.

Guo, D., Westra, S. and Maier, H. R.: An R package for modelling actual, potential and reference evapotranspiration, Environ. Model. Softw., 78, 216–224, doi:10.1016/j.envsoft.2015.12.019, 2016.

Hartmann, A., Lange, J., Weiler, M., Arbel, Y. and Greenbaum, N.: A new approach to model the spatial and temporal variability of recharge to karst aquifers, Hydrol. Earth Syst. Sci., 16(7), 2219–2231, doi:10.5194/hess-16-2219-2012, 2012.

Hartmann, A., Wagener, T., Rimmer, A., Lange, J., Brielmann, H. and Weiler, M.: Testing the realism of model structures to identify karst system processes using water quality and quantity signatures, Water Resour. Res., 49(6), 3345–3358, doi:10.1002/wrcr.20229, 2013.

Hartmann, A., Gleeson, T., Rosolem, R., Pianosi, F., Wada, Y. and Wagener, T.: A large-scale simulation model to assess karstic groundwater recharge over Europe and the Mediterranean, Geosci. Model Dev., 8(6), 1729–1746, doi:10.5194/gmd-8-1729-2015, 2015.

Hartmann, A., Gleeson, T., Wada, Y. and Wagener, T.: Enhanced groundwater recharge rates and altered recharge sensitivity to climate variability through subsurface heterogeneity, Proc. Natl. Acad. Sci., 114(11), 2842–2847, doi:10.1073/pnas.1614941114, 2017.

Haughton, N., Abramowitz, G. and Pitman, A. J.: On the predictability of land surface fluxes from meteorological variables, Geosci. Model Dev., 11(1), 195–212, doi:10.5194/gmd-11-195-2018, 2018.

Hogue, T. S., Bastidas, L. A., Gupta, H. V. and Sorooshian, S.: Evaluating model performance and parameter behavior for varying levels of land surface model complexity, Water Resour. Res., 42(8), W08430, doi:10.1029/2005WR004440, 2006.

Hong, E.-M., Pachepsky, Y. A., Whelan, G. and Nicholson, T.: Simpler models in environmental studies and predictions, Crit. Rev. Environ. Sci. Technol., 47(18), 1669–1712,

doi:10.1080/10643389.2017.1393264, 2017.

Van den Hoof, C., Vidale, P. L., Verhoef, A. and Vincke, C.: Improved evaporative flux partitioning and carbon flux in the land surface model JULES: Impact on the simulation of land surface processes in temperate Europe, Agric. For. Meteorol., 181, 108–124, doi:10.1016/j.agrformet.2013.07.011, 2013.

IPCC: Climate Change 2013: The Physical Science Basis. Contribution of Working Group I to the Fifth Assessment Report of the Intergovernmental Panel on Climate Change, edited by V. B. and P. M. M. Stocker, T.F., D. Qin, G.-K. Plattner, M. Tignor, S.K. Allen, J. Boschung, A. Nauels, Y. Xia, Cambridge University Press., 2013.

IPCC: Climate Change 2014: Impacts, Adaptation, and Vulnerability. Contribution of Working Group II to the Fifth Assessment Report of the Intergovernmental Panel on Climate Change. Part B: Regional Aspects, edited by V. R. Barros, C. B. Field, D. J. Dokken, M. D. Mastrandrea, K. J. Mach, T. E. Bilir, E. S. Kissel, M. Chatterjee, K. L. Ebi, Y. O. Estrada, R. C. Genova, B. Girma, A. N. Levy, S. MacCracken, P. R. Mastrandrea, and L. L. White, Cambridge University Press., 2014.

Jarvis, P. G.: The Interpretation of the Variations in Leaf Water Potential and Stomatal Conductance Found in Canopies in the Field, Phil. Trans. R. Soc. Lond. B., 273(927), 593–610, doi:10.1098/rstb.1976.0035, 1976.

Kavetski, D. and Clark, M. P.: Numerical troubles in conceptual hydrology: Approximations, absurdities and impact on hypothesis testing, Hydrol. Process., 25(4), 661–670, doi:10.1002/hyp.7899, 2011.

Kelliher, F. M., Whitehead, D. and Pollock, D. S.: Rainfall interception by trees and slash in a young Pinus radiata D. Don stand, J. Hydrol., 131(1–4), 187–204, doi:10.1016/0022-1694(92)90217-J, 1992.

Kergoat, L.: A model for hydrological equilibrium of leaf area index on a global scale, J. Hydrol., 212–213, 268–286, doi:10.1016/S0022-1694(98)00211-X, 1998.

Körner, C.: Leaf diffusive conductances in the major vegetation types of the globe, in Ecophysiology of photosynthesis, edited by E. D. Schulze and M. M. Caldwell, pp. 463–490, Springer Study Edition, vol 100, Springer, Berlin, Heidelberg., 1995.

Kuppel, S., Peylin, P., Chevallier, F., Bacour, C., Maignan, F. and Richardson, A. D.: Constraining a global ecosystem model with multi-site eddy-covariance data, Biogeosciences, 9(10), 3757–3776, doi:10.5194/bg-9-3757-2012, 2012.

Lenhard, J. and Winsberg, E.: Holism, entrenchment, and the future of climate model pluralism, Stud. Hist. Philos. Sci. Part B - Stud. Hist. Philos. Mod. Phys., 41(3), 253–262, doi:10.1016/j.shpsb.2010.07.001, 2010.

Lewis, S. C. and Karoly, D. J.: Evaluation of historical diurnal temperature range trends in CMIP5 models, J. Clim., 26(22), 9077–9089, doi:10.1175/JCLI-D-13-00032.1, 2013.

Liang, X., Lettenmaier, D. P., Wood, E. F. and Burges, S. J.: A simple hydrologically based model of land surface water and energy fluxes for general circulation models, J. Geophys. Res., 99(D7), 14415–14428, doi:10.1029/94JD00483, 1994.

Lu, Y., Liu, S., Weng, L., Wang, L., Li, Z. and Xu, L.: Fractal analysis of cracking in a clayey soil under freeze-thaw cycles, Eng. Geol., 208, 93–99, doi:10.1016/j.enggeo.2016.04.023, 2016.

Medlyn, B. E., Robinson, A. P., Clement, R. and McMurtrie, R. E.: On the validation of models of forest CO2 exchange using eddy covariance data: some perils and pitfalls, Tree Physiol., 25, 839–857, doi:10.1093/treephys/25.7.839, 2005.

Mendoza, P. A., Clark, M. P., Barlage, M., Rajagopalan, B., Samaniego, L., Abramowitz, G. and Gupta, H.: Are we unnecessarily constraining the agility of complex process-based models?, Water Resour. Res., 51(1), 716–728, doi:10.1002/2014WR015820, 2015.

Miralles, D. G., Jiménez, C., Jung, M., Michel, D., Ershadi, A., McCabe, M. F., Hirschi, M., Martens, B., Dolman, A. J., Fisher, J. B., Mu, Q., Seneviratne, S. I., Wood, E. F. and Fernaìndez-Prieto, D.: The WACMOS-ET project - Part 2: Evaluation of global terrestrial evaporation data sets, Hydrol. Earth Syst. Sci., 20(2), 823–842, doi:doi.org/10.5194/hess-20-823-2016, 2016.

Mu, Q., Zhao, M. and Running, S. W.: Improvements to a MODIS global terrestrial evapotranspiration algorithm, Remote Sens. Environ., 115(8), 1781–1800, doi:10.1016/j.rse.2011.02.019, 2011.

Murray, S. J.: Trends in 20th century global rainfall interception as simulated by a dynamic global vegetation model: Implications for global water resources, Ecohydrology, 7(1), 102–114, doi:10.1002/eco.1325, 2014.

Muzylo, A., Llorens, P., Valente, F., Keizer, J. J., Domingo, F. and Gash, J. H. C.: A review of rainfall interception modelling, J. Hydrol., 370(1–4), 191–206, doi:10.1016/j.jhydrol.2009.02.058, 2009.

Pearce, A. J., Rowe, L. K. and Stewart, J. B.: Night time, wet canopy evaporation rates and the water balance of an evergreen mixed forest, Water Resour. Res., 16(5), 955–959, doi:10.1029/WR016i005p00955, 1980.

Pereira, L. S., Allen, R. G., Smith, M. and Raes, D.: Crop evapotranspiration estimation with FAO56: Past and future, Agric. Water Manag., 147, 4–20, doi:10.1016/j.agwat.2014.07.031, 2015.

Reusser, D. E. and Zehe, E.: Inferring model structural deficits by analyzing temporal dynamics of model performance and parameter sensitivity, Water Resour. Res., 47(7), W07550, doi:10.1029/2010WR009946, 2011.

Rosero, E., Yang, Z. L., Wagener, T., Gulden, L. E., Yatheendradas, S. and Niu, G.-Y.: Quantifying parameter sensitivity, interaction, and transferability in hydrologically enhanced versions of the Noah land surface model over transition zones during the warm season, J. Geophys. Res. Atmos., 115(D3), D03106, doi:10.1029/2009JD012035, 2010.

Rosolem, R., Gupta, H. V., Shuttleworth, W. J., Zeng, X. and De Gonçalves, L. G. G.: A fully multiple-criteria implementation of the Sobol' method for parameter sensitivity analysis, J. Geophys. Res., 117(D7), D07103, doi:10.1029/2011JD016355, 2012.

Running, S., Mu, Q. and Zhao, M.: MOD16A2 MODIS/Terra Net Evapotranspiration 8-Day L4 Global 500m SIN Grid V006, [Data set], , doi:10.5067/MODIS/MOD16A2.00, 2017.

Savenije, H. H. G.: Determination of evaporation from a catchment water balance at a monthly time scale, Hydrol. Earth Syst. Sci., 1(1), 93–100, doi:10.5194/hess-1-93-1997, 1997.

Scanlon, B. R., Keese, K. E., Flint, A. L., Flint, L. E., Gaye, C. B., Edmunds, W. M. and Simmers, I.: Global synthesis of groundwater recharge in semiarid and arid regions, Hydrol. Prpcesses, 20(15), 3335–3370, doi:doi.org/10.1002/hyp.6335, 2006.

Sellers, P. J., Randall, D. A., Collatz, G. J., Berry, J. A., Field, C. B., Dazlich, D. A., Zhang, C., Collelo, G. D. and Bounoua, L.: A revised land surface parameterization (SiB2) for atmospheric GCMs. Part I: Model formulation, J. Clim., 9(4), 676–705, doi:10.1175/1520-0442(1996)009<0676:ARLSPF>2.0.CO;2, 1996.

Shuttleworth, W. J.: Terrestrial Hydrometeorology, John Wiley & Sons, Ltd, Chichester, UK., 2012.

Sperna Weiland, F. C., Van Beek, L. P. H., Kwadijk, J. C. J. and Bierkens, M. F. P.: Global patterns of change in discharge regimes for 2100, Hydrol. Earth Syst. Sci., 16(4), 1047–1062, doi:10.5194/hess-16-1047-2012, 2012.

Stewart, J. B.: Modelling surface conductance of pine forest, Agric. For. Meteorol., 43(1), 19–35, doi:10.1016/0168-1923(88)90003-2, 1988.

Sugita, M. and Brutsaert, W.: Daily evaporation over a region from lower boundary layer profiles measured with radiosondes, Water Resour. Res., 27(5), 747–752, doi:10.1029/90WR02706, 1991.

Van Vuuren, D. P., Edmonds, J., Kainuma, M., Riahi, K., Thomson, A., Hibbard, K., Hurtt, G. C., Kram, T., Krey, V., Lamarque, J. F., Masui, T., Meinshausen, M., Nakicenovic, N., Smith, S. J. and Rose, S. K.: The representative concentration pathways: An overview, Clim. Change, 109, 5–31, doi:10.1007/s10584-011-0148-z, 2011.

Wada, Y., Van Beek, L. P. H. and Bierkens, M. F. P.: Nonsustainable groundwater sustaining irrigation: A global assessment, Water Resour. Res., 48(6), W00L06, doi:10.1029/2011WR010562, 2012.

Walter, I. A., Allen, R. G., Elliott, R., Itenfisu, D., Brown, P., Jensen, M. E., Mecham, B., Howell, T. A., Snyder, R., Eching, S., Spofford, T., Hattendorf, M., Martin, D., Cuenca, R. H. and WrightIvan, J. L.: The ASCE standardized reference evapotranspiration equation, ASCE-EWRI Task Committee Report, Environmental and Water Resources Institute of the American Society of Civil Engineers., 2005.

Warszawski, L., Frieler, K., Huber, V., Piontek, F., Serdeczny, O. and Schewe, J.: The Inter-Sectoral Impact Model Intercomparison Project (ISI–MIP): Project framework, Proc. Natl. Acad. Sci., 111(9), 3228–3232, doi:10.1073/pnas.1312330110, 2014.

Ye, H., Fetzer, E. J., Behrangi, A., Wong, S., Lambrigtsen, B. H., Wang, C. Y., Cohen, J. and Gamelin, B. L.: Increasing daily precipitation intensity associated with warmer air temperatures over northern Eurasia, J. Clim., 29(2), 623–636, doi:10.1175/JCLI-D-14-00771.1, 2016.

Young, P., Parkinson, S. and Lees, M.: Simplicity out of complexity in environmental modelling: Occam's razor revisited, J. Appl. Stat., 23(2–3), 165–210, doi:10.1080/02664769624206, 1996.

Zhao, Y., Ciais, P., Peylin, P., Viovy, N., Longdoz, B., Bonnefond, J. M., Rambal, S., Klumpp, K., Olioso, A., Cellier, P., Maignan, F., Eglin, T. and Calvet, J. C.: How errors on meteorological variables impact simulated ecosystem fluxes: A case study for six French sites, Biogeosciences, 9(7), 2537–2564, doi:10.5194/bg-9-2537-2012, 2012.

---

## Author Response (AR1)

Ms Fanny Sarrazin Research Associate, Water and Environmental Engineering Department of Civil Engineering, University of Bristol Queen's Building, University Walk, Bristol BS81TR, UK fanny.sarrazin@bristol.ac.uk

Dear Dr Roche,

Please find enclosed a revised version of our manuscript "V2Karst V1.0: A parsimonious large-scale integrated vegetation-recharge model to simulate the impact of climate and land cover change in karst regions" in which we address the Reviewers' comments. The Reviewers' detailed comments helped us to significantly improve our manuscript, although we have found it difficult to address some of Reviewer #2's specific comments, because he/she did not provide any reference to support his/her statements. We attached a detailed reply to the Reviewers' comments and a marked-up version of the manuscript that highlights in red the changes we made. Since we substantially revised our manuscript, we did not keep track of minor editorial changes to facilitate readability of the marked-up manuscript. Specifically, we wish to report some key changes:

- To address the concern of Reviewer #2 regarding the suitability of using a daily simulation time step to run V2Karst, we performed additional simulations at hourly time step (reported in our Supplementary material). Our results suggest that there are no significant differences between V2Karst predictions for daily and hourly time steps for our study purpose, i.e. simulation of monthly and annual recharge.
- We clarified our rationale for development of our parsimonious large-scale hydrological model (Sect. 2.1). We followed an approach to large-scale hydrological modelling that has been widely used for climate change impact studies. We aimed to limit model complexity given the limited amount of information available at large-scales to constrain model simulations.
- Following the advice of Reviewer #1, we made an effort to shorten the manuscript and improve its readability, while also addressing the Reviewers' comments.
- We rewrote the discussion section to better convey the meaning and implications of our sensitivity analyses results and to discuss large-scale applications of V2Karst (Sect. 6).

Again, we wish to thank you and the Reviewers' for time and effort taken to review our manuscript and for any further comment and suggestion.

Sincerely,

Fanny Sarrazin

FSauazin

**Response to Reviewer #1**

**COMMENT 1:** The manuscript presents a modified version of the large-scale karst recharge model VarKarst. The here presented model (V2Karst V1.0) replaces the simplified evapotranspiration (ET) component (empirical Priestley-Taylor equation) by the physical based Penman-Monteith equation (for potential evapotranspiration). The authors also include a separate calculation of the different evaporation processes in order to use the model for climate and land cover change impact studies. The model extension increases the number of parameters. The general functioning as well as the influence of the new parameters are tested by applying the new model to four study sides, different in climate and vegetation. The manuscript is a novel extension of previous work published by the research group. The conceptual description and the numerical adaptation of the processes are sound. The results of the model application on the four test sides prove the general functioning of the new model. However, the manuscript also has weak points, which are mainly related to the presentation of the method and the results. The manuscript can easily be shortened by 10-20% without losing important information. The presentation of the results needs to be improved, especially since it is difficult to distinguish between observed values and modeled results. My detailed comments are listed below.

**REPLY 1:** We wish to thank the Reviewer for appreciating the novelty and soundness of our work and for the detailed comments, especially regarding the karst aspect of our work. **Line numbers in our replies refer to the marked-up version of our revised manuscript** (included in this file). In the marked-up version of the manuscript we highlighted in red the insertion and deletion we made. We took into account the Reviewers' concerns regarding the length of the manuscript and the presentation of the methods and the results as explained in the following.

**1.** Shortening the manuscript**

We realise that the manuscript was rather long in parts. We made an effort to shorten our revised version and to improve its readability and in particular:

- We followed the suggestions of the Reviewer (COMMENT 6). We revised the introductions of the sections to make them more informative (Sect. 4 p19 L10-19) and deleted them where appropriate (Sect. 2 p5 L29-30, in Sect. 2 p8 L3-5, Sect. 5 p23 L21-23). We deleted wordy expressions such as 'Regarding the data processing' p18 L9, 'using numerical models' p22 L6 and 'the sensitivity indices analysed in this study' that we replaced by 'we analyse' p21 L15. We rewrote wordy sentences such as p26 L27-28.
- In Sect. 1, we rewrote the paragraph that describes the objectives of the study p4 L31-p5 L12 more concisely and deleted unnecessary repetitions. We also shorten the sentence p5 L13 by deleting 'that help us to overcome the previous limitation'.
- In Sect. 2.1 we removed the expression p6 L5-6 'and uncertainty in large-scale weather forcing variables [...]', which is not further discussed in our manuscript. We simplified the paragraph p6 L31-p7 L3. We removed the sentence p7 L13-14 which provides unnecessary details on a previous karst model.
- In Sect. 3.2, we deleted the reference to the 'residual estimate' of ET p18 L29-p19 L8 and we now only discuss the 'bowen ratio estimate' p18 L15-18 that we actually used in our analyses. Further analyses of the uncertainty in ET observations using the 'residual estimate' are reported in Sect. S4 of our Supplementary material as indicated p18 L19-21.
- In Sect. 5, we deleted more specifically the sentences p24 L24-25 and p27 L6-8 that pertained to Sect. 6 (discussion section). We moved the sentence p26 L15-16 into Sect. S7 of our Supplementary material, since it is a detailed comment on additional sensitivity analysis results that are reported in Sect. S7.

- We rewrote Sect. 6 more concisely, while also clarifying the meaning and implications of our sensitivity analyses and addressing other comments of the Reviewers (in particular COMMENT 3 and 5 below).

**2. Presenting the methods and the results**

We clarified the presentation of the methods and the results and specifically:

- We revised the introduction to the method section (Sect. 4) to make it more informative and to clarify the distinction between observed and simulated values (p19 L10-19). In brief, the parameter estimation approach (Sect. 4.1) uses measurements of weather variables to force the model and measurements of model output to define the soft rules. The global sensitivity analysis (Sect 4.2) uses measurements of weather variables to force the model, but no measurements of model output. Finally, the virtual experiments (Sect 4.3) do not use any measurements but synthetic data to force the model.
- We rewrote parts of Sect. 4.3 to clarify the set-up of the virtual experiments p23 L4-18.
- We prepared a new version of Fig. 4 in which we corrected some plotting issues (as mentioned in COMMENT 13), which may bring some clarity to the meaning of this figure.
- We simplified Fig. 5 which may have brought some confusion regarding the distinction between observed and simulated values. We now only report the bowen ratio estimate of observed ET of Eq. (9) p18 and we removed the lines that corresponded to the other estimates of observed ET (estimate without correction and with residual correction) that are now reported in Fig. S2 in our Supplementary material. We clarified in the caption of Fig. 5 and p24 L26-28 the variables that are observed and simulated.
- We also made numerous edits to the results section (Sect. 5) to improve its readability, for instance in the presentation of the results of the virtual experiments p28 L15-27.

**MAIN COMMENTS**

**COMMENT 2:** The purpose of V2Karst V1.0 is to predict recharge in karst regions. The authors mention that "a large part of the groundwater recharge occurs as concentrated and fast flow in large apertures and the other part as diffuse and slow flow in the matrix (Hartmann and Baker, 2017)." Especially concentrated recharge, e.g. fast infiltration into sinkholes, can be considered as a short-term process and is entirely uncoupled from soil and/or vegetation properties (overland flow -> percolation). I assume that your model, calculating the water balance, underestimates the recharge in karst regions dominated by concentrated recharge. Do you think your model is able to equally represent both recharge processes?

**REPLY 2:** We agree with the need for representing concentrated recharge processes. In karst systems, infiltration and recharge can be slow and diffuse in the matrix and fast and concentrated in large conduits or fissures that act as preferential flow pathways. Lateral flow at the surface and in the epikarst is an important mechanism that concentrates the infiltrating water into the preferential flow pathways (Jeannin and Grasso, 1997; Williams, 1983, 2008). In particular, the epikarst plays a role of temporary storage that can redistribute fast and concentrated recharge (Williams, 1983, 2008). Figure R1.c below provides a conceptual model of the soil and epikarst processes of a real karst system.

V2karst's representation of concentrated and diffuse infiltration and recharge, which is the same as in the VarKarst model (Hartmann et al., 2015), follows this conceptual model (Figure R1.a). V2karst represents the spatial variability of subsurface properties observed in karst systems by dividing each model simulation unit into a number of vertical compartments that have different soil and epikarst properties. Parameters for each model compartment (soil and epikarst storage capacities and epikarst outflow coefficient) are estimated using a distribution function. The daily water balance is explicitly

evaluated for each model vertical compartment. Additionally, when a given compartment saturates (both soil and epikarst stores), its saturation excess generates a surface lateral flow to the next unsaturated compartments that have higher storage capacities and higher permeabilities. In this representation, surface and subsurface lateral flow are thus lumped together. Conceptually, in V2Karst, the direct contribution of precipitation to infiltration and recharge can be associated with diffuse infiltration and recharge, while the contribution of lateral flow can be associated with concentrated infiltration and recharge. Hence, the V2karst structure allows to account for the interplay between diffuse and concentrated infiltration and recharge processes.

The representation of karst processes in V2Karst and VarKarst is based on a previous karst model developed for applications at the local scale introduced in Hartmann et al. (2012). The structure of this previous models explicitly represents lateral flow both at the surface and in the epikarst, in agreement with understanding of the flow mechanisms in the epikarst (e.g. Williams, 1983, 2008). It was tested using hydrodynamic and hydrochemical information at stalactite drips in a karstic cave in Hartmann et al. (2012). However, simplifications have been introduced for applications at the large-scale, given the limited information available to constrain the additional parameters required in the previous karst model to represent lateral flow in the epikarst.

We are aware of the fact that, in V2Karst's and VarKarst's representation, concentrated recharge is not entirely uncoupled from the soil and vegetation properties as it is observed in real karst systems. However, a previous study by Hartmann et al. (2017) compared simulated recharge with VarKarst and independent estimates of recharge and showed that there was no systematic bias in the simulations (Figure R2 below). Moreover, the study by Hartmann et al. (2017) showed that recharge values simulated with VarKarst were significantly higher than recharge values simulated with models that do not include karst processes (Figure R2 below).

As also stated in REPLY 9, in the revised version of our manuscript, we briefly explained how diffuse and concentrated recharge is represented in VarKarst and V2karst in Sect. 2.2. p8 L12-17.

**Figure R1**. (a) Schematic description of the VarKarst model for one model grid cell including the soil (yellow) and epikarst storages (grey) and the simulated fluxes, (b) its gridded discretization over karst regions and (c) the subsurface heterogeneity that its structure represents for each grid cell. Figure taken from Hartmann et al. (2015, Figure 1)

Figure R2. Comparison of simulated and observed recharge. In blue are reported values simulated with the VarKarst model (heterogeneous representation), in yellow values simulated with the PCR-GLOBWB model (homogeneous representation, i.e. absence of karst processes in the model representation), and in green the Varkarst model with subsurface heterogeneity processes turned off. Whiskers indicate the simulation uncertainty (1 SD) for simulations with VarKarst. No statistical difference (5% significance level) was found between simulations with Varkarst and the observations. mm/a, millimeters per year.

Figure taken from Hartmann et al. (2017, Figure 2)

**COMMENT 3:** *I* am aware of the fact that the manuscript is focused on the implementation and the testing of the new evapotranspiration component. Since soil layers in karst regions can be thin or even totally absence the authors should consider this fact in the interpretation of the results.

**REPLY 3:** Large differences in soil depth can indeed be observed across different karst landscapes. To apply the VarKarst model over Europe and the Mediterranean, Hartmann et al. (2015) have identified four main karst landscapes with different soil depths and degrees of karstification based on climate and topography (Humid, Mountain, Mediterranean and Desert landscapes). Different values of the parameter  $V_{soi}$  (mean soil water capacity) have been applied in the different landscapes. In particular, very small values of  $V_{soi}$  have been used in arid areas (i.e. desert landscape in Hartmann et al. (2015)),where soils tend to be very thin.

We are also aware of the fact that soils may be absent in some karst areas (e.g. karren field in high mountain areas, see e.g. Hartmann et al. (2014a)), and that these areas may consequently produce very high recharge amounts. The model does not account for the fact that the soil may be absent and always includes a soil layer, although the soil layer can be very thin and therefore can have a limited impact on recharge. This assumption seems reasonable given the large extent of the simulation units for large-scale applications ( $0.25^{\circ}x0.25^{\circ}$  or  $0.5^{\circ}x0.5^{\circ}$  cell in previous applications). In fact, we can assume that soil layers can always be found in such large simulation units. However, for model applications at high resolutions, we recognise the fact that an explicit consideration of bare rock regions should be included in the model.

We added a brief discussion of this point in Sect. 6.3 of our revised manuscript p33 L6-9.

**COMMENT 4:** The manuscript lacks a description/characterization of the four karst regions (e.g. by describing dominant karst features or by the interpretation of spring hydrographs).

**REPLY 4:** In this study, we focus on groundwater recharge, which is a key component of the water balance as we clarified in our revised manuscript p5 L17-19. Recharge characterises the amount of renewable groundwater, and therefore the amount of groundwater available to human consumption and ecosystems (e.g. Scanlon et al., 2006; Döll and Fiedler, 2008; Wada et al., 2012). We do not model groundwater flow and storage nor spring discharge. Therefore, to test the model, we focus on datasets that can be related to the fluxes and states simulated by V2Karst, i.e. soil moisture and evapotranspiration as in Hartmann et al. (2015). No datasets providing time series of recharge are available. We do not use spring hydrographs, because spring discharge depends on groundwater routing and is not commensurate with groundwater recharge. Spring discharge measurements were used to extensively test a previous versions of the VarKarst model including groundwater storage and routing to the spring, at different locations in Europe and the Mediterranean (Hartmann et al., 2013a, 2013b, 2014b).

In the manuscript, we briefly describe the sites in Sect. 3.1 p17. Table B1 describes the four sites in detail, and more specifically the soil depth and bedrock. We have realised that we have exchanged the name of the two French sites in Table B1 (Puéchabon is actually the French 2 site and Font-Blanche the French 1 site). We corrected this mistake in the revised version of the manuscript. We did not add more details on the sites in the main text to limit the length of the manuscript.

**COMMENT 5:** In general, a differentiation between different karst systems and therefore the wide variety of**

**REPLY 5**: A large variability in hydraulic properties and in recharge patterns can indeed be observed across karst systems (e.g. Klimchouk and Ford, 2000; Hartmann et al., 2014a). For applications over large-scale domain, Hartmann et al. (2015) identified four typical karst landscapes with different hydraulic properties, based on climate and topography (as mentioned in REPLY 3). This simplified classification of the simulation domain in four landscapes was introduced to enable large-scale applications of the model.

In Sect. 6.3 (discussion of V2karst large-scale application), we better highlighted the fact that future large-scale applications of V2Karst will need to account for the variability observed across karst systems, for instance using a simplified classification as in Hartmann et al. (2015) p33 L1-4.

**COMMENT 6:** As already mentioned, the current manuscript is too long and needs to be shortened: **1**) (Almost) every section starts with a short introduction on the section. Most of them are redundant.**

2) The authors use wordy descriptions instead of clear words for describing their work. Is a "virtual experiment with synthetic data to assess the sensitivity" (Page 1, Line 19) not simply a "sensitivity analysis"?
3) Discussion chapter: Consists of sentences/paragraphs, which can be defined as general knowledge (e.g. Page 26. Line 16; Page 26, Line 24) or which should be familiar by the readers of the journal (e.g. Page 28, Line 12).

**REPLY 6:** We thank the Reviewer for suggesting how to shorten the manuscript and we reply below to the three points raised by the Reviewer.

1) As already stated in REPLY 1, we revised the introductions of the sections to make them more informative (Sect. 4 p19 L10-19) and deleted them where appropriate (Sect. 2 p5 L29-30, in Sect. 2 p8 L3-5, Sect. 5 p23 L21-23).

**2)** The virtual experiments are indeed sensitivity analyses. However, an important aspect is that we used synthetic data, which has been done only in few studies (e.g. the studies reported p22 L12-15). The reason for using synthetic data is that it allows to explore conditions beyond what was historically observed, and it is therefore useful to better understand potential impact of changes in climate. It also permits unequivocal attribution of changes in model output to changes in model inputs. Therefore, we think that it is important to mention that we used synthetic data in the abstract p1 L23. However, as discussed in REPLY 7, we made substantial changes to the abstract for clarification and readability, including the discussion of the virtual experiments.

Furthermore, as stated in REPLY 1, we deleted wordy expressions in the manuscript, such as 'Regarding the data processing' p18 L9, 'using numerical models' p22 L6 and 'the sensitivity indices analysed in this study' that we replaced by 'we analyse' p21 L15. We rewrote wordy sentences such as p26 L27-28.

**3**) We agree that the discussion section can be shortened and clarified and more specifically at the places indicated by the Reviewer. We rewrote Sect. 6 more concisely, while also clarifying the meaning and implications of our sensitivity analyses in Sect. 6.1 and 6.2 p29-32 and addressing other comments of the Reviewers (including COMMENT 3 and 5 above).

We clarified the discussion of the virtual experiments (Sect. 6.2 p31-32). Firstly, the virtual experiments confirm that the model behaves reasonably, since it shows sensitivities to both precipitation (overall amount and temporal distribution) and land cover, in agreement with previous studies and general understanding. Secondly, the virtual experiments allow to unequivocally and quantitatively characterise the relationship between simulated recharge and both precipitation (overall mean and temporal distribution) and land cover. Therefore, virtual experiments are a complementary approach to model applications using site-specific data to assess the impact of climate and land cover changes on simulated recharge.

**SECONDARY COMMENTS**

**COMMENT 7:** - Page 1, Line 21: ". . . and they suggest that simulated recharge is sensitive to both precipitation (overall amount and temporal distribution) and land cover." Is this one of the main results of your work and is it really a new finding?

**REPLY 7:** We agree that we need to reformulate this sentence. We refer to REPLY 6 for a clarification of the objectives of the virtual experiments.

We rewrote this part of the abstract (see p1 L23-27 in the revised version of the manuscript).

**COMMENT 8:**

- Page 2, Line 30: The sentence is difficult to understand.

- Page 4, Line 17: Please, rephrase the long sentence (and consider deleting the first part of the sentence).

**REPLY 8:** We deleted the fist sentence mentioned by the Reviewer p3 L29-32. The split in two the second sentence mentioned by the Reviewer (p6 L2-7). We also removed the expression p6 L5-6 'and uncertainty in large-scale weather forcing variables [...]', which is not further discussed in our manuscript.

**COMMENT 9:** Page 6, Line 12: Could you please add a bit more information on how diffuse and concentrated recharge is considered by the model.

**REPLY 9:** We refer to REPLY 2 for a detailed explanation of the conceptualisation of recharge processes in V2Karst. In the revised version of our manuscript, we briefly explained how diffuse and concentrated recharge is represented in VarKarst and V2karst in Sect. 2.2. p8 L12-17.

**COMMENT 10:** *Page 12, Line 12/13/14: Please, consider using SI-Units.*

**REPLY 10:** We have adopted these units because they are typically used in the evapotranspiration literature (e.g. Allen et al., 1998; Shuttleworth, 1993, 2012). The revised version of the manuscript includes the unit of net radiation (Rn [MJ m-2 d-1]) p15 L10 which was missing.

**COMMENT 11:** Page 13 Seasonality of vegetation: Are you using the same seasonality on every study site irrespective of the local climate and vegetation type?

**REPLY 11**: Typically, in hydrological models, vegetation seasonality is represented using different schemes with different levels of complexity or is neglected completely (Table A3). We chose to implement a simple representation (piecewise linear function). In this way, we can test the impact of vegetation seasonality by assessing the sensitivity of recharge to the seasonality parameter LAImin. We applied the same function at all sites, but we used different values of the seasonality parameter LAImin as indicated in Table 3. The timings of the four phases reported in Eq. (16) are appropriate for study sites that are located in the northern hemisphere and that have natural vegetation.

We added a sentence p16 L18-19 to clarify the fact that the timings of the four phases of the seasonality model should be adapted to the application domain. We also added 'in this study' p16 L11 and L12 to clarify the fact that the choice of growing and dormant season is appropriate for application at the sites used in this study.

**COMMENT 12:**

- Page 15, Line 4: Please consider splitting the sentence.

- Equation 17/18: Please remove the units from the equation and mention both parameters in the text, e.g. "1. Eact, bow [mmmonth1], a corrected value that assumes that latent heat ( $\delta$  'IR $\pm \delta$  ' 'IR $\pm$  [MJ.m ' -2.month-1]) and sensible heat ( $\delta$  'IR $\pm z$  [MJ.m ' -2.month-1]) have similar errors (referred to as Bowen ratio estimate): - Page 17, Line 29: Please, rephrase the sentence.

- Page 24, Line 4: Please, rephrase the sentence.

**REPLY 12:** We have applied these changes in the revised manuscript:

- p18 L2-4
- p18 L18
- p21 L5-7
- p27 L29-p28 L2

**COMMENT 13:** Figure 4: The Figure presents the results in a confusing way and some of the values exceed the constrained parameter ranges according to Table 3.

**REPLY 13**: We realise that the plot we used -a parallel coordinate plot -is not familiar to all readers in earth sciences. A parallel coordinate plot is a two-dimensional plot that allows to visualise a

multidimensional space (here the space of the model parameters and outputs). In Figure 4, each line represents a combination of model parameters values (normalised) and the corresponding model output values (normalised). Parallel coordinate plots are increasingly used (e.g. Inselberg, 2009; Kasprzyk et al., 2013; Pianosi et al., 2017), and are implemented for instance in the Matlab Statistics and Machine Learning Toolbox (function "parallelcoords") and in the SAFE toolbox for sensitivity analysis we utilised in our paper (Pianosi et al., 2015).

Initially, we sampled the parameter space within wide ranges (Table 2), and we applied a priori information on parameter ranges (Table 3) only in rule 5. Therefore, prior to application of rule 5, parameter values can exceed the ranges of Table 3 (yellow, light blue, green and dark blue lines in Figure 4). Instead, posterior to application of rule 5, all parameter values should be within the ranges of Table 3 (red lined in Figure 4).

The Reviewer may be referring to the fact that some red lines slightly exceed the black vertical lines in Figure 4. This is a plotting issue and we corrected it in the revised version of the manuscript.

**MINOR COMMENTS AND TYPOGRAPHICAL ERRORS**

**COMMENT 14**: Please, use a consistent citation style.

**REPLY 14**: We corrected the style of the reference p6 L7.

**COMMENT 15**: Please, use a consistent style for figure references. Two different versions exist: Fig. and Figure.

**REPLY 15:** We replaced 'Figure' by 'Fig.' p6 L22, p8 L26, p17 L11 so that the referencing of the figures is consistent with guidelines of GMD available at (https://www.geoscientific-model-development.net/for\_authors/manuscript\_preparation.html): *The abbreviation "Fig." should be used when it appears in running text and should be followed by a number unless it comes at the beginning of a sentence, e.g.: "The results are depicted in Fig. 5. Figure 9 reveals that..."*.

**COMMENT 16:**

- Units -> Replace the dots by multiplication sign or even better delete them.
- Page 2, Line 2: . . .world. . .
- Page 2, Line 2: For instance, ...
- Page 5, Line 4: ... (Hartmann et al., 2015). This ... (space missing)
- Page 5, Line 33: . . . to represent. . .
- Page 7, Line 13: ... the following formulas ...
- Page 12, Line 13: ... is the psychrometric constant, ...
- Page 15, Line 18: ... data processing are reported ...
- Page 20, Line 18: red lines -> the a priori information are indicated by black lines in Figure 4!
- Page 29, Line 14: We, therefore, ...
- Figure 5, Line 4: . . . percentage of Eact . . .
- Figure 6: Line 5: Blue . . .
- Figure 9: Line 4: remove the open bracket

**REPLY 16:** The revised version of the manuscript will include these corrections. We chose to add 'More importantly' p2 L2.

**References to support our reply to Reviewer #1**

Allen, R. G., Pereira, L. S., Raes, D. and Smith, M.: Crop evapotranspiration: Guidelines for computing crop requirements, FAO Irrigation and Drainage Paper 56, Food and Agriculture Organization (FAO), Rome, Italy., 1998.

Döll, P. and Fiedler, K.: Global-scale modeling of groundwater recharge, Hydrol. Earth Syst. Sci., 12(3), 863–885, doi:10.5194/hess-12-863-2008, 2008.

Hartmann, A., Lange, J., Weiler, M., Arbel, Y. and Greenbaum, N.: A new approach to model the spatial and temporal variability of recharge to karst aquifers, Hydrol. Earth Syst. Sci., 16(7), 2219–2231, doi:10.5194/hess-16-2219-2012, 2012.

Hartmann, A., Weiler, M., Wagener, T., Lange, J., Kralik, M., Humer, F., Mizyed, N., Rimmer, A., Barberá, J. A., Andreo, B., Butscher, C. and Huggenberger, P.: Process-based karst modelling to relate hydrodynamic and hydrochemical characteristics to system properties, Hydrol. Earth Syst. Sci., 17(8), 3505–3521, doi:10.5194/hess-17-3305-2013, 2013a.

Hartmann, A., Barberá, J. A., Lange, J., Andreo, B. and Weiler, M.: Progress in the hydrologic simulation of time variant recharge areas of karst systems - Exemplified at a karst spring in Southern Spain, Adv. Water Resour., 54, 149–160, doi:10.1016/j.advwatres.2013.01.010, 2013b.

Hartmann, A., Goldscheider, N., Wagener, T., Lange, J. and Weiler, M.: Karst water resources in a changing world: Review of hydrological modeling approaches, Rev. Geophys., 52(3), 218–242, doi:10.1002/2013RG000443, 2014a.

Hartmann, A., Mudarra, M., Andreo, B., Marin, A., Wagener, T. and Lange, J.: Modelling spatiotemporal impacts of hydroclimatic extremes on groundwater recharge at a Mediterranean karst aquifer, Water Resour. Res., 50(8), 6507–6521, doi:10.1002/2014WR015685, 2014b.

Hartmann, A., Gleeson, T., Rosolem, R., Pianosi, F., Wada, Y. and Wagener, T.: A large-scale simulation model to assess karstic groundwater recharge over Europe and the Mediterranean, Geosci. Model Dev., 8(6), 1729–1746, doi:10.5194/gmd-8-1729-2015, 2015.

Hartmann, A., Gleeson, T., Wada, Y. and Wagener, T.: Enhanced groundwater recharge rates and altered recharge sensitivity to climate variability through subsurface heterogeneity, Proc. Natl. Acad. Sci., 114(11), 2842–2847, doi:10.1073/pnas.1614941114, 2017.

Inselberg, A.: Parallel coordinates: Visual multidimensional geometry and its applications, Springer, New York., 2009.

Jeannin, P.-Y. and Grasso, D. A.: Permeability and hydrodynamic behavior of karstic environment, in Karst Waters Environmental Impact, edited by G. Gunay and A. I. Johnson, pp. 335–342, A.A. Balkema, Rotterdam, Netherlands., 1997.

Kasprzyk, J. R., Nataraj, S., Reed, P. M. and Lempert, R. J.: Many objective robust decision making for complex environmental systems undergoing change, Environ. Model. Softw., 42, 55–71, doi:10.1016/j.envsoft.2012.12.007, 2013.

Klimchouk, A. B. and Ford, D. C.: Types of karst and evolution of hydrogeologic setting, in Speleogenesis. Evolution of Karst Aquifers, edited by A. B. Klimchouk, D. C. Ford, A. Palmer, and W. Dreybrodt, pp. 45–53, National Speleological Society, Huntsville, Alabama, U.S.A., 2000.

Pianosi, F., Sarrazin, F. and Wagener, T.: A Matlab toolbox for Global Sensitivity Analysis, Environ. Model. Softw., 70, 80–85, doi:10.1016/j.envsoft.2015.04.009, 2015.

Pianosi, F., Iwema, J., Rosolem, R. and Wagener, T.: A multimethod Global Sensitivity Analysis approach to support the calibration and evaluation of Land Surface Models, in Sensitivity Analysis in Earth Observation Modelling, edited by G. Petropoulos and P. Srivastava, pp. 125–144, Elsevier Inc., 2017.

Scanlon, B. R., Keese, K. E., Flint, A. L., Flint, L. E., Gaye, C. B., Edmunds, W. M. and Simmers, I.: Global synthesis of groundwater recharge in semiarid and arid regions, Hydrol. Prpcesses, 20(15), 3335–3370, doi:doi.org/10.1002/hyp.6335, 2006.

Shuttleworth, W. J.: Evapotranspiration, in Handbook of Hydrology, edited by D. R. Maidment, p. 4.1-4.53, McGraw-Hill inc., New York., 1993.

Shuttleworth, W. J.: Terrestrial Hydrometeorology, John Wiley & Sons, Ltd, Chichester, UK., 2012.

Wada, Y., Van Beek, L. P. H. and Bierkens, M. F. P.: Nonsustainable groundwater sustaining irrigation: A global assessment, Water Resour. Res., 48(6), W00L06, doi:10.1029/2011WR010562, 2012.

Williams, P. W.: The role of the subcutaneous zone in karst hydrology, J. Hydrol., 61(1-3), 45-67, doi:10.1016/0022-1694(83)90234-2, 1983.

Williams, P. W.: The role of the epikarst in karst and cave hydrogeology : a review, Int. J. Speleol., 37, 1–10, doi:10.5038/1827-806X.37.1.1, 2008.

**Response to Reviewer #2**

**OVERALL REPLY:** We thank the Reviewer for the detailed review and take the opportunity to clarify some important points of our approach, while responding to the criticisms made. We provide detailed responses to specific comments below, but we thought it helpful to summarise the key point here: our manuscript introduces a new evapotranspiration component into a previously developed large-scale hydrological model for karst areas (Hartmann et al., 2015, GMD). We are therefore following an approach to large-scale hydrological modelling that has been widely used for climate change impact studies (e.g. Beyene et al., 2010; Sperna Weiland et al., 2012; Gosling et al., 2017). The Reviewer focuses on the use of land surface models to the same issue, which is an alternative approach currently taken to simulate climate and land cover change impacts on hydrological worldels. However, we are not attempting to build a land surface model here, but rather to advance a hydrological model for large-scale applications. Indeed, we ourselves have argued in the past that these two communities should come closer together and learn from each other (Archfield et al., 2015, WRR).

We provide below a point-by-point reply to the comments of the Reviewer with additional references to previous studies that help us support our modelling choices (see list of references at the end of this report). Line numbers in our replies refer to the marked-up version of our revised manuscript (included in this file). In the marked-up version of the manuscript we highlighted in red the insertion and deletion we made. We found it difficult to address some of the Reviewers' specific comments, because he/she did not provide any reference to support his/her statements.

In particular, the revised version of the manuscript includes a comparison between hourly and daily simulations of the V2Karst model as discussed in REPLY 1 and 3 below.

**COMMENT 1**: The paper proposed by Sarrazin et al. aims at adding a new evaporation formulation to the recharge model VarKarst which specialises on the hydrology of karst systems. The aim of this development is to make the model suitable for exploring the impact of climate and land surface changes on these very sensitive hydrological structures. The main themes of these improvements are to be applicable at the large scale and to be parsimonious.

I believe this model fails on both accounts for a simple reason, the authors have neglected the fact that evaporation is strongly controlled by the diurnal cycle of radiation and atmospheric processes. One of the main consequences of climate change is to modify the diurnal cycle at the surface and in the atmosphere. Thus the application of V2Karst to climate change is bound to produce unrealistic sensitivities. The model would be more parsimonious and more robust (because based on stronger physical grounds) if it would explicitly represent the diurnal cycle.

**REPLY 1**: Our model is in line with widely published approaches to climate change assessment using large-scale hydrological models (e.g. Beyene et al., 2010; Sperna Weiland et al., 2012; Gosling et al., 2017), that all neglect the diurnal cycle and apply hydrological models at a daily time step. We provide further details on this issue in the reply to the reviewer's COMMENT 3 (Penman-Monteith equation) and 9 (canopy interception). We agree that there is indeed a strong need for better comparison studies to understand how neglected processes (e.g. of diurnal cycles) affect climate change impact studies, but this is beyond the scope of the study presented here.

Moreover, the reviewer wrote "One of the main consequences of climate change is to modify the diurnal cycle". We agree that global average radiative forcing is projected to increase (IPCC, 2013; Van Vuuren et al., 2011), and average land surface air temperature is documented to have already increased globally (IPCC, 2013, Chapter 2 p187-188 for a global assessment, 2014, Chapter 23 p1275-1276 for an assessment for Europe) and is projected to further increase (IPCC, 2013, Chapter 12 p1062-1064 for a global assessment, 2014, Chapter 23 p1276 for an assessment for Europe). However, to our knowledge,

changes in the diurnal cycle appear to be much more uncertain. Changes in temperature are more documented, possibly because the historical temperature record is more accurate than the other climate variables (Allen and Ingram, 2002). We are aware of multiple studies that analysed the past changes in diurnal temperature range (i.e. difference between minimum and maximum daily temperature). A summary of these studies is provided in the Fifth Assessment Report of the Intergovernmental Panel on Climate Change (IPCC, 2013). They overall indicate that decreases in diurnal temperature range have been observed, but these decreases were found to be smaller than changes in average temperature (IPCC, 2013, Chapter 2, p188). Moreover, some studies suggest that these apparent changes in diurnal temperature range may be attributed to non-climatic factors (IPCC, 2013, Chapter 2, p188). In this regard, a more recent study points out that the drivers of the past observed changes in diurnal temperature ranges are still not well understood (Davy et al., 2017). Additionally, it has been shown that climate models involved in the Coupled Model Intercomparison Project Phase 5 (CMIP5, scientific basis for the IPCC fifth Assessment Report) are reproducing poorly the past observed changes in diurnal temperature range (Lewis and Karoly, 2013), which suggests that future projections in diurnal temperature range have large uncertainties.

**Changes in the revised version of the manuscript**

To prepare the revised version of our manuscript, we have performed additional simulations of V2Karst using an hourly time step. We compare the results of the parameter estimation strategy (described in Sect. 4.1) obtained for daily and hourly simulation. We briefly discuss the results in the manuscript p9 L15-19, while detailed results are reported in Sect. S8 of our Supplementary material. We did not find significant differences between daily and hourly simulations for assessing recharge at monthly and annual time scale, which is the focus of our study. This demonstrates that it is reasonable to apply V2Karst at daily time step for our application.

To be able to run the model at hourly time step, we modified the canopy interception routine of V2Karst. We introduced a state variable that represents the interception storage so that the model accounts for the carry-over of interception storage from one time step to the next for hourly time step. The revised interception routine is described p12-13 in the revised manuscript. We have also added the ground heat flux in the description of the Penman-Monteith equation (Eq.(14) p15). For daily simulations, ground heat flux is neglected, while it is accounted for hourly simulations as explained p15 L15-17 and in Sect. S8 of our Supplementary material.

**COMMENT 2**: Furthermore this enhancement of VarKarst neglects 30 years in the developments of land surface models. These models do not represent hydrological processes and even less karst systems, and are rightfully criticized for this. But they have specialized on the surface/atmosphere exchanges and in particular the simulation of evaporation, vegetation processes and infiltration. At no moment do the authors refer to developments in one of the three leading land surface models (JULES, ORCHIDEE and CLM) or their application to the 4 FLUXNET stations used here. A simple Google search would have shown to the authors that these open-access codes (Thanks in great part to GMD !) perform much better on these sites and do not require the tuning of so many parameters. Furthermore they are designed to be applicable at the large scale. I would recommend to reject the paper and encourage the authors to download one of the above mentioned land surface models and couple it to VarKarst. This would produce a model for these sensitive hydrological regions which is much more robust and produces more credible result for the impact of climate and land-cover changes. I am sorry to have to make such a harsh recommendation to GMD and in the following I will detail where I believe the basic assumptions of the authors to be wrong and where the usage of developments made for land surface models would help.

**REPLY 2**: For clarity, we structured our reply in three parts: (1) we explain why we chose to develop V2karst as an evolution of a parsimonious hydrological model, rather than using more complex land

surface models, (2) we attempt to compare the performance of V2Karst and land surface models at the four FLUXNET sites used in our study and (3) we specify the changes that we introduced in the revised version of the manuscript to clarify these points. We do not discuss V2karst representations here, as we will explain them more specifically in REPLY 3 (Penman-Monteith equation), REPLY 6 (soil water balance), and REPLY 9 (interception).

**1. Reasons for developing a parsimonious hydrological model**

Land Surface Models are undeniably crucial tools because they include state-of-the-art scientific understanding of moisture and energy processes. However, parallel to the development of land surface models, a wealth of studies have drawn attention to the problem of dealing with model complexity in the context of natural systems. In fact, in natural systems, controlled experimentation to ascertain model formulations is not possible and model components and parameters tend to be poorly defined, especially at large-scales, because of a lack of data and knowledge (e.g. Young et al., 1996; Abramowitz et al., 2008; Beven and Cloke, 2012; IPCC, 2013, Chapter 9, pp790-791; Hong et al., 2017; Haughton et al., 2018). For example, Beven and Cloke (2012) highlight the fact that more complex models do not necessarily produce more robust predictions, presumably because of our lack of knowledge of natural processes and because of the uncertainty in estimates/observations of the variables needed to run such complex models. Therefore, regarding future modelling challenges, Beven and Cloke (2012) have argued that understanding which parameterisation may be more appropriate and assessing model uncertainties may be more relevant than further increasing the detail of process representation.

A recent study published in GMD (Haughton et al., 2018) argues that: "In general, numerical LSMs [Land Surface Models] have become increasingly complex over the last 5 decades, expanding from basic bucket schemes to models that include tens or even hundreds of processes involving multiple components of the soil, biosphere, and within-canopy atmosphere. Model components may have been added on to existing models without adequate constraint on component parameters (Abramowitz, 2013) or without adequate system closure (Batty and Torrens, 2001). New component parameters may be calibrated against existing model components, leading to problems of equifinality (Medlyn et al., 2005), non-identifiability (Kavetski and Clark, 2011), and epistemological holism (Lenhard and Winsberg, 2010). These problems can often only be overcome by ensuring that each component is itself well constrained by data and numerically stable. As noted earlier, these conditions rarely exist for any given component." In fact, although land surface models have a strong physical basis, they also include many empirical functions that are typically difficult to constrain (Mendoza et al., 2015). More critically, the fact that land surface models include a large number of parameters (many of which hard-coded, as highlighted in Cuntz et al. (2016) or Mendoza et al. (2015)) hampers an exhaustive assessment of uncertainty and sensitivity of model predictions (Young et al., 1996). We further discuss the issue of parameter estimation in land surface models in REPLY 15. Moreover, the study by Haughton et al. (2018, GMD) shows that land surface models can be outperformed by simple empirical models, in line with the results of previous studies conducted within the Land Surface Model Benchmarking Evaluation Project (PLUMBER, Best et al., 2015).

Importantly, land surface models simulate a large range of different fluxes (e.g. sensible heat, latent heat, ground heat flux, radiation, runoff, CO2) and the sensitivity of model parameters has been reported to vary depending on the simulated flux considered (Cuntz et al., 2016; Rosero et al., 2010; Rosolem et al., 2012). For instance, Cuntz et al. (2016) showed that a large number of parameters of the Noah land surface model are non-influential or have a very small influence on total simulated runoff. This means that parts of the land surface models may be simplified when the objective is to simulate hydrological

variables. In this sense, Hong et al. (2017) highlighted the fact that model development should account for the model intended uses.

V2Karst aims to simulate seasonal and annual groundwater recharge, because these variables are appropriate to characterise the amount of renewable groundwater, and hence the amount of groundwater available to human consumption and ecosystems (e.g. Scanlon et al., 2006; Döll and Fiedler, 2008; Wada et al., 2012). As generally done in hydrological models (Table A1 of our manuscript), V2Karst focuses on solving the water balance, while it does not solve the energy balance, as land surface models do. In fact, V2Karst is not meant to be used for assessing the energy fluxes (radiation, sensible heat, latent heat and ground heat flux). An additional motivation for us not to solve the energy balance is that its inclusion increases model complexity and computational cost tremendously, which makes it difficult to perform a full uncertainty and sensitivity analysis to assess the adequateness of the different model components. By focusing on the water balance instead and by using parsimonious representations, we enable all components of V2Karst to be subject to uncertainty and sensitivity analysis, as we have explained in Sect. 2.1 p6 L20-30 in our manuscript.

**2. Performance of Land Surface Models at the four FLUXNET sites**

We are not aware of studies that would allow us to directly infer that JULES, ORCHIDEE and CLM have better performance compared to V2Karst at the four FLUXNET sites, as stated by the reviewer. We found six studies in which the JULES, ORCHIDEE, CLM or Noah land surface models were applied at some of the FLUXNET sites we have used (Anav et al., 2010; Davin et al., 2011; Zhao et al., 2012; Kuppel et al., 2012; Van den Hoof et al., 2013; Chaney et al., 2016). In the following paragraphs, we explain in detail that either the results of these studies cannot be compared to our results, or that the performance of the land surface models appears to be similar to or slightly inferior to those of V2Karst.

The results of four of these studies cannot be compared with our results. The study by Chaney et al. (2016) does not specifically report performance results for any of the four FLUXNET sites. The study by Davin et al. (2011) analysed a full Regional Climate Model including the CLM model and did not present performance results regarding latent heat and soil moisture simulations, which are the two variables we analysed in our manuscript. The study by Kuppel et al. (2012) did not analyse the bias or the correlation coefficient between measured and simulated latent heat/evapotranspiration, while these two metrics were used in our analyses (Sect 4.1 of our manuscript). In fact, both metrics are important to characterise the hydrological performance of the recharge model, since the bias assesses the performance in reproducing the overall water balance, while the correlation coefficient assesses the consistency between the temporal pattern of simulated and observed latent heat/evapotranspiration. Finally, the study by Anav et al. (2010) did not report quantitative performance metrics for the individual sites.

In the two "comparable" studies ( Zhao et al., 2012; Van den Hoof et al., 2013), land surface models show similar or slightly inferior performance compared to V2Karst, although it is difficult to make a fair comparison because the time periods analysed were not the same as in our manuscript. Zhao et al. (2012) assessed the squared correlation coefficient  $R^2$  between monthly observed and simulated latent heat at the French 2 site (Puéchabon) using the ORCHIDEE model and local forcing data. They found a value of  $R^2$  of 0.59, i.e. a correlation coefficient of  $\sqrt{0.59} = 0.77$  (Zhao et al., 2012, fourth row in Figure 6). This is comparable to the performance of the V2Karst model, since we identified simulations for which the correlation coefficient between monthly simulated and observed ET was higher than 0.77 (Figure 4 in our manuscript). We wish to mention that there appears to be a mistake in the labels in Figure 6 in Zhao et al. (2012) and therefore we are not completely sure we have read the figure correctly.

Finally, in Van den Hoof et al. (2013), the JULES model was tested at the German site (Hainich) using local forcing data. The bias between observed and simulated evapotranspiration for the best performing version of the model is around 29% (this percentage is inferred from Figure 2.b in Van den Hoof et al. (2013), which reports that the mean annual observed ET is 280 mm.y-1 and the simulated mean annual ET is 360 mm.y-1). The bias is therefore above the limit of acceptance of 20% that we have used in our manuscript. Van den Hoof (2013) did not provide the monthly correlation values for individual sites, but we can infer that, for the German site, the monthly correlation coefficient is between 0.7 and 1 (Van den Hoof et al., 2013, Figure 5.b), which is comparable to our study.

**3. Changes in the revised version of the manuscript**

Based on the Reviewer's comment, we realise that in our manuscript we have not stated clearly enough our philosophy in developing V2Karst in line with other large-scale hydrological models. We clarified this point in the introduction p5 L13-19 (in the paragraph that states the objectives of the study). We also added further justifications and references to support our choice of developing a parsimonious model in Sect. 2.1 p6 L20-30.

We agree that for completeness our manuscript should also refer to the literature on land surface modelling. In the revised version of our manuscript we include a discussion of the ET components of land surface models in Sect. 2.1 p7 L15-21. We also explain in this paragraph why we chose to develop a hydrological model.

The paragraph in the conclusions section p34 L16-25 that was already present in the previous version of our manuscript summarises our modelling strategy.

However, we would avoid including a comparison between the performance of the V2Karst model and other land surface models at FLUXNET sites because, as shown in our point (2) above, it is difficult to make a fair comparison between different studies that used different metrics and different time periods to evaluate the models.

**COMMENT 3:** Rational to explicitly represent land cover properties :**

It is laudable for the authors to use the Penman-Monteith formulation for potential evaporation. But should they have paid attention to its derivation, they would have noted that it only provides potential evaporation over a infinitesimal time intervals as it assumes that atmospheric variables and surface states do not evolve through other processes. A constant Rn(t) or ra,can(t) over the course of the day is a very unsatisfactory assumption, especially under a changing climate. Because of very contrasted impact of changing atmospheric composition on long-wave and short-wave radiation, we can encounter the same Rn but with very different radiation balance, turbulent fluxes and surface temperatures. The authors will find in the literature a number of paper which examine the impact of climate change on the different potential evaporation formulation. They all recommend to use sub-diurnal solutions because of the modified diurnal dynamics.

**REPLY 3:** We are aware of the fact that the Penman-Monteith equation has been determined over an infinitesimal time step (e.g. Shuttleworth, 2012) and that, therefore, using sub-daily rather than daily time step for the calculation of the equation provides conditions that are closer to its theoretical derivation. Nevertheless, Penman-Monteith equation has been shown to be applicable at daily time step and is widely used at daily time step.

In particular, Allen et al. (2006) commented on applications of the Penman-Monteith reference crop framework, that was designed by the Food and Agriculture Organization of the United Nations (FAO) and the American Society of Civil Engineers (ASCE). The framework has standardised the use of the Penman-Monteith equation for both hourly and daily time step (Allen et al., 1998; Walter et al., 2005) with a focus on agricultural crop. Allen et al. (2006) stated the following: *"The favorable performance"*

of the PM equation in many studies, when applied with 24-h (and even monthly) time steps, is somewhat surprising, since the formulation of the combination equation (combined energy balance and aerodynamic components) theoretically requires weather inputs on a nearly instantaneous basis. The general consistency and accuracy of the PM method for 24-h time steps speaks to the combination equation's robustness in estimating evaporative behavior given a particular set of meteorological conditions." More recent papers still support both hourly and daily applications of the Penman-Monteith reference crop framework (e.g. Pereira et al., 2015). The daily framework has been included for instance in an R package for simulating evapotranspiration at daily time step (Guo et al., 2016).

The Penman-Monteith equation is widely used at daily time step. Among its applications, we can cite the Moderate Resolution Imaging Spectroradiometer (MODIS) Evapotranspiration product (Mu et al., 2011; Running et al., 2017). Furthermore, the Penman-Monteith equation is implemented in the large-scale hydrological models PCR-GLOBWB, MacPDM and VIC (these models are reviewed in Table A1-A3 of our manuscript). These models have been applied at daily time step in climate change impact studies. For instance, Gosling and Arnell (2016) used MacPDM, Sperna Weiland et al. (2012) used PCR-GLOBWB and Beyene et al. (2010) used VIC at daily time step. These three hydrological models are also included in the Inter-Sectoral Impact Model Intercomparison Project (ISI–MIP, Warszawski et al., 2014) that aims to assess the impact of climate change. In that context, daily simulations of PCR-GLOBWB, MacPDM and VIC are analysed for instance by Gosling et al. (2017).

**Changes in the revised version of the manuscript**

Given all the above, we think that applying the Penman-Monteith equation at daily time step is one possible choice that is consistent with a wide range of published studies. Additionally, our comparison between hourly and daily time step performed in the revised version of the manuscript (as explained in REPLY 1) demonstrates that it is reasonable to apply V2Karst at daily step for our application. This comparison is briefly discussed in the manuscript p9 L15-19, while detailed results are reported in Sect. S8 of our Supplementary material.

Additionally, our manuscript refers to other hydrological models that applied the Penman-Monteith equation at a daily time (Table A2). In the revised version of the manuscript, when discussing the Penman-Monteith equation, we added a reference to Allen et al. (2006) and Pereira et al. (2015) to further justify the use of the Penman-Monteith equation at both sub-daily and daily time step p15

---

## Referee Report (RR1)

Review comments on the revised version: **"V2Karst V1.0: A parsimonious large-scale integrated vegetation-recharge model to simulate the impact of climate and land cover change in karst regions"** by F. Sarrazin, A. Hartmann, F. Pianosi, R. Rosolem, and T. Wagener.

In general, I agree with the way the authors addressed my main comments. While I am OK with the responses the authors presented in the response letter, I still do not totally agree with the general approach used for the prediction of karst recharge implemented in VarKarst/V2Karst. In my opinion (!), the repeated citation of work done by the same research group does not validate the applied approach and also does not foster discussion about pro, cons and restrictions of the model. I am aware of the need for distinguishing between diffuse and direct recharge. The applied approach in VarKarst/V2Karst and, hence, the recharge estimations is definitely a step into the right direction and an improvement to the frequently applied percentage breakdown of total precipitation. The estimation of recharge (especially direct recharge -> short-term process) by using (hydrological) budget calculation might be applicable for certain conditions (Hartmann et al, 2012: Both [experiments] highlight the presence of large water storages in the soil of the epikarst, which need to become saturated before the drips activate) but might also fail in other karstic system with different properties. Therefore, a description of typical karstic features in the study area is demanded.

Nevertheless, since the paper focuses on the enhancement of an already existing model approach (supported by several publications), the manuscript can be considered for publication.

References

Hartmann, A., Lange, J., Weiler, M., Arbel, Y. and Greenbaum, N.: A new approach to model the spatial and temporal variability of recharge to karst aquifers, Hydrol. Earth Syst. Sci., 16, 2219-2231, 2012.

---

## Author Response (AR2)

Dr Fanny Sarrazin
Research Associate, Water and Environmental Engineering
Department of Civil Engineering, University of Bristol
Queen's Building, University Walk, Bristol BS81TR, UK
fanny.sarrazin@bristol.ac.uk

Dear Dr Roche,

Please find enclosed a revised version of our manuscript "V2Karst V1.1: A parsimonious large-scale integrated vegetation-recharge model to simulate the impact of climate and land cover change in karst regions". We wish to thank you and the Reviewers' for the time and effort taken to review our manuscript. We made some final adjustments to the manuscript to address the concerns of the Reviewer and to correct a few remaining errors. We have enclosed in this document a marked-up version of the manuscript that highlights in red the changes we made. Specifically, we report below these changes:

- We clarified the representation of recharge in the VarKarst/V2Karst model. We added a few sentences p5 L32–p6 L8. Specifically, we stressed the fact that recharge production in VarKarst/V2Karst does not require the saturation of the epikarst layer, as understood by the Reviewer.

- We added a few sentences p27 L10–15 to discuss the limitations of the V2Karst model (that are mainly due to the limited amount of information available at large-scales) and the need for more information on typical karstic features over large domains to improve process representation in future developments of the V2Karst model. These two issues were raised by the Reviewer and we took the opportunity to better address them in the manuscript.

- We adjusted the versioning of the model (we replaced 'V2Karst V1.0' by 'V2Karst V1.1'). In fact, our revisions to the manuscript led to introducing new features into the model to allow for sub-daily simulations (only daily simulations were possible with the initial version of the model). We released a new version of the model that implement these improvements (V2Karst V1.1, available through a Github repository as indicated in the section 'code availability' of the manuscript).

- We made a few minor editorial changes. In particular, we modified the figures so that the labels of the panels are consistent with the GMD guidelines and we removed the dots in the units of measurement.

Again, we thank you and the Reviewer for appreciation of our manuscript.

Sincerely,

Fanny Sarrazin

[revised manuscript text omitted]